# Data-driven discovery of dimensionless numbers and governing laws from scarce measurements

Xiaoyu Xie[1], Arash Samaei [1], Jiachen Guo[2], Wing Kam Liu [1] ✉ & Zhengtao Gan[1,3] ✉

Dimensionless numbers and scaling laws provide elegant insights into the characteristic properties of physical systems. Classical dimensional analysis and similitude theory fail to identify a set of unique dimensionless numbers for a highly multi-variable system with incomplete governing equations. This paper introduces a mechanistic data-driven approach that embeds the principle of dimensional invariance into a two-level machine learning scheme to automatically discover dominant dimensionless numbers and governing laws (including scaling laws and differential equations) from scarce measurement data. The proposed methodology, called dimensionless learning, is a physics-based dimension reduction technique. It can reduce high-dimensional parameter spaces to descriptions involving only a few physically interpretable dimensionless parameters, greatly simplifying complex process design and system optimization. We demonstrate the algorithm by solving several challenging engineering problems with noisy experimental measurements (not synthetic data) collected from the literature. Examples include turbulent Rayleigh-Bénard convection, vapor depression dynamics in laser melting of metals, and porosity formation in 3D printing. Lastly, we show that the proposed approach can identify dimensionally homogeneous differential equations with dimensionless number(s) by leveraging sparsity-promoting techniques.

All physical laws can be expressed as dimensionless relationships with fewer dimensionless numbers and in a more compact form[1]. Dimensionless numbers are power-law monomials of some physical quantities[2]. A dimensionless number has no physical dimension (such as mass, length, or energy), which provides the property of scale invariance, i.e., dimensionless numbers are invariant when the length scale, time scale, or energy scale of the system varies. More than 1200 dimensionless numbers have been discovered in an extremely wide range of fields, including physics and physical chemistry; fluid and solid mechanics; thermodynamics; electromagnetism; geophysics and ecology; and engineering[3].

There are several significant advantages to describing a physical process or system using dimensionless numbers, including reducing the number of variables, enabling cross-scales experiments, and increasing physical interpretability. First, using dimensionless numbers can considerably simplify a problem by reducing the number of variables that describe the physical process, thereby reducing the number of experiments (or simulations) required to understand and design the physical system. The Reynolds number (Re), for example, is a well-known dimensionless number in fluid mechanics named after Osborne Reynolds, who studied fluid flow through pipes in 1883[4]. The Reynolds number is defined as a power-law based on four physical

[1]Department of Mechanical Engineering, Northwestern University, Evanston, IL 60208, USA. [2]Theoretical and Applied Mechanics, Northwestern University, Evanston, IL 60208, USA. [3]Present address: Department of Aerospace and Mechanical Engineering, The University of Texas at El Paso, El Paso, TX 79968, USA. ✉e-mail: w-liu@northwestern.edu; zgan@utep.edu

quantities: fluid density, average fluid velocity, the diameter of the pipe, and dynamic fluid viscosity. The flow characteristics (laminar or turbulent) in a pipe are best determined by the Re rather than the four individual dimensional quantities. Second, the scale-invariance property of dimensionless numbers plays a critical role in similitude theory[5]. Many small-scale experiments have been designed to understand and predict the behaviors of full-scale applications in aerospace[6], nuclear[7], and marine engineering[8], where full-scale applications are typically extremely expensive and even dangerous. All dimensionless numbers should be identical between the small-scale and full-scale experiments, resulting in perfect geometry, dynamic, and kinematic similarities between the two scales. Third, dimensionless numbers are ratios of two forces, energies, or mechanisms. Thus, they are physically interpretable and can provide fundamental insights into the behavior of complex systems. For example, the Péclet number (Pe) represents the ratio of the convection rate of a physical quantity by flow to the gradient-driven diffusion rate, which enables order-of-magnitudes analysis for the transport phenomena of a process.

Despite the scientific significance and widespread use of dimensionless numbers, discovering new dimensionless numbers and their relationships (i.e., scaling laws) from experiments remains challenging, especially for a complex physical system lacking complete governing equations. A traditional solution is dimensional analysis[2] based on Buckingham's Pi theorem[9], which provides a systematic approach to examining the units of a physical system and forming a set of dimensionless numbers that satisfy the principle of dimensional invariance[10]. However, dimensional analysis has several well-known limitations. First, the dimensionless numbers derived are not unique. Buckingham's Pi theorem[9], from the standpoint of mathematics, provides a linear subspace of exponents that produces dimensionless numbers. Any basis for the subspace is equally valid. Thus, it fails to identify the dimensionless numbers that are dominant for the physical system given a specific choice of basis. Second, dimensional analysis alone cannot reveal the mathematical relationship between dimensionless numbers (i.e., the scaling law). A common approach to establishing the scaling law is to leverage the results of the dimensional analysis with experimental measurements of the physical system. The experimental measurements are transformed into dimensionless numbers obtained through dimensional analysis and fitted onto a high-dimensional response surface to represent the scale-invariant relationship. However, because the dimensional analysis does not provide unique dimensionless numbers, this procedure is very time-consuming and heavily relies on the experience of domain experts to select a set of appropriate dimensionless numbers through a long process of trial and error.

These limitations could be overcome by integrating dimensional analysis with advanced data science and artificial intelligence (AI). Mendez and Ordonez introduced the SLAW (Scaling LAWs) algorithm to identify the form of a power law from experimental data (or simulation data)[11]. The proposed SLAW combines dimensional analysis with multivariate linear regressions. This approach has been applied to some engineering areas, such as ceramic-to-metal joining[11] and plasma confinement in Tokamaks[12]. However, for the sake of simplification, this algorithm assumes that the relationship between dimensionless numbers obeys a power law, which is invalid in many applications. Constantine, Rosario, and Iaccarino proposed a rigorous mathematical framework to estimate unique and relevant dimensionless groups[13,14]. Active subspace methods are connected to dimensional analysis, which reveals that all physical laws are ridge functions[14]. However, their method is only applicable to idealized physical systems. In these systems, experiments can be conducted for arbitrary values of the independent input variables (or dependent input variables with a known probability density function), and noises or errors in the input and output are negligible.

In this study, we propose a mechanistic data-driven approach, called dimensionless learning. This method consists of two main workflows to discover scientific knowledge from data. The first workflow embeds the principle of dimensional invariance (i.e., physical laws are independent of an arbitrary choice of basic units of measurements[1]) into a two-level machine learning scheme to automatically discover dominant dimensionless numbers and scaling laws from noisy experimental measurements of complex physical systems. This invariance incentivizes the learning of scale-invariant and physically interpretable low-dimensional patterns of complex high-dimensional systems. We demonstrate the first workflow by solving three challenging problems in science and engineering with noisy experimental measurements collected from the literature. The problems include turbulent Rayleigh–Benard convection, vapor depression dynamics, and porosity formation during 3D printing. In the second workflow, dimensionless learning is integrated with sparsity-promoting techniques (such as SINDy[15] and proposed symmetric invariant SINDy) to identify dimensionally homogeneous differential equations and dimensionless numbers from data. The analyses are performed on five differential equations with and without noisy data effect, including Navier–Stokes, Euler, vorticity equations, the governing equations for spring–mass–damper systems and dynamic loading beam systems.

## Results

### Turbulent Rayleigh–Bénard convection

In this section, we demonstrate the first workflow of the proposed dimensionless learning using a classical fluid mechanics problem: turbulent Rayleigh–Bénard convection. The goal is to directly rediscover the Rayleigh number (Ra) from experimental measurements. The Ra is named after Lord Rayleigh, who investigated a non-isothermal buoyancy-driven flow in 1916[16], which is now known as Rayleigh–Bénard convection. Turbulent Rayleigh–Bénard convection is a paradigmatic system to study turbulent thermal flow in a planar horizontal layer of fluid in a container heated from below. The internal fluid could develop complex turbulent dynamics due to the effects of buoyancy, fluid viscosity, and gravity (Fig. 1a).

The heat flux through the container, $q$, can be measured experimentally, which depends on the height of the container $h$, the temperature difference between the top and bottom surfaces $\Delta T$, gravitational acceleration $g$, and fluid properties such as thermal conductivity $\lambda$, thermal expansion coefficient $\alpha$, viscosity $\nu$, and thermal diffusivity $\kappa$. To obtain a causal relationship, we need to specify the dependent (i.e., output) and independent (i.e., input) variables from the physical quantities describing the system. To simplify the demonstration, we assume the form of the output variable as the Nusselt number $\text{Nu} = \frac{qh}{\lambda \Delta T}$ (a more general case using $q$ as the output will be presented later) and a list of physical quantities $\boldsymbol{p}$ as input variables. The causal relationship to be determined can be represented as

$$\text{Nu} = \frac{qh}{\lambda \Delta T} = f(h, \Delta T, \lambda, g, \alpha, \nu, \kappa) = f(\boldsymbol{p}). \tag{1}$$

This is a high-dimensional parameter space. To explore it, we collect an experimental dataset of turbulent Rayleigh–Bénard convection from two different articles[17,18], including 182 experiments with various input variables and corresponding output measurements (Fig. 1b). Many machine learning models can fit the data. However, the majority of them are black-box models, such as neural networks, with poor interpretability and physical insights. Alternatively, we aim to identify a low-dimensional scale-invariant scaling law that best represents the dataset. In the scaling law, the products of powers of the input variables $\boldsymbol{p}$ form a dimensionless number $\Pi$. Thus, the causal

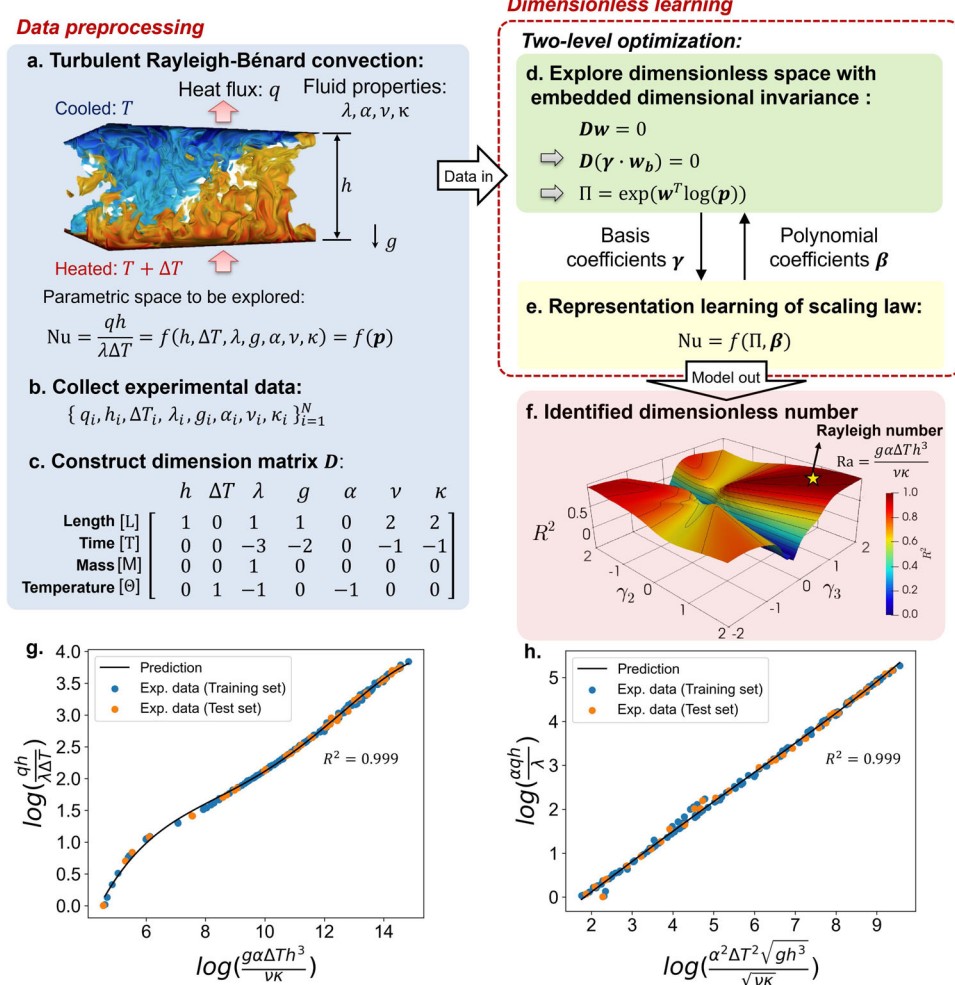

**Fig. 1 | The proposed dimensionless learning demonstrated on turbulent Rayleigh–Bénard convection. a** A schematic of Rayleigh–Benard convection[49] with associated physical quantities. **b** Collected experimental measurements. **c** Constructed dimension matrix $D$ of the input variables. **d** The first level of the two-level optimization scheme for training the coefficients $\gamma$ with respect to the computed basis vectors. **e** The second level of the two-level optimization scheme for optimizing the unknown coefficients $\beta$ in representation learning. **f** Explored dimensionless space with a measure of $R^2$. The location with the maximum $R^2$ is marked with a yellow star and corresponds to the classical Rayleigh number. **g** Identified one-dimensional scaling law between Nu and Ra. **h** Discovered linear scaling law between two identified dimensionless numbers.

relationship can be rewritten as follows:

$$\mathrm{Nu} = f_1(\Pi), \qquad (2)$$

$$\Pi = h^{w_1} \Delta T^{w_2} \lambda^{w_3} g^{w_4} \alpha^{w_5} \nu^{w_6} \kappa^{w_7}, \qquad (3)$$

where $\boldsymbol{w} = [w_1, \ldots, w_7]^{\mathrm{T}}$ denotes the powers that generate the dimensionless number and are to be determined. In this problem, we assume that the process is governed by a single input dimensionless number. Section 1.3 of the Supplementary Information (SI) provides an algorithm for determining the number of dimensionless numbers required from the data.

To embed the physical constraint of dimensional invariance, we perform dimensional analysis, i.e., the powers $\boldsymbol{w} = [w_1, \ldots, w_7]^{\mathrm{T}}$ need to satisfy a linear system of equations

$$\boldsymbol{Dw} = 0, \qquad (4)$$

where $\boldsymbol{D}$ is the dimension matrix of the input variables (Fig. 1c). Each column of the dimension matrix is the dimension vector of the corresponding variable. The dimension vector represents the exponents of the physical quantity with respect to the fundamental dimensions. It is worth noting that there are only seven fundamental dimensions in nature: mass [M], length [L], time [T], temperature [Θ], electric current [I], luminous intensity [J], and amount of substance [N][19]. All of the other dimensions are power-law monomials of the fundamental dimensions[1]. In this problem, we use four fundamental dimensions: [M], [L], [T], and [Θ] (Fig. 1c). The dimension matrix includes the physical dimensions of the input variables. The linear system of equations $\boldsymbol{Dw} = 0$ ensures that the power-law monomial of the input variables (Eq. (3)) is dimensionless[20]. Since the linear system is underdetermined (i.e., the number of unknown variables exceeds the number of equations), there are infinitely many solutions, indicating that the dimensional analysis can yield infinitely many forms of dimensionless numbers. Furthermore, we can represent the solutions of the linear system (Eq. (4)) as linear combinations of three basis vectors $\boldsymbol{w_{b1}}$, $\boldsymbol{w_{b2}}$, and $\boldsymbol{w_{b3}}$

$$\boldsymbol{w} = \gamma_1 \boldsymbol{w_{b1}} + \gamma_2 \boldsymbol{w_{b2}} + \gamma_3 \boldsymbol{w_{b3}}, \qquad (5)$$

where $\boldsymbol{\gamma} = [\gamma_1, \gamma_2, \gamma_2]^{\mathrm{T}}$ are the coefficients with respect to the three basis vectors in this case. The number of basis vectors is equal to the number of input variables (seven in this case) minus the rank of the dimension

matrix (four in this case). This formula aligns with the Buckingham's PI theorem[9]. Since the basis vectors can be computed using Eq. (4) (an algorithm for computing basis vectors is provided in Section 1.2 of the SI), the basis vectors' coefficients (or simply "basis coefficients") are the unknowns to be determined. For this case, a set of computed basis vectors is as follows:

$$\boldsymbol{w}_{b1} = [0, 0, 0, 0, 0, 1, -1]^{\mathrm{T}}, \tag{6}$$

$$\boldsymbol{w}_{b2} = [0, 1, 0, 0, 1, 0, 0]^{\mathrm{T}}, \tag{7}$$

$$\boldsymbol{w}_{b3} = [3, 0, 0, 1, 0, -2, 0]^{\mathrm{T}}. \tag{8}$$

Once the basis coefficients $\gamma_1$, $\gamma_2$, and $\gamma_3$ are obtained, the form of the dimensionless number $\Pi$ can be determined by Eqs. (3) and (5) (Fig. 1d).

To determine the values of the basis coefficients using the collected dataset, a model representing the scaling relation between the input and output dimensionless numbers is required, which introduces another set of unknown parameters $\boldsymbol{\beta}$ (i.e., the representation learning shown in Fig. 1e). In this case, we use a fifth-order polynomial model (more advanced models, such as tree-based models and deep neural networks, are optional depending on the complexity of the problem to be solved; see the section "Porosity formation in 3D printing of metals" of the paper and Section 4 of the SI for more demonstrations). The polynomial model can be expressed as

$$\mathrm{Nu} = \beta_0 + \beta_1 \Pi + \beta_2 \Pi^2 + \dots + \beta_5 \Pi^5, \tag{9}$$

where $\boldsymbol{\beta} = [\beta_0, \beta_1, \dots, \beta_5]^{\mathrm{T}}$ denotes polynomial coefficients that represent the scaling relation.

We design an iterative two-level optimization scheme to determine the two sets of unknown parameters in the regression problem, namely the basis coefficients $\boldsymbol{\gamma}$ and polynomial coefficients $\boldsymbol{\beta}$. The optimization scheme includes multiple iterative steps. At each step, we adjust the first-level basis coefficients $\boldsymbol{\gamma}$ while holding the second-level polynomial coefficients $\boldsymbol{\beta}$ constant, and then optimize the second-level polynomial coefficients $\boldsymbol{\beta}$ while keeping the first-level basis coefficients $\boldsymbol{\gamma}$ constant. This process is repeated until the result is converged, that is, the values of $\boldsymbol{\gamma}$ and $\boldsymbol{\beta}$ remain unchanged. There are several advantages to the proposed two-level approach over a single-level approach that combines the two sets of unknowns together during optimization. We can use different optimization methods and parameters (such as the learning rate) for these two-level models to significantly improve the efficiency of the optimization. More importantly, we can utilize physical insights to inform the learning process. The first-level basis coefficients $\boldsymbol{\gamma}$ have a clear physical meaning, which is related to the powers that produce the dimensionless number. Thus, those values have to be rational numbers to maintain dimensional invariance. Moreover, their typical range is limited. It is worth noting that the absolute values of the coefficients in most of the dimensionless numbers and scaling laws are less than four[1]. To leverage those physical insights or constraints, we design several methods for optimizing the first-level basis coefficients, including a simple grid search (used in this section) and a much more efficient pattern search (Section 4.2 of the SI). For the second-level coefficients, we conduct multiple standard representation learning methods, including the polynomial regression used in this section, tree-based extreme gradient boosting (XGBoost[21]) used in the section "Porosity formation in 3D printing of metals", and general gradient descent method (Section 4.1 of the SI). Details on the two-level optimization framework are provided in Section 4 of the SI.

We illustrate the first-level grid search for $\gamma_2$ and $\gamma_3$ with values ranging from −2 to 2 and 100 grids for each basis coefficient (Fig. 1f). We set $\gamma_1$ to 1 to avoid the identification of equivalent dimensionless numbers with different powers and reduce the computational cost. For each $\boldsymbol{\gamma}$ in the dimensionless space, the polynomial coefficients $\boldsymbol{\beta}$ are trained based on the collected data. The dataset is divided into an 80% training set and a 20% test set. The coefficient of determination ($R^2$) of the test set is shown in Fig. 1f as a measure of learning performance. We can identify a unique point with the maximum $R^2$ (0.999) from Fig. 1f (marked as a yellow star), where $\gamma_1 = \gamma_2 = \gamma_3 = 1$. Using these optimized basis coefficients, the expression of the dominant dimensionless number can be identified as

$$\Pi = \frac{g \alpha \Delta T h^3}{\nu \kappa}. \tag{10}$$

This form is identical to the classical Rayleigh number, indicating that the proposed dimensionless learning can directly rediscover the well-known dimensionless number from data. Moreover, we demonstrate that for the given parameter list, the Rayleigh number is the unique dimensionless number to best fit the dataset because there is only one global maximum of $R^2$ within the dimensionless space (Fig. 1f). The log–log scaling relation between Ra and Nu is a simple one-dimensional pattern in which all the data points collapse onto a single curve (Fig. 1g).

The proposed dimensionless learning can deal with dimensional output variables as well. A combination of input variables with the same dimension as the output variable can be searched to non-dimensionalize the output variable (the detailed algorithm for output non-dimensionalization is provided in Section 1.3 of the SI). Using the heat flux $q$ as the output variable (rather than Nu, which was used in the previous case study), the dimensionless space is expanded, allowing for the discovery of more dominant dimensionless numbers and scaling laws. We discover a new set of dimensionless numbers to best represent ($R^2 = 0.999$) the collected experimental measurements. More interestingly, the identified log-log scaling relation between dimensionless numbers is almost linear (Fig. 1h). This finding could lead to new physical insights into the complex turbulent Rayleigh–Bénard dynamics.

## Vapor depression dynamics in laser–metal interaction

Another challenging problem in the application of dimensionless learning is laser–metal interaction dynamics. People have been curious about the physical responses of a metallic material to high-power laser irradiation since 1964 when Patel invented an electric discharge $CO_2$ laser[22] that was dramatically scaled up in power shortly after. During the laser–metal interaction, a vapor-filled depression (called a keyhole) frequently forms on a puddle of liquid metal melted by the laser. The keyhole is caused by vaporization-induced recoil pressure, and its dynamics are inherently difficult to understand due to its complex dependence on many physical mechanisms. However, quantifying keyholes is critical because it is closely related to energy absorption and defect formation in a wide range of industrial and military applications, including laser-based materials processing and manufacturing[23], high-energy laser weapons[24], and aerospace laser-propulsion engines[25].

High-speed X-ray imaging made high-quality in-situ experimental data on keyhole dynamics available[26]. Using X-ray pulses, images of the keyhole region inside the metals can be recorded with micrometer spatial resolution[27]. The keyhole depth $e$ can be measured from the X-ray images (Fig. 2a), and it varies with the materials used and a number of process parameters, such as the effective laser power $\eta P$, the laser scan speed $V_s$, and the laser beam radius $r_0$. We collect a dataset of keyhole X-ray images from the literature, including 90

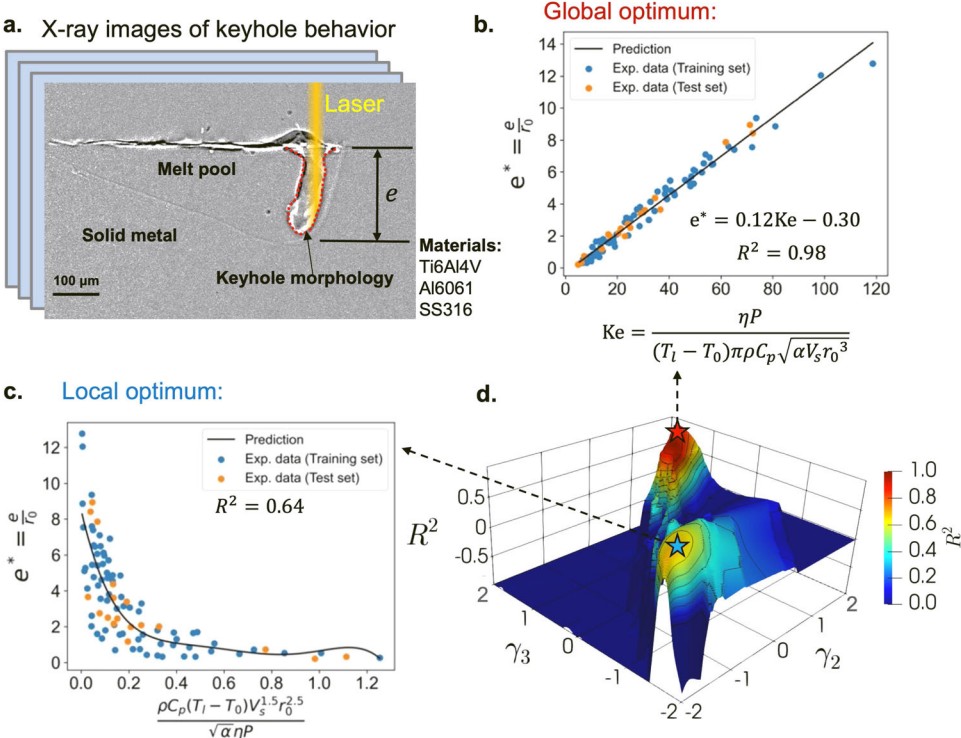

**Fig. 2 | Discover dimensionless numbers governing keyhole dynamics in laser–metal interactions. a** An illustrative X-ray image of keyhole morphology[28]. The dataset includes X-ray imaging experiments on three different materials. **b** Global optimum of the dimensionless space, which represents the scaling law between the keyhole aspect ratio and the identified keyhole number using

dimensionless learning. **c** Local optimum of the dimensionless space. **d** Dimensionless space using $\gamma_2$ and $\gamma_3$ as coordinates. The values of $R^2$ indicate the learning performance for the corresponding dimensionless number in the dimensionless space. Values of $R^2$ less than −1 are shown as −1.

experiments with various process parameters and three different materials: titanium alloy (Ti6Al4V), aluminum alloy (Al6061), and stainless steel (SS316)[23,28]. We represent a material using a set of material properties: the thermal diffusivity $\alpha$, the material density $\rho$, the heat capacity $C_\mathrm{p}$, and the difference between melting and ambient temperatures $T_\mathrm{l}$–$T_0$. Therefore, the causal relationship can be expressed as

$$e = f(\eta P, V_\mathrm{s}, r_0, \alpha, \rho, C_\mathrm{p}, T_\mathrm{l} - T_0). \qquad (11)$$

We can use the dimensionless learning described in the previous section to extract a low-dimensional scale-free relation from the parameter list. The dimension matrix $\boldsymbol{D}$ and computed basis vectors $\boldsymbol{w_{b1}}$, $\boldsymbol{w_{b2}}$, and $\boldsymbol{w_{b3}}$ of this problem are provided in Section 3 of the SI. We first demonstrate the grid search ranging from −2 to 2 with 100 grids for the first-level optimization and fifth-ordered polynomial regression for the second-level optimization. We set $\gamma_1$ to 0.5 and normalize the output variable as the keyhole aspect ratio $e^* = \frac{e}{r_0}$, which is a widely used dimensionless parameter to represent the keyhole characteristic[29]. By searching the dimensionless parameter space, we can find one local optimum in terms of the $R^2$ criteria, marked by a blue star ($R^2 = 0.64$) in Fig. 2d. The expression of the dimensionless number $\Pi = \frac{\rho C_p (T_l - T_0) V_s^{1.5} r_0^{2.5}}{\sqrt{\alpha} \eta P}$ is computed based on the basis coefficients $\gamma_2 = \gamma_3 = -1$. However, the data points are scattered, as shown in Fig. 2c, indicating that the dimensionless number located at the local maximum of the dimensionless space is not a good scaling parameter for this problem. The global optimum of dimensionless space, where $\gamma_2 = \gamma_3 = 1$, provides much better scaling behavior, with a 0.98 $R^2$ score (Fig. 2b). The dominant dimensionless number that emerged from the

keyhole dynamics is

$$\Pi = \frac{\eta P}{(T_l - T_0)\pi\rho C_\mathrm{p}\sqrt{\alpha V_\mathrm{s} r_0^3}}. \qquad (12)$$

This dimensionless number times $1/\pi$ is identified directly from data and has the same form as the newly discovered keyhole number Ke[28] (also known as normalized enthalpy[30]), which can be derived from heat transfer theory. In this paper, we call Eq. (12) as keyhole number Ke. Even if we use the dimensional variable $e$ as the output, the dimensionless learning algorithm still confirms the formation of the keyhole number (i.e., Eq. (12)) is unique and dominant for controlling the value of the keyhole aspect ratio. Details of the procedure and its results are provided in Section 5.1 of the SI. Using the identified dimensionless number, a simple scaling law emerges to control the keyhole aspect ratio, which simplifies the original high-dimensional problem into a univariate scaling law as

$$e^* = 0.12\mathrm{Ke} - 0.30. \qquad (13)$$

Providing a sufficient parameter list is critical for dimensionless learning. If one or more important quantities are omitted, it is impossible to achieve a high $R^2$ for learning and identifying the correct form of the dimensionless number(s). In Section 6.1 of the SI, we demonstrate that if we assume a parameter list that excludes the thermal diffusivity $\alpha$, the maximum $R^2$ over the dimensionless space is <0.80, which is much less than the value for the sufficient parameter list (i.e., Eq. (11)). Another scenario that frequently occurs in applications is that we consider more quantities than are necessary, including some irrelevant or unimportant quantities. In Section 6.2 of the SI, we demonstrate this scenario by considering one more quantity in the parameter list, such as the latent heat of melting $L_\mathrm{m}$ or the difference

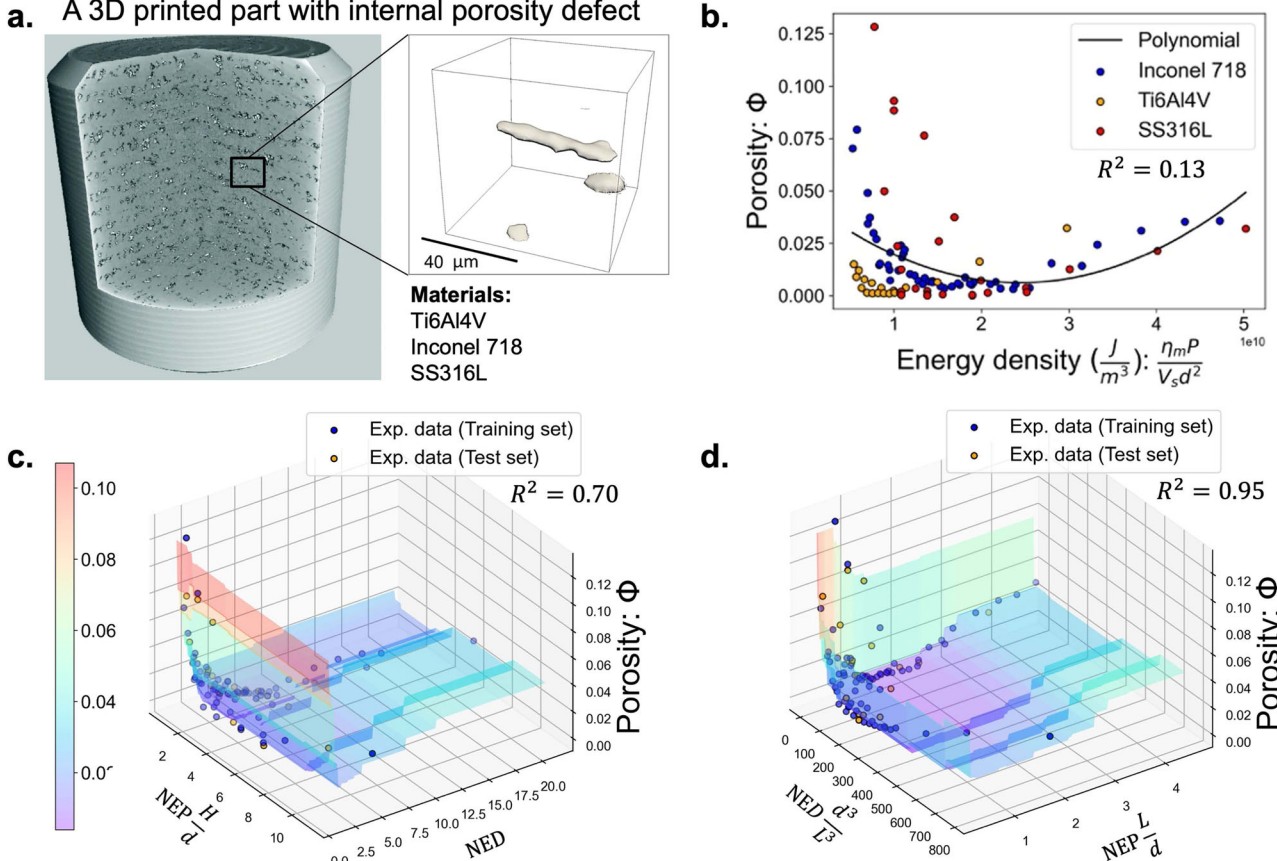

**Fig. 3 | Discover dimensionless numbers governing porosity formation during 3D printing. a** A schematic of a 3D printed metal part with internal porosity defects[50]. The dataset includes X-ray micro-computed tomography (micro-CT) measurements on three different materials. **b** Porosity measurements with varying energy density values, a traditionally combined parameter for correlating porosity with process parameters. **c** Identified 2D scaling relation combining both lacks of fusion and keyhole porosity with two discovered dimensionless numbers. **d** Another identified 2D scaling relation with a higher $R^2$ score. The reduced parameter space is simple to visualize and interpret, while the original high-dimensional problem is difficult since the porosity is governed by nine parameters.

between boiling temperature and ambient temperature $T_l - T_0$. The form of the keyhole number can still be identified in the scenario. Moreover, there are a few more dimensionless numbers, which consist of the added quantity, providing a high $R^2$ as the keyhole number. This implies that additional experiments are required to select the distinguished one among the identified dimensionless numbers.

In Section 4 of the SI, we provide two efficient algorithms, namely, gradient-based and pattern search-based two-level optimization schemes, to improve the efficiency of the optimization used in this section. These algorithms are especially useful for exploring a high-dimensional parameter space that contains many parameters to describe the physical system as well as several dimensionless numbers to construct the low-dimensional pattern.

**Porosity formation in 3D printing of metals**

Three-dimensional (3D) printing, also known as additive manufacturing, is a disruptive technology that produces three-dimensional solid objects from a digital file, introducing a new manufacturing paradigm[31]. In metal 3D printing, metallic parts are built layer by layer by local melting with a laser or electron beam and resolidifying metallic powders. 3D printing allows for remarkable freedom when it comes to designing local geometrical and compositional features. However, this process has a large number of parameters to consider when making a part, and it has a tendency to produce defects, such as internal porosity, during the printing process if inappropriate process parameters are used (Fig. 3a).

To extract elegant insights into the complex behavior of porosity formation in 3D printing, we collect an experimental dataset from six independent studies[32–37], including 93 3D printed parts with measured porosity volume fraction and various process parameters. Three different materials were used: titanium alloy (Ti6Al4V), nickel-based alloy (Inconel 718), and stainless steel (SS316L). The porosity volume fraction $\Phi$ depends on many process parameters and materials used in the experiments, which can be expressed as

$$\Phi = f(\eta_m P, V_s, d, \rho, C_p, \alpha, T_l - T_0, H, L), \qquad (14)$$

where $\eta_m P$ is the effective laser power input, $V_s$ is the laser scan speed, $d$ is the laser beam diameter, $\rho$ is the material density, $C_p$ is the material heat capacity, $\alpha$ is the thermal diffusivity, $T_l - T_0$ is the difference between melting and ambient temperatures, $H$ is the hatch spacing between two adjacent laser scans, and $L$ is the layer thickness of the metallic powders. It is a high-dimensional relation and is difficult to understand and visualize. Traditionally, some combined parameters, such as energy density $\frac{\eta_m P}{V_s d^2}$, are used to simplify this relation. However, the $R^2$ score of a polynomial model with energy density as input is very low (0.13), as shown in Fig. 3b. This indicates that a universal physical relation, which is valid for different materials and processing conditions, cannot be built by using the energy density alone because it is not a scale-free parameter. The form of the relation must be modified when the energy scale is changed in experiments with varying process parameters or materials.

We apply dimensionless learning to this challenging engineering problem and discover some dominant dimensionless numbers that provide a universal physical relation that remains accurate across all experimental conditions. Section 3 of the SI provides the dimension matrix and computed basis vectors for this case study. The two-level optimization applied in this problem includes a pattern search for the first level and an XGBoost method to capture the second-level relationships (Section 4.2 of the SI). We find that two dimensionless numbers are necessary to represent the dataset since no high value of the $R^2$ score (e.g., >0.5) can be achieved if we set only one dimensionless number in the training. A systematic algorithm for determining the number of dimensionless numbers required to govern a physical system is provided in Section 1.3 of the SI.

We identify several low-dimensional patterns using the scale-free property of data. They can achieve a high $R^2$ for both the training and test sets (a table summarizing the identified dimensionless numbers is provided in Section 5.2 of the SI). Interestingly, we identify another dimensionless number (besides the keyhole number), which has been discovered by the theory-driven approach[28,32]: the normalized energy density (NED) (Fig. 3c). It can be expressed as

$$\text{NED} = \frac{\eta_m P}{V_s \rho C_p (T_1 - T_0) H L}. \tag{15}$$

The NED represents the ratio of laser energy input within the powder layer to sensible heat of melting. This dimensionless number governs the lack of fusion porosity in metal 3D printing, which is a well-known porosity mechanism caused by insufficient laser energy input to fully melt the powder material[38]. The other dimensionless number in Fig. 3c is related to another porosity mechanism, namely keyhole porosity, caused by trapped bubbles of gas beneath the surface during the fluctuation of an unstable keyhole[27]. This dimensionless number is a modified normalized enthalpy product, i.e., $\text{NEP}\frac{H}{d}$, where the normalized enthalpy product NEP is proven to be related to keyhole instability, and an unstable keyhole with a high NEP could lead to keyhole pores[30]. The NEP can be expressed as

$$\text{NEP} = \frac{\eta_m P}{V_s \rho C_p (T_1 - T_0) d^2}. \tag{16}$$

Since the NEP is derived from the single-track laser scan condition[30], the modified term $\frac{H}{d}$ emerges to account for the effect of multiple-track scanning. Another identified low-dimensional pattern $\Phi = f(\text{NEP}\frac{L}{d}, \text{NED}\frac{d^3}{L^3})$ achieves an even higher $R^2$ (0.95), as shown in Fig. 3d. Two geometrical ratios ($\frac{L}{d}$ and $\frac{d^3}{L^3}$) are involved to maximize the fitting performance. These two ratios have clear physical meanings: the term $\frac{L}{d}$ means the linear ratio of powder bed layer thickness and laser beam diameter, while the term $\frac{d^3}{L^3}$ means the volumetric ratio of laser beam diameter and powder bed layer thickness. These two ratios account for the effect of multiple-track and multiple-layer scanning. By reducing high-dimensional parameter space, fewer experiments would be required to determine optimal processing conditions and parameters for new materials, easing the Edisonian burden endemic among current metal 3D printing practitioners.

## Vorticity form of dimensionless Navier−Stokes equation

In this section, we describe the second workflow of dimensionless learning: identifying dimensionally homogeneous differential equations and dimensionless numbers from time-varying data. This approach combines dimensionless learning with sparsity-promoting methods. Like the method discussed in the sections "Turbulent Rayleigh−Bénard convection", "Vapor depression dynamics in laser−metal interaction" and "Porosity formation in 3D printing of metals" for incorporating dimensional invariance into machine learning, we enhance the sparsity-promoting method SINDy with another

fundamental physical invariance, symmetric invariance. We refer to this physically enhanced SINDy as symmetric invariant SINDy.

Figure 4 shows a schematic of this workflow for identifying the underlying governing equation and dimensionless number(s) from simulation snapshots of the Kármán vortex street problem. This fluid mechanics problem involves three cylinders with diameters of $l$ (see Fig. 4a). Different fluid flow patterns can be obtained through simulations by changing fluid density $\rho$, dynamic viscosity $\mu$, inlet velocity $V$, and the pressure difference between upstream and downstream $p_0$.

In the first step (Fig. 4a), three CFD simulations are carried out to generate datasets for the discovery of the governing equation. The dataset for each simulation contains not only the above-mentioned geometry and fluid properties but also time-dependent variables (i.e., velocities $u$ and $v$ and vorticity $\omega$) in the spatiotemporal domain. Then, 4000 velocity and vorticity measurements from different locations and time steps are randomly sparsely sampled. A detailed description of data generation and preprocessing can be found in Section 7.1.1 of the SI.

Next, we apply symmetric invariant SINDy on the dataset for each simulation case to discover temporal governing equations (Fig. 4b). To incorporate symmetric invariance, we flip the original data along $y = x$ for each simulation case to obtain the transformed data. This is because we assume the governing equation should be invariant to the symmetric transformation along $y = x$. This assumption helps double the dataset for temporal governing equation discovery while incurring no additional computational cost to run more simulations. More information about this operation can be found in Section 7.1.2 of the SI.

Based on these measurements, a regression library is built to identify the governing equation using linear and quadratic terms for $u, v, \omega, \frac{\partial \omega}{\partial x}, \frac{\partial^2 \omega}{\partial x^2}, \frac{\partial^2 \omega}{\partial y^2}$. The regression library contains 29 terms in total. Detailed information on candidate terms is shown in Section 7.5 of the SI.

After preparing the regression library, the proposed symmetric invariant SINDy trains all the measurements from the original and transformed data together. This operation implicitly ensures that the symmetry terms have the same coefficients. That is, the coefficients for symmetry terms are physically constrained. For example, $u\frac{\partial \omega}{\partial x}$ and $v\frac{\partial \omega}{\partial y}$ can be regarded as symmetry terms and have the same coefficient as the symmetric invariant SINDy. See Section 7.1.2 of the SI for more detailed descriptions of this operation.

By optimizing the symmetric invariant SINDy for all cases, we obtain three temporary governing equations with only four non-zero regression coefficients (Fig. 4c). $\xi_{12}$ and $\xi_{19}$ are identical and close to a constant, while $\xi_6$ and $\xi_7$ are also identical but vary with parameters such as $\rho$, $\mu$, and so on. The following steps are to build a consistent parameterized governing equation that is valid in all simulations, as explained below.

Since the varying coefficients ($\xi_6$ and $\xi_7$) are due to changes in geometry and fluid properties, we apply dimensionless learning to find the expression for these two coefficients (Fig. 4d). The parametric space, which includes variables affecting the behavior of the dynamical system, to be explored for $\xi_6 = \xi_7$ can be expressed as follows:

$$\xi_6 = \xi_7 = f(\mu, \rho, V, l, p_0). \tag{17}$$

In contrast to the standard dimensionless learning, we simplify the representation function $f(\cdot)$ as a power law with a constant coefficient rather than a high-order polynomial, as applied in the sections "Turbulent Rayleigh−Bénard convection" and "Vapor depression dynamics in laser−metal interaction", or an XGBoost, as used in the section "Porosity formation in 3D printing of metals". This is because parametric differential equations usually consist of derivatives and/or derivatives multiplied by variable coefficients, which are power-law functions.

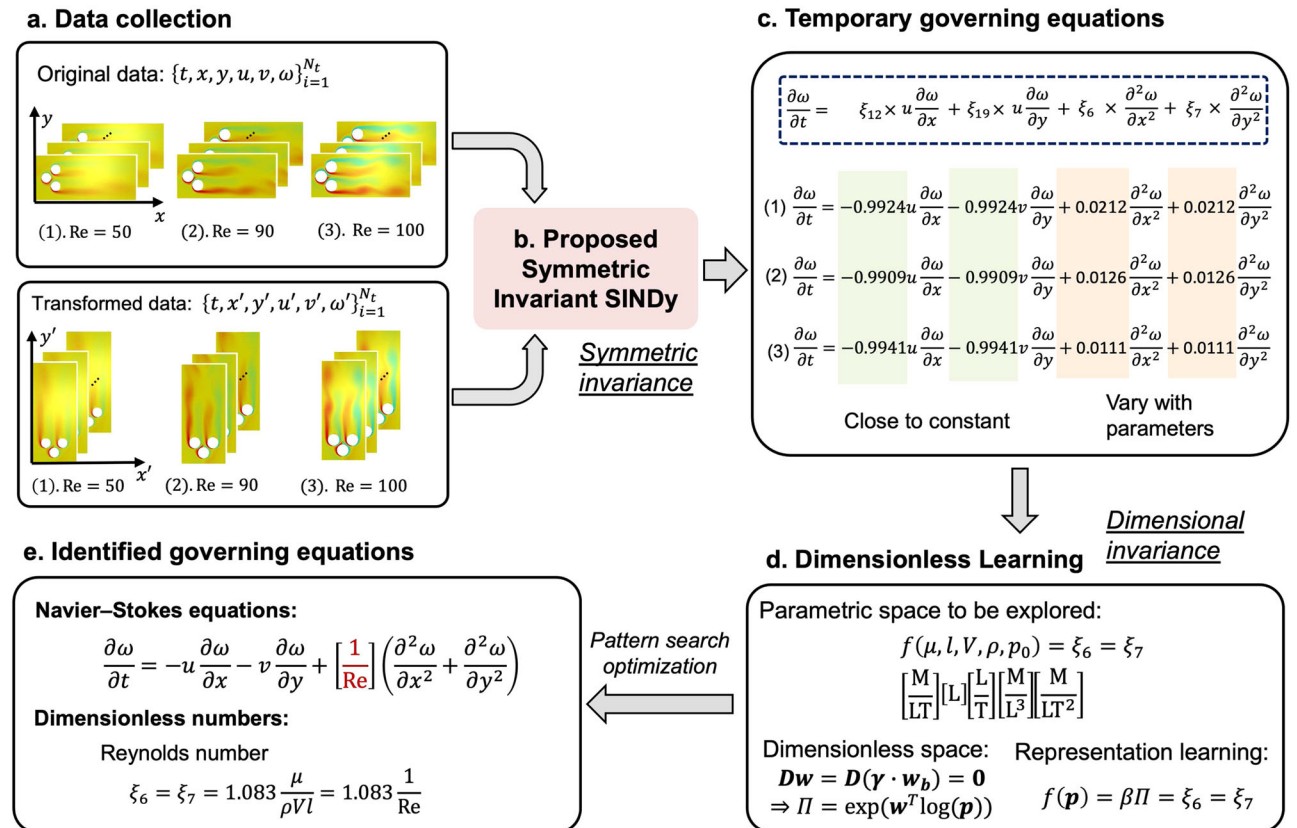

**Fig. 4 | Integration of dimensionless learning with symmetric invariant SINDy for identifying the Navier–Stokes equation with Reynold number. a** Original data are generated from parametric simulations. To achieve symmetric invariance, another set of transformed data is obtained by flipping the original data along $y = x$. **b** The original and transformed data are concatenated for symmetric invariant SINDy, which implicitly incorporates symmetric invariance into SINDy to ensure that symmetric invariant terms have the same coefficients. **c** The identified temporary governing equations for each simulation case were obtained by optimizing the symmetric invariant SINDy. Some of the coefficients are close to constant, while others vary depending on the simulation case. All the other candidate terms have zero coefficients. **d** Dimensionless learning is applied to identify an explicit expression for the varying coefficients. The parametric space to be explored includes five parameters. By incorporating dimensional invariance, we need to optimize basis coefficients $\gamma$ and fitting coefficient $\beta$. **e** Substituting the discovered regression coefficients (1/Re) into the temporary governing equation. In this step, a consistent dimensionally homogeneous governing equation, which is identical to the Navier–Stokes equation in the vorticity form, is obtained.

In this case, we choose the pattern search-based optimization to solve Eq. (17). A detailed description of this proposed optimization algorithm can be found in Section 4.3 of the SI. The identified expression for $\xi_6$ and $\xi_7$ is $1.083 \frac{\mu}{\rho Vl} \approx \frac{\mu}{\rho Vl}$ (Fig. 4e), which is the reciprocal of the well-known Reynolds number $\text{Re} = \frac{\rho Vl}{\mu}$. By substituting the constant regression coefficients for $\xi_{12}$ and $\xi_{19}$ and the discovered expression for $\xi_6$ and $\xi_7$ into the temporary governing equations, we obtain a consistent dimensionless governing equation in all cases as follows:

$$\frac{\partial \omega}{\partial t} = -u \frac{\partial \omega}{\partial x} - v \frac{\partial \omega}{\partial y} + \frac{1}{\text{Re}} \left( \frac{\partial^2 \omega}{\partial x^2} + \frac{\partial^2 \omega}{\partial y^2} \right), \qquad (18)$$

which is identical to the well-known vorticity form of the Navier–Stokes equation. This demonstrates the effectiveness of the proposed method in discovering governing equations and dimensionless number(s).

We further apply the proposed method to data with 1% Gaussian noise. Following the same procedure, the proposed method successfully identifies the correct governing equation as Eq. (18). The detailed results for noisy data are shown in Section 7.1.3 of the SI. More applications of the proposed method in fluid and solid mechanics and dynamics systems with and without noise are demonstrated in Section 7 of the SI.

## Discussion

The proposed dimensionless learning is a powerful technique to identify scientific knowledge from data at multiple levels: dimensionless number at the feature level, scaling law at the algebraic equation level, and governing equation at the differential equation level. Unlike purely data-driven approaches that easily suffer from overfitting on small or noisy datasets, this method incorporates fundamental physical knowledge of dimensional invariance and symmetric invariance as physical constraints or regularizations into data-driven models to perform well on limited and/or noisy data. The embedded physical invariance reduces the learning space and eliminates the strong dependence between variables. This method is a physics-based dimension reduction approach that represents features as dimensionless numbers and transforms data points into a low-dimensional pattern that is unaffected by units and scales. Thus, in addition to being applicable to limited and/or noisy data, the presented approach significantly improves the interpretability of representation learning because dimensionless numbers are physically interpretable. Lower dimension and better interpretability also allow for qualitative and quantitative analysis of the systems of interest. This has been demonstrated in three complex engineering problems in earlier sections.

Another advantage of the embedded dimensional invariance in dimensionless learning is improved generalization capability. To show this, in the vapor depression dynamics case, we compared the

performance of dimensionless learning and popular machine learning algorithms on unseen material data points. The proposed method achieves the best generalization in the test set, while all other algorithms only achieve a poor generalization. This improvement is due to ensuring geometric, kinematic, and dynamic similarities based on similitude theory within different systems. A detailed description of the generalization comparison can be found in Section 6.3 of the SI. Aside from dimensional invariance, we also used symmetric invariance in this study. The benefits of symmetric invariance are that it intrinsically ensures symmetry terms have the same coefficients and effectively reduces the number of learnable regression coefficients in SINDy.

Dimensionless learning is also very flexible in terms of choosing the representation learning function because of the proposed two-level optimization scheme. Since the first-level scheme guarantees dimensional invariance (or dimensional homogeneity), many representation learning methods can be used to capture scale-free relationships in the second-level scheme. We demonstrated polynomial and tree-based method XGBoost[21] in the previous sections. However, the capability of dimensionless learning can be improved by leveraging more methods, including deep neural networks[39], symbolic regression[40], and Bayesian machine learning[41].

The optimization of dimensionless learning is different from general regression optimization approaches because only dimensionless numbers with small rational powers are preferred, such as −1, 0.5, 1, or 2, etc. Therefore, instead of searching for the best basis coefficients with a lot of decimals like other neural network-based methods, such as DimensionNet[42], zero-order optimization methods are used in this work. It includes grid search or pattern search-based two-level optimization and can be more efficient in finding the best basis coefficients. No gradient information and learning rate are required and the choice of grid interval is more flexible. Even though these zero-order optimization approaches can get stuck in local minima, increasing the number of initial points can easily eliminate this issue. More detailed pros and cons of different optimization methods are described in Section 4.5 of the SI.

The proposed method divides the identification process of differential equations into two steps to identify consistent parameterized governing equations efficiently. The first step is to identify a temporary governing equation in which the regression coefficients can be a constant or variable depending on how the simulation or experiment parameters are set. In the next step, dimensionless learning aims to recover the expression of the varying coefficients by leveraging the dimension of these coefficients. By combining these two steps, the proposed method can efficiently obtain a consistent dimensionally homogeneous governing equation with a small amount of data. In contrast, the standard SINDy falls short of achieving a consistent parameterized differential equation for the same system with different parameters[15,43]. For example, the governing equation for the spring–mass–damper system is $\frac{dx}{dt} = -\frac{k}{c}x - \frac{m}{c}\frac{d^2x}{dt^2}$. If we use different parameters (damping coefficient $c$, spring constant $k$, or mass $m$) in this system, SINDy can only provide scalar coefficients for $x$ and $\frac{d^2x}{dt^2}$ rather than the expressions $-\frac{k}{c}$ and $-\frac{m}{c}$, respectively. Other advanced SINDy approaches deal with this issue by multiplying the candidate terms by a set of predetermined parameters[15,44]. Although these approaches can address this inconsistent governing equation problem, it couples the optimization of identifying candidate terms and parameterized coefficients, making the optimization more difficult. If there are many combinations of parametric derivative or non-derivative terms, this problem can become more difficult and unmanageable.

In order to determine the sensitivity and sensibility of the proposed method, we studied three major factors affecting the discovery results. The first factor is the noisy data effect. we demonstrated the proposed algorithm by solving three challenging problems with noisy experimental measurements, which are described in detail in the sections "Turbulent Rayleigh–Bénard convection", "Vapor depression dynamics in laser-metal interaction" and "Porosity formation in 3D printing of metals". It is found that in these three problems, even with the noisy data effect, the method achieves high fitting performance in both training and test sets (all $R^2$ scores are >0.95). The second factor is the scarce data effect. Most machine learning algorithms rely on a large amount of data to achieve good generalization and minimum out-of-bag error. However, because of the complexity and cost of experiments, it is not always feasible to obtain a big dataset for engineering problems. To deal with scarce data and obtain a universal model, the proposed method embeds dimensional invariance with input variables and successfully reduces the solution space to a manageable size. The dimensional invariance can be regarded as a physical regularization and changes the model structure, which enables the proposed method to train a universal model with limited data points. For example, even though we only used 182, 90, and 92 experimental measurements, respectively, in three complex engineering examples from the sections "Turbulent Rayleigh–Bénard convection", "Vapor depression dynamics in laser–metal interaction" and "Porosity formation in 3D printing of metals", the identified scaling laws fit very well in all these cases. We also compare the proposed method with popular machine learning algorithms, which have poor generalization in the test set, as described in Section 6.3 of the SI. The third factor is the involved variables. We demonstrate that missing necessary variables or involving redundant variables has no effect on the discovered scaling laws in Sections 6.1 and 6.2 of the SI. Sensitive analysis can also be found in Section 6.2 of the SI.

For the discovery of governing equations, as the second part of the method, the accuracy of the discovered equation can be influenced by data noise and the setting of the sparse regression library. The noisy data analyses are performed on five differential equations, including the Navier–Stokes equation (0.5% Gaussian noise), Euler equation (1%), vorticity equation (1%), and the governing equations for spring–mass–damper systems (4%) and dynamic loading beam systems (2%). Even with the noisy data effect, the method successfully discovers the correct governing equations, as demonstrated in the section "Vorticity form of dimensionless Navier–Stokes equation" of the main manuscript and Section 7 of the SI. A summary of the demonstrated case studies including data type, noise level, and approach can be found in Section 2 of the SI. The tolerable noise level can be further increased by combining the proposed method with some newly developed approaches which apply physics-informed neural networks and/or deep learning approaches to reduce noise and obtain robust derivatives[43,45,46]. To study the sparse regression library effect, we build a general sparse regression library to achieve more generalizable results. Specifically, we use 29 terms in the vorticity equation case, as described in Section 7 of the SI. In general, adding candidate terms to the library relies heavily on the researchers' experience and understanding of the problem. Yet, we provide a guideline for choosing the regression library given in Section 7.5 of the SI.

In summary, the proposed dimensionless learning enables systematic and automatic learning of scale-free low-dimensional laws from a high-dimension parameter space, including many experimental conditions with different parameter settings. It can be applied to a wide range of physical, chemical, and biological systems to discover new dimensionless numbers or modify existing ones. Furthermore, it can be combined with other data-driven methods, such as SINDy, to discover dimensionless differential equations from high-resolution measurements. In material science, the identified compact mathematical expressions provide simple transition rules that translate optimal process parameters from one material (or existing materials) to another (or new materials). Dimensionless learning can reduce complex, highly multivariate problem spaces into descriptions involving only a few dimensionless parameters with clear physical meanings.

This approach is particularly useful for engineering problems involving many adjustable parameters with various dimensions or units, such as advanced materials processing and manufacturing[47], microfluidic flow control for precise drug delivery, and solar energy systems design[48].

## Methods

This work has two main workflows for discovering scaling laws and differential equations, as well as the corresponding dimensionless numbers. These two workflows are built on integrating dimensional invariance into the proposed two-level optimization schemes and sparsity-promoting techniques such as SINDy, respectively. Section 1 of the SI shows the general theory of the first workflow, including the problems statement, the algorithm flowchart, how to construct and determine the number of dimensionless numbers, and more. Section 4 of the SI contains a detailed description of the proposed two-level optimization scheme, including the training procedure, pseudo-code, optimization results, a summary of hyperparameter settings, and more. For the second workflow, a detailed description of the proposed symmetric invariant SINDy and the integration of dimensionless learning with SINDy can be found in Section 7.1 of the SI.

## Data availability

All datasets used in this study are available on GitHub at https://github.com/xiaoyuxie-vico/PyDimension.

## Code availability

All source codes used in this manuscript are available on GitHub at https://github.com/xiaoyuxie-vico/PyDimension(https://doi.org/10.5281/zenodo.7317017).

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

## Acknowledgements

This study was supported by the National Science Foundation (NSF) through grants CMMI-1934367. Discussions with Prof. Gregory J Wagner at Northwestern University on the methodology and fluid dynamics simulations are gratefully acknowledged.

## Author contributions

Z.G. proposed the original ideas, supervised the project, and wrote the manuscript and SI. X.X. developed the method, performed the research, wrote codes, and wrote the manuscript and SI. A.S. conducted the fluid mechanics problems, helped with the solid mechanics problem, and wrote the manuscript and SI. J.G. did the solid mechanics problem and wrote the SI. W.K.L. contributed to numerous discussions and advice, and the supervision of the project. All the authors reviewed and edited the manuscript.

## Competing interests
The authors declare no competing interests.
