## [Peer Review File · Nature Communications]

REVIEWER COMMENTS

Reviewer #1 (Remarks to the Author):

**Review paper
Nature Communications**

Title : Data- Driven discovery of dimensionless number and scaling laws from experimental measurements

The paper proposes a new methodology to reduce system describe with high dimensional parameter into description involving few physically- interpretable dimensionless parameters. Th authors for demonstrate the validity of methodology proposed applied it at some well knows physical examples.

Although the method proposed seem to be more interesting to dispose of mechanism of data driven based on two level of machine learning able to automatically discover dominant dimensionless numbers and scaling laws from experimental data

Is not completely clear what is the sensitivity and sensibility of the proposed methodology? How does the output change if the measurements are very noisy? How is the optimal model selected? The examples proposed are certainly illuminating but examples known except the application to 3D printing. How does the methodology work with physically more complex and less common examples such as those of ref 11 and 12?

The suggestion is that the paper cannot be published in this form before the above doubts are clarified

Reviewer #2 (Remarks to the Author):

The manuscript "DATA-DRIVEN DISCOVERY OF DIMENSIONLESS NUMBERS AND SCALING LAWS FROM EXPERIMENTAL MEASUREMENTS" presents a novel method for the discovery of scientific laws expressed by means of non-dimensional numbers. In recent times, there has been a growing interest in the development of machine learning techniques able to distill scientific laws from data, and this manuscript can be seen as a product of this interest.

The authors develop first a method for the determination of possible non-dimensional candidates (Eqs. (3)-(5)). This procedure follows closely ref. [21] of the manuscript, so there is no novelty on it.

Once non-dimensional candidate numbers are determined, the authors develop the method able to unveil their significance in expressing the data. To this end, they develop first a method based on the assumption that one of the relevant non-dimensional numbers is a polynomial combination of the other, Eq. (9). In this case, they choose, rather arbitrarily, a fifth-order polynomial. The rest of the paper is devoted to provide the reader with details on how to determine the coefficients of this polynomial expansion and to give some examples on its performance.

Up to this moment, the relevance of the paper is rather low: they employ a known method for the determination of non-dimensional numbers relevant for the physical phenomenon at

hand, and then assume a polynomial dependence of the result on them.

However, at the conclusions section, they choose to introduce one ingredient that makes the method much more interesting to me: they employ the SINDy technique by Kutz and Brunton, based on the just found non-dimensional numbers. The resulting technique is now much more interesting and intellectually appealing, but the price paid for this is the (almost) lack of novelty of the proposal. The resulting method is nothing more than a particular case of the SINDy technique introduced in ref. [15].

In sum, the paper is correct and well written, but in my humble opinion, it lacks of significance. The only novelty is the application of the SINDy technique on top of non-dimensional numbers found through the employ of a known algorithm.

Reviewer #3 (Remarks to the Author):

The paper presents three strategies to learn dimensionless numbers and scaling laws for complex systems, respectively relying on grid search, gradient descent and pattern search. The proposed approaches are shown to result in physically interpretable dimensionless parameters and dynamic models, providing a compact yet informative mean to understand the behavior of complex systems. The effectiveness of the approaches in learning dimensionless parameters is shown over two case studies, respectively devoted to characterize turbulent Rayleigh-Bérnard convection and the porosity formation in 3D printed materials. On the other hand, the approach is applied to a linear spring-damper-mass system to highlight its benefits in retrieving dimensionless governing equations for dynamical systems.

Overall, the contribution seems novel, the proposed approaches sound and most of the presented case study suitably support the claims of the authors. Although I believe that the contribution can have a positive impact, especially for what concerns the estimation of dimensionless parameters, I have some concerns on how the proposed strategies compare to alternative approaches and I have some perplexities on some features of the presented techniques. As such, I think the paper should be revised to be accepted.

First of all, I was wondering how the methods compare with suitably constructed Neural Networks (e.g., the one proposed by some of the authors in "Hierarchical Deep Learning Neural Network (HiDeNN): An artificial intelligence (AI) framework for computational science and engineering"). I acknowledge that Neural Networks might be less interpretable than the proposed models, but I feel that this consideration is likely to hold only when simple parameterisations of the relationship between the input and output dimensionless parameters are considered. I thus think a comparison would be important to highlight the benefit of the introduced approach.

Secondly, I believe that the case study considered to show the capability of the method to reconstruct the governing equations is excessively simple. Indeed, for the considered system one can use standard machine learning (and system identification) techniques to learn its model from data with little effort. Therefore, why would one prefer your approaches to fairly standard (and easier) ones? The choice of the proposed methods over existing ones should be motivated and a comparison highlighting its benefits over state-of-the-art techniques should be presented to support the conclusions drawn by the authors. Alternatively, I suggest them to consider a different (and possibly more complex) system to assess the performance of the approach presented in the work.

As it concerns the proposed approaches, the grid search one used to attain the results proposed in the main manuscript is never formalized. This makes it difficult to grasp the

details of the latter and, thus, understand its benefits and flows. In addition, if the proposed method is a naive grid search, the approach seems to me a rather simplistic one to detect the coefficients of the basis combinations, possibly leading to poor results when the mesh of the grid over which the search is performed is not dense enough. I was thus wondering if the authors can comment on this aspect, possibly providing a detailed explanation of their design choices in the supplementary material.

Moreover, several thresholds and parameters (e.g., the gradient descent step) have to be selected at different stages of the procedure. Nonetheless, their choice is never explained, insights on how to select these parameters are chosen are never provided nor a sensitivity analysis is performed to explain how the latter should be selected. I believe this severely undermines the possibility to exploit the approach in other applications. I thus suggest the authors to at least provide some guidelines on their choices. I would also like to point out that, to the best I could detect, several threshold and parameters needed within the different routines are never mentioned, thus making it impossible to replicate the results presented in the paper. Similar consideration holds for the exploration spaces and the density of the grid reported in Algorithms 2 and 3 (see the Supplementary material), whose generality is somewhat obscure.

Still concerning the grid search-based approach, I was wondering if the authors can comment on how it scales with respect to the number of coefficients to be learned and compared with the other two strategies they propose. Indeed, I expect that the grid search approach becomes rather computationally demanding when the dimensions increase. Since three possible techniques are presented, I think it would be important to better highlight their differences and advantages/disadvantages over the others.

Additionally, it is known that alternated approaches like the gradient based and pattern search based ones presented in the Supplementary material can get trapped in local minima and that the quality of the resulting solution is heavily dependent on the chosen initial condition. Are the authors testing several initial conditions and then picking the best result or are they using any other strategy to cope with this phenomenon? If this is not the case, can the authors comment on this?

In the following please find some additional comments on both the manuscript and the supplementary material. All the references and the equation numbering corresponds to those in the paper.

On the manuscript:

- I believe that the limitations of the approach proposed in [14] should be better highlighted and its performance should be eventually compared with the one of the proposed approaches at least for one of the considered case study, to support the validity of the approach proposed by the authors.
- I was wondering what is the reason why you fix γ_1 to 0.5 (see line 199). Can you explain this choice and eventually link it to the computational complexity of the grid search method, if connected to you decision of fixing one parameter beforehand?
- Can you please clarify how you shift from the values of γ_2 and γ_3 presented at line 202 and the ones reported in line 206? Are you considering a different point of the grid, increasing its density or changing some parameter of the grid search algorithm?
- I would like the authors to clarify their comment at line 285. Indeed, how can one be sure to have considered all possible candidates for the dominant dynamics?
- In the discussion, the authors mention several Neural Networks where some sort of invariance is enforced in the network structure. I do understand that these are introduced to highlight the importance of embedding invariance within learning but, at the same time, I do not truly see the relevance of this in the discussion. I thus suggest the authors to at least rethink the location of these comments.
- Can you please further comment on the applications you have in mind that can benefit

from your approach for what concerns low control of precise drug delivery and financial market analysis? Alternatively, can you cite some sources where they highlight some requirements for these applications that are aligned with the features of the proposed method?

On the supplementary material:

- I advise the authors to introduce the NEP index (used in the Manuscript) when discussing the results on porosity formation in the Supplementary Material.
- In Algorithm 1, I believe there is an exceeding squared bracket in $\text{abs}(\min([W$
- Can you please provide additional details on the degree of exploration of the parameter space you need to obtain satisfactory results with your approaches? Do you need specific excitation condition/data informativity conditions to be fulfilled for the approach to properly work? How does the latter compare with the one required by other approaches?
- I was wondering whether the procedure reported in the flow chart in Figure 1 is performed in cross-validation or by using the training and tests sets only. If the latter is used in this second configuration, what are the chances to overfit the available training and validation data? Can the authors discussed this?
- I think the introduction of a mathematical formulation for the problem solved in Section 1.3 would clarify it and highlight the parameters one needs to select at this stage, considering that a lasso regularization seems to be exploited.
- In general, how do you decide the shape of the function f characterising the relationship between the input and output dimensionless parameters? Is it selected based on some priors on the physics of the considered system, by cross-validation or with other methods? By clarifying this, the approach would be easier to apply over problems other than the ones presented in the paper.
- Please check the Supplementary Material as in several occasions the coefficients γ are wrongly indicated as λ (see e.g., point 1. of the list at the beginning of Section 1.4).
- There is a typo in Section 3.1: "show gradient the" should be "show the gradient descent"
- I was wondering whether the Algorithm 2 and 3 can be generalised for a generic parameterizations of f , rather than focusing on the polynomial one.
- In Algorithm 3 you consider 3^{K+1} points nearby R_{center}^{2+} where to compute the R^{2+} . Can you please provide some insights that guided your design choice (eventually formal ones)?
- Given the relation identified from data, in Section 4.1 you then simplify it removing the bias, claiming that the latter is negligible. Can you please show what is the actual impact of such reduction when validating your results?
- Can you provide a physical interpretation to the increase in the R^{2+} you highlight with respect to the results reported in Table 1?
- The results obtained by including the additional temperature difference shown in Section 5.2 highlight that both the first and the second options attain similar performance in testing, with the one accounting for the input you claim to be negligible resulting in a slightly better fit in training. Given this result, why would one pick the second option rather than the first? Has this choice to be guided by the relatively simpler dimensionless parameters obtained in the second case? Although I think that this is the reason driving the conclusions of the authors, I think it would be clearer if they highlight this.
- The axis labels in figure 10 are misplaced.
- By looking at the dataset considered used in the example with the spring-mass-damper system the data seem noiseless. Can the author explain this, possibly introducing noise in case the data they used are noiseless? Moreover, in one of the datasets they use, the displacement is dangerously close to zero for most of the time. I believe that this dataset is not that informative (thus not be helpful in retrieving an accurate result). Can the authors comment on this?
- Are the libraries and parameterisations in (34)-(35) driven by insights on the features of the true dynamics of the system? Please highlight your parameterization choices.

Rebuttal letter

Table of Contents

Reviewer #1 (Remarks to the Author)	1
Reviewer #2 (Remarks to the Author)	15
Reviewer #3 (Remarks to the Author)	30
On the manuscript	44
On the supplementary material	48

Reviewer #1 (Remarks to the Author)

The paper proposes a new methodology to reduce system describe with high dimensional parameter into description involving few physically- interpretable dimensionless parameters. The authors for demonstrate the validity of methodology proposed applied it at some well knows physical examples.

We appreciate the respected reviewer’s instructive comments and suggestions. Inspired by the reviewer’s comments/suggestions, we revised the manuscript and SI to resolve uncleared statements and added sections in response to the questions about sensitivity and noisy data. We believe that the revised manuscript and SI are clearer and informative.

Although the method proposed seem to be more interesting to dispose of mechanism of data driven based on two level of machine learning able to automatically discover dominant dimensionless numbers and scaling laws from experimental data.

Is not completely clear what is the sensitivity and sensibility of the proposed methodology? How does the output change if the measurements are very noisy?

To address these questions, we analyzed the sensitivity and sensibility of the two main parts of discovery included in the proposed methodology, which are the scaling law and governing equation discoveries. Accordingly, we have added plots and descriptions to the main text and supplementary document, explained below.

The method aims to discover scaling laws and dimensionless numbers as well as identify governing equations from scarce data. For discovering scaling laws and dimensionless numbers, as the first part of the method, the associated results can be affected by data noise, the number of data points, and the variables involved.

- 1) Noisy data effect. We demonstrated the proposed algorithm by solving three challenging engineering problems with noisy experimental measurements (not synthetic data) collected from the literature, which is described in detail in Sections 2.1, 2.2, and 2.3 of the main manuscript. It is found that in these three problems, even with the noisy data effect, the proposed method achieves high fitting performance in both training and test sets (all R^2 scores are greater than 0.95).*
- 2) Scarce data effect. Given limited data, it is very challenging to achieve good performance in unseen data. Most machine learning algorithms rely on a large amount of data to achieve good generalization and minimum out-of-bag error. However, because of the complexity and cost of*

experiments, it is not always feasible to gather a big dataset for engineering problems. To deal with scarce noisy data and obtain a universal model, the proposed method incorporates fundamental physical knowledge, the dimensional invariance, as a physical constraint or regularization into data-driven models to perform well on limited and/or noisy data. The embedded physical invariance reduces the learning space and eliminates the strong dependence between variables. This method is a physics-based dimension reduction approach that represents features as dimensionless numbers and transforms data points into a low-dimensional pattern that is unaffected by units and scales. Therefore, the proposed method is capable of training a universal model with limited noisy data points. For example, even though we only used 182, 90, and 92 experimental measurements, respectively, in three complex engineering examples from Section 2.1 to 2.3, the fitting performances in all these cases are excellent. In addition, the dataset used in Section 2.2 of the main manuscript includes three kinds of materials: Ti6Al4V, Al6061, and SS316. Instead of shuffling the dataset and then splitting it in Section 2.2, we only use the first two kinds of material in the training set and test the model on the third material to compare the generalization ability. The proposed method achieves the best generalization in the test set, while all other algorithms only achieve a poor generalization. This improvement is due to ensuring geometric, kinematic, and dynamic similarities based on similitude theory within different systems. A generalization description is shown in the first two paragraphs of the discussion section of the main manuscript. Another detailed description of the generalization comparison can be found in Section 5.3 of the SI.

[The first two paragraphs of the discussion section of the main manuscript]

The proposed dimensionless learning is a powerful technique to identify scientific knowledge from data at multiple levels: dimensionless number at the feature level, scaling law at the algebraic equation level, and governing equation at the differential equation level. Unlike purely data-driven approaches that easily suffer from overfitting on small or noisy datasets, this method incorporates fundamental physical knowledge of dimensional invariance and symmetric invariance as physical constraints or regularizations into data-driven models to perform well on limited and/or noisy data. The embedded physical invariance reduces the learning space and eliminates the strong dependence between variables. This method is a physics-based dimension reduction approach that represents features as dimensionless numbers and transforms data points into a low-dimensional pattern that is unaffected by units and scales. Thus, in addition to being applicable to limited and/or noisy data, the presented approach significantly improves the interpretability of representation learning because dimensionless numbers are physically interpretable. Lower dimensions and better interpretability also allow for qualitative and quantitative analysis of the systems of interest. This has been demonstrated in three complex engineering problems in earlier sections.

Another advantage of the embedded dimensional invariance in dimensionless learning is improved generalization capability. To show this, in the vapor depression dynamics case, we compared the performance of dimensionless learning and popular machine learning algorithms on unseen material data points. The proposed method achieves the best generalization in the test set, while all other algorithms only achieve a poor generalization. This improvement is due to ensuring geometric, kinematic, and dynamic similarities based on similitude theory within different systems. A detailed description of the generalization comparison can be found in Section 5.3 of the SI. Aside from dimensional invariance, we also used symmetric invariance in this study. It is a data augmentation technique for SINDy that intrinsically ensures symmetry terms have the same coefficients and effectively reduces the number of learnable regression coefficients in SINDy.

[Section 5.3 of the SI]

5.3 Comparison of different machine learning algorithms with dimensionless learning

The out-of-bag error is a critical metric for machine learning algorithms, especially when dealing with problems with limited data. In this section, we compare the performance of five popular machine

learning algorithms with our proposed method over two small datasets. The proposed method shows the best generalization in datasets with different materials and scales.

5.3.1 Model generalization for different materials

The first dataset is the same as the one used in Section 2.2 of the manuscript. This dataset contains three different materials: titanium alloy (Ti6Al4V), aluminum alloy (Al6061), and stainless steel (SS316). Here, we do not shuffle all the data and split it into training and test sets. We only use Ti6Al4V and Al6061 data in the training set, while other data (SS316) is used in the test set. 5-fold cross-validation is used to select the best parameters of each model.

Fig. 9 shows the R^2 score for each method. Even though the Feed Forward Neural Network (FFNN) achieves very good performance on the training and validation sets, the R^2 score for the test set is below -0.1. Random Forest (RF) has relatively high R^2 scores for training and validation, but it still performs poorly on the test set. On the other hand, the proposed method has the highest R^2 score on the test set, indicating that it has the best generalization in different materials. This improvement is the result of ensuring geometric, kinematic, and dynamic similarities based on similitude theory within different systems.

Figure 10: Comparisons of six candidate models for the generalization of different materials. The training and validation sets come from two materials (titanium alloy (Ti6Al4V) and aluminum alloy (Al6061)), while the test set comes from the other material (stainless steel (SS316)). The highest the R^2 score, the better the model. The y-axis only shows R^2 scores above -0.1. The regression methods include Feed-forward Neural Network (FFNN), Linear regression (LR), Extreme Gradient Boosting (XGBoost), K-Nearest Neighbors (KNN), Random Forest (RF), and the proposed method.

5.3.2 Model generalization for different scales

In this section, we use a rough pipe flow example to demonstrate that the proposed method has the best generalization ability in data with different scales. Pipe flow is a classic example to analyze the fluid flow from laminar to turbulent. Reynolds number (Re) is widely used in fluid mechanics to distinguish between laminar and turbulent flow. In general, several measurements can be collected in the steady state of a pipe flow system: fluid velocity v , dynamic viscosity μ , pipe diameter d , fluid density ρ , and pressure p .

Two types of synthetic data are created to build the dataset: small pipe flow data for the training and validation sets, and large pipe flow data for the test set. All the data are generated randomly. The detailed range of each parameter is shown in Table 5

Table 5: **Different parameter ranges for small and large pipe flows.**

Source	Parameter	Range	Source	Parameter	Range
Small pipe	d	0.1 - 1 cm	Large pipe	d	10 - 100 cm
	μ	$1 \times 10^{-3} - 1 \times 10^{-2}$ g/cm/s		μ	$1 \times 10^{-5} - 1 \times 10^{-3}$ g/cm/s
	v	100 - 1×10^3 cm/s		v	1 - 10 cm/s
	ρ	$1 \times 10^{-3} - 1 \times 10^{-2}$ g/cm ³		ρ	$1 \times 10^{-4} - 1 \times 10^{-3}$ g/cm ³
	p	$1 \times 10^4 - 1 \times 10^5$ Pa		p	$1 \times 10^4 - 1 \times 10^5$ Pa

Fig. 11 compares the performance of six regression models on the training, validation, and test sets. Because the test set consists entirely of large pipe flow rather than small pipe flow, all machine learning algorithms perform very poorly on the test set (below 0). On the other hand, the proposed method can perform very well on the test set. Even though this dataset is synthetic, it shows that the proposed method can be used for data with different scales.

Figure 11: Comparisons of six candidate models to identify a generalized model for data from different scales. The training and validation sets are generated using small pipe cases, while the test set is associated with large pipe cases. The higher the R^2 score, the more accurate the model. The y-axis only shows R^2 scores greater than -0.1.

- 3) *Involved variables effect.* We demonstrated that missing necessary variables or involving more variables has no effect on the discovered scaling laws in Sections 5.1 and 5.2 of the Supplementary Information (SI).

For the discovery of governing equations, as the second part of the method, the accuracy of the discovered equation can be influenced by data noise and the setting of the sparse regression library.

- 1) *Noisy data effect.* We ran the analysis on four differential equations, including Navier-Stokes (NS) equation, Euler equation, vorticity equation, the governing equation for spring-mass-damper systems and dynamic loading beam systems, with and without noise. The noise levels considered for the abovementioned equations are 0.5% (NS), 1% (Euler), 1% (Vorticity), 4% (spring-mass-damper), 2% (dynamic loading) respectively. Even with noisy data effects, the proposed method can discover the correct governing equations. The detailed description can be found in Section 2.4 and the discussion section of the main manuscript and Section 6 of the SI.
- 2) *Sparse regression library effect.* We built a general sparse regression library to achieve more generalizable results. Specifically, we used 29 terms in the vorticity equation case and 17 terms in the NS equation case. The detailed description can be found in Section 6 of the SI. We also give

detailed suggestions for readers to choose their own regression library, which is described in Section 6.5 of the SI. Furthermore, we made the code and data available on GitHub (<https://github.com/xiaoyuxie-vico/PyDimension>) so that readers could replicate this work.

[Section 2.4 of the main manuscript]

Section 2.4 Vorticity form of dimensionless Navier-Stokes equation

In this section, we demonstrate dimensionless learning in discovering the governing equation using the Kármán vortex street problem. The goal is to explicitly identify the well-known dimensionless vorticity equation as well as the Reynolds number (Re) *only* from simulation snapshots. The Re is the ratio of inertial forces to viscous forces, which is frequently used to distinguish between the laminar and turbulent flow.

Fig. 4 shows a schematic of integrating dimensionless learning with a proposed physically enhanced SINDy, namely, symmetric invariant SINDy, to identify the dimensionless governing equations as well as dimensionless number(s). Here, we demonstrate the proposed method for a fluid flow problem involving three cylinders with diameter l , fluid density ρ , dynamic viscosity μ , inlet velocity V , and the pressure difference between the upstream and downstream p_0 . Three CFD simulations for different flow regimes are carried out to generate a dataset for the partial differential equation (PDE) discovery. The velocity and vorticity measurements are sampled sparsely (Fig. 4a).

Figure 4. Integration of dimensionless learning with symmetric invariant SINDy for identifying the Navier-Stokes equation with Reynolds number. (a) Original data are generated from parametric simulations. Because of symmetric invariance, another set of transformed data is obtained by flipping the original data along $y = x$. (b) The original and transformed data are concatenated together for symmetric invariant SINDy, which incorporates symmetric invariance into SINDy to ensure that symmetric invariant terms have the same coefficients. (c) The identified temporary governing equations for each simulation case. Some of the coefficients are close to constant, while others vary depending on the simulation case. All the other candidate terms have zero coefficients. More importantly, symmetry terms (e.g., $u \frac{\partial \omega}{\partial x}$ and $v \frac{\partial \omega}{\partial y}$) have identical coefficients. (d) Dimensionless learning is applied to recover the explicit expression for the varying coefficients. The parametric space to be explored includes five parameters. By incorporating dimensional invariance, we need to optimize basis coefficients $\boldsymbol{\gamma}$ and fitting coefficient β . (e) Substituting the discovered regression coefficients ($1/\text{Re}$) into the temporary governing equation. In this step, a consistent dimensionally homogeneous governing equation, which is identical to the Navier-Stokes equation in a vorticity form, is obtained.

With an assumption that the governing equation is invariant to the symmetric transformation along $y = x$, we flip the original data along $y = x$ for each simulation case to obtain the transformed data. The original and transformed data are then sampled with 4,000 measurements each on the same relative position. Based on these measurements, two regression libraries are built to identify the governing equation using linear and quadratic terms for $u, v, \omega, \frac{\partial \omega}{\partial x}, \frac{\partial^2 \omega}{\partial x^2}, \frac{\partial^2 \omega}{\partial y^2}$. In total, there are 29 terms in the regression library. Unlike SINDy, which uses the library from the original data directly, the proposed symmetric invariant SINDy trains the measurements from the original and transformed data together to implicitly ensure the symmetry terms have the same coefficients. That is, the coefficients for symmetry terms are physically constrained. The sum of symmetry terms, such as $\frac{\partial \omega}{\partial x}$ and $v \frac{\partial \omega}{\partial y}$, are invariant to symmetry transformation along $y = x$. The identified temporary governing equation with only four non-zero regression coefficients is shown in Fig. 4c. ξ_{12} and ξ_{19} are identical and close to constants, while ξ_6 and ξ_7 are also the same but vary with parameters such as ρ, μ , and so on. The proposed symmetric invariant SINDy is described in detail in Section 6.1.2 of the SI.

Examining Fig. 4c, we notice that the identified temporary governing equations vary from case to case. To identify a consistent governing equation, we apply dimensionless learning to find the expression for varying regression coefficients. The parametric space to be explored for $\xi_6 = \xi_7$ can be expressed as follows:

$$\xi_6 = \xi_7 = f(\mu, \rho, V, l, p_0) \quad (17)$$

In contrast to standard dimensionless learning, we simplify the representation function $f(\cdot)$ as a power law with a constant coefficient rather than a fifth-order polynomial in Sections 2.1 and 2.2 or an XGBoost in Section 2.3. Using pattern search-based optimization, the identified expression for ξ_6 and ξ_7 is $1.083 \frac{\mu}{\rho V l} \approx \frac{\mu}{\rho V l}$, which is the reciprocal of the well-known Reynolds number $\text{Re} = \frac{\mu}{\rho V l}$. Section 6.1.3 of the SI contains detailed information on dimensionless learning. By substituting the constant regression coefficients and this discovered expression into the temporary governing equations, we obtain a consistent dimensionless governing equation as follows:

$$\frac{\partial \omega}{\partial t} = -u \frac{\partial \omega}{\partial x} - v \frac{\partial \omega}{\partial y} + \frac{1}{\text{Re}} \left(\frac{\partial^2 \omega}{\partial x^2} + \frac{\partial^2 \omega}{\partial y^2} \right) \quad (18)$$

which is identical to the well-known vorticity form of the Navier-Stokes equation. We further apply the proposed method to 1% Gaussian noisy data. The proposed method successfully identifies the correct governing equation as Eqns. 18. The detailed results for noisy data are shown in Section 6.1.3 of the SI. More applications in fluid and solid mechanics and dynamics systems with and without noise are demonstrated in Section 6 of the SI.

[Sections 6.1 and 6.2 of the SI are shown below, Sections 6.3 and 6.4 can be found in the SI.]

6. Governing equation discovery from limited data

In this section, we integrate dimensionless learning with symmetric invariant SINDy or SINDy to identify governing equations as well as dimensionless numbers in fluid and solid mechanics. The identified Partial Differential Equations (PDEs) include the vorticity equation, Navier-Stokes equation, Euler equation, and the governing equation for spring-mass-damper systems.

6.1 Vorticity equation

6.1.1 Data generation and preprocessing

The vorticity equation is a vorticity form of the Navier-Stokes equation that describes fluid flow by using the evolution of vorticity. To generate the original data, we conducted three sets of simulations

for the Kármán vortex street problem with three circular cylinders using Flow3D [10]. The geometry parameters include the diameter of one cylinder l , the length in x direction l_x , and the length in y direction l_y . The operational and material parameters considered for this problem include the inlet flow velocity V , the dynamics viscosity μ , the fluid density ρ , and the pressure difference between upstream and downstream p_0 . Other simulation parameters include the number of mesh cells in x and y directions (N_x and N_y). The detailed information on these parameters is shown in Table 6.

Table 6: Simulation parameters of fluid flow through three circular cylinders to identify vorticity equation.

Parameter Unit	l cm	V cm/s	μ kg/cm/s	ρ g/cm ³	p_0 Pa	Re	l_x cm	l_y cm	N_x 1	N_y 1
Case 1	0.0850	68.00	0.1156	0.0010	1.30×10^4	50	0.9	0.4	500	222
Case 2	0.1000	90.00	0.1100	0.0011	8.42×10^4	90	0.9	0.4	500	222
Case 3	0.1000	100.00	0.1100	0.0011	8.53×10^4	100	0.9	0.4	500	222

Figure 12: Vorticity contour plot from a simulation with $Re=100$. The red color rectangle represents the sampling region.

Table 7: Coordinate transformation using flipping operation.

Original data	t	x	y	u	v	ω
Transformed data	$t^0 = t$	$x^0 = y$	$y^0 = x$	$u^0 = v$	$v^0 = u$	$\omega^0 = -\omega$

To identify the governing equation from data, 100 time-step snapshots are prepared and sparsely sampled. Fig. 12 shows a typical snapshot and the sampling region. In the sampling region, 4,000 data points are randomly sampled. For each data point, there are six variables: a specific time step (t), two spatial Cartesian coordinates (x and y), and three time-varying measurements (velocity in x direction u , velocity in y direction v , and vorticity ω). These six variables are non-dimensionalized by the following characteristic variables: characteristic length l (the cylinder diameter), characteristic velocity V (inlet flow velocity), characteristic time $t_{ref} = \frac{l}{V}$, and characteristic vorticity $\omega_{ref} = \frac{l}{V}$.

For each simulation case, we flip the original data along $y = x$ to obtain symmetrically transformed data. The original and transformed variables are shown in Table 7.

6.1.2 Symmetric invariant SINDy

After augmenting the dataset, we conduct symmetric invariant SINDy for each case in Table 6 to identify the corresponding temporary governing equations. Fig. 13 shows a schematic of concatenating the sparse regression terms from the original and transformed data.

There are several implicit benefits for sparse regression because of this concatenation. First, terms with the same absolute value in the original and transformed data must have the same coefficients, indicating that the regression coefficients are physically constrained. For example, the coefficients for $\frac{\partial \omega}{\partial x}$ (the fourth column) and $\frac{\partial \omega}{\partial y}$ (the fifth column) must be the same ($\xi_5 = \xi_6$), and the coefficients for u (the first column) and v (the second column) must be zero ($\xi_1 = \xi_2=0$). Second, the learnable coefficients can be reduced because of the physical constraint, which reduces the solution space and the required number of training data accordingly. Third, data augmentation is one of the important regularization techniques to avoid overfitting [11]. By leveraging symmetric invariance, the training data is doubled, which significantly improves the learning accuracy for the regression coefficients.

In this case, we include 29 general terms in the library as the candidate terms:

$$\begin{aligned} \frac{\partial \omega}{\partial t} = & [u, v, \omega, \frac{\partial \omega}{\partial x}, \frac{\partial \omega}{\partial y}, \frac{\partial^2 \omega}{\partial x^2}, \frac{\partial^2 \omega}{\partial y^2}, \frac{\partial^2 \omega}{\partial x \partial y}, uu, uv, u\omega, u \frac{\partial \omega}{\partial x}, u \frac{\partial \omega}{\partial y}, u \frac{\partial^2 \omega}{\partial x^2}, u \frac{\partial^2 \omega}{\partial y^2}, u \frac{\partial^2 \omega}{\partial x \partial y}, \\ & vv, v\omega, v \frac{\partial \omega}{\partial x}, v \frac{\partial \omega}{\partial y}, v \frac{\partial^2 \omega}{\partial x^2}, v \frac{\partial^2 \omega}{\partial y^2}, v \frac{\partial^2 \omega}{\partial x \partial y}, \omega\omega, \omega \frac{\partial \omega}{\partial x}, \omega \frac{\partial \omega}{\partial y}, \omega \frac{\partial^2 \omega}{\partial x^2}, \omega \frac{\partial^2 \omega}{\partial y^2}, \omega \frac{\partial^2 \omega}{\partial x \partial y}] \xi, \end{aligned} \quad (27)$$

where the dimension of ξ is 29×1 and the derivative terms are calculated by a local polynomial interpolation approach [12, 13]. A second-order polynomial is fitted to each data point using eight neighboring points, and the derivatives can be calculated by the fitted polynomial.

Figure 13: Concatenating the regression library for symmetric invariant SINDy from the original and transformed data. Columns with the same color have the same absolute value.

Table 8: Dataset including computed non-zero regression coefficients and parameters for the vorticity simulation.

Parameter Unit	l cm	V cm/s	μ kg/cm/s	ρ g/cm ³	p_0 Pa	Re 1	$\xi_{12} = \xi_{19}$ 1	$\xi_6 = \xi_7$ 1
Case 1	0.0850	68.00	0.1156	0.0010	1.30×10^3	50	-0.9925	0.0212
Case 2	0.1000	90.00	0.1100	0.0011	8.42×10^4	90	-0.9909	0.0126
Case 3	0.1000	100.00	0.1100	0.0011	8.53×10^4	100	-0.9941	0.01101

To find sparse regression coefficients ξ , we use a sequential threshold least-squares algorithm [12]. After optimization, each case will have only four non-zero regression coefficients. Thus, we are able to construct a non-zero coefficient dataset in combination with the parameters (and their values) listed in Table 6 for the three different cases, as shown in Table 8.

From Table 8, we notice that $\xi_{12} = \xi_{19}$ is very close to a constant value (-1), while $\xi_6 = \xi_7$ varies with different cases. Therefore, the temporary governing equation can be simplified as:

$$\frac{\partial \omega}{\partial t} = -u \frac{\partial \omega}{\partial x} - v \frac{\partial \omega}{\partial y} + \xi_6 \left(\frac{\partial^2 \omega}{\partial x^2} + \frac{\partial^2 \omega}{\partial y^2} \right) \quad (28)$$

6.1.3 Dimensionless learning

To obtain a consistent governing equation, we apply dimensionless learning to identify the expression of ξ_6 and ξ_7 . The causal relationship can be expressed as:

$$\xi_6 = \xi_7 = f(\mu, l, V, \rho, p_0). \quad (29)$$

The dimension matrix \mathbf{D} for Eqns. 29 is:

$$\begin{bmatrix} -1 & 1 & 1 & -3 & -1 \\ -1 & 0 & -1 & 0 & -2 \\ 1 & 0 & 0 & 1 & 1 \end{bmatrix} \quad (30)$$

The columns of the matrix, from right to left, represent the dimensions of μ, l, V, ρ , and p_0 respectively. The rows, from top to bottom, represent the dimensions of length, time, and mass respectively.

Unlike Sections 2.1 and 2.2 of the main manuscript, where a polynomial is used to fit the relationship between dimensionless numbers, we can simplify the fitting function $f(\cdot)$ in Eqns. 29 as a power law with a constant coefficient β . Then, the Eqns. 29 can be simplified as:

$$\xi_6 = \beta \mu^{w_1} l^{w_2} V^{w_3} \rho^{w_4} p_0^{w_5}, \quad (31)$$

where $w = [w_1, w_2, w_3, w_4, w_5]^T$ denotes the powers that generate the dimensionless number and are to be determined.

The powers w need to satisfy the following linear system of equations:

$$\mathbf{D}w = 0. \quad (32)$$

Furthermore, the solutions of the linear system (Eqns. 32) can be written as a linear combination of two basis vectors w_{b1} and w_{b2} :

$$w = \gamma_1 w_{b1} + \gamma_2 w_{b2}, \quad (33)$$

where $w_{b1} = [-1, 1, 1, 1, 0]^T$ and $w_{b2} = [-1, 1, -1, 0, 1]^T$.

To summarize, there are three unknown parameters in the identification of ξ_6 and ξ_7 : two basis coefficients (γ_1 and γ_2) and a constant coefficient β . The training data come from the non-zero regression coefficient dataset in Table 8. Applying a pattern search-based optimization method with a grid search range of $[-2, 2]$ and a grid interval of 0.5, the optimized expressions for ξ_6 and ξ_7 can be written as:

$$\xi_6 = \xi_7 = 1.083 \frac{\mu}{\rho \nu l} \approx \frac{\mu}{\rho \nu l}. \quad (34)$$

This form is similar to the reciprocal of the classical Reynolds number, indicating that the integration of dimensionless learning can directly identify the expression of sparse regression coefficients. By substituting the expression for ξ_6 and ξ_7 (Eqns. 34) into Eqns. 28, we obtain a consistent dimensionless governing equation for all cases:

$$\frac{\partial \omega}{\partial t} = -u \frac{\partial \omega}{\partial x} - v \frac{\partial \omega}{\partial y} + \frac{1}{\text{Re}} \left(\frac{\partial^2 \omega}{\partial x^2} + \frac{\partial^2 \omega}{\partial y^2} \right), \quad (35)$$

which is identical to the well-known Navier-Stokes equation in a vorticity form.

Furthermore, we test the noisy data effect on this problem. After adding 1% Gaussian noise to the measurements, the identified regression coefficients are: (1) case 1, $\xi_{12} = \xi_{19} = -0.9994$, $\xi_6 = \xi_7 = 0.0206$; (2) case 2, $\xi_{12} = \xi_{19} = -0.9909$, $\xi_6 = \xi_7 = 0.0126$; and (3) case 3, $\xi_{12} = \xi_{19} = -0.09941$, $\xi_6 = \xi_7 = 0.0111$. Applying dimensionless learning, both ξ_6 and ξ_7 are identified as $1.0581 \frac{1}{\text{Re}}$, which is close to the reciprocal of the Reynolds number. Therefore, even with the noisy data effect, the proposed method can identify the governing equation.

6.2 Navier-Stokes equation

In this section, we demonstrate that the dimensionless learning in conjunction with symmetric invariant SINDy can be used to identify the Navier-Stokes equation in a Cartesian coordinate system. We performed three series of simulations for the Kármán vortex street problem with three circular cylinders to prepare the original data using Flow3D. The simulation parameters with their values for different cases are shown in Table 10. The time-varying measurements include a specific time step (t), two spatial Cartesian coordinates (x and y), and three time-varying measurements (fluid velocity in x direction u , velocity in y direction v , and pressure p).

Following the same procedures described in Section 6.1, the regression model is defined as:

$$\begin{aligned} \frac{\partial u^*}{\partial t} = & [u^*, uu^*, vu^*, \frac{\partial u^*}{\partial x}, \frac{\partial u^*}{\partial y}, \frac{\partial^2 u^*}{\partial x^2}, \frac{\partial^2 u^*}{\partial y^2}, u \frac{\partial u^*}{\partial x}, u \frac{\partial u^*}{\partial y}, u \frac{\partial^2 u^*}{\partial x^2}, u \frac{\partial^2 u^*}{\partial y^2}, \\ & v \frac{\partial u^*}{\partial x}, v \frac{\partial u^*}{\partial y}, v \frac{\partial^2 u^*}{\partial x^2}, v \frac{\partial^2 u^*}{\partial y^2}, p^*, \frac{\partial p^*}{\partial x}] \xi, \end{aligned} \quad (36)$$

Table 9: Simulation parameters for the fluid flow through three circular cylinders to identify the Navier-Stokes equation.

Parameter Unit	l cm	V cm/s	μ kg/cm/s	ρ g/cm ³	p_0 Pa	Re	l_x cm	l_y cm	N_x 1	N_y 1
Case 1	0.1000	100.00	0.1100	0.0011	8.53×10^5	100	0.9	0.4	500	222
Case 2	0.0950	165.43	0.1040	0.0011	8.53×10^5	170	0.9	0.4	500	222
Case 3	0.1050	170.43	0.1020	0.0011	9.13×10^5	200	0.9	0.4	500	222

Table 10: Identified regression coefficients accompanied by several ground truth dimensionless numbers for the Navier-Stokes problem (clean data).

Case ID	$\xi_8 = \xi_{13}$	$\xi_6 = \xi_7$	ξ_{17}	Re	1/Re	Eu
Case 1	-0.9759	0.0105	-14127.23	100	0.0100	14545.45
Case 2	-0.9814	0.0063	-3850.28	170	0.0059	3897.63
Case 3	-0.9820	0.0059	-2977.75	200	0.0050	3019.97

where ξ is a vector with 17 terms, u^* is the combination of u and v , v^* is the combination of v and u , and p^* is the combination of p and p . It is worth mentioning that u^* , v^* , and p^* are designed for the flipping operation so that we can concatenate the regression libraries for the original and transformed data and train them together. This operation helps to incorporate symmetric invariance into the learning scheme. After the transformation, $u^{*'}$, and $v^{*'}$ become v^* and u^* , respectively, while $p^{*'}$ remains the same as p^* .

Following the same procedure as Section 6.1, we obtain the optimized sparse regression coefficients. There are only five terms with non-zero regression coefficients, as shown in Table 10.

Because $\xi_8 = \xi_{13}$ are close to -1, it is reasonable to assign -1 to both ξ_8 and ξ_{13} . Next, we apply dimensionless learning to identify the expression for the other three non-zero coefficients. Using the same pattern search optimization described in Section 6.1, the identified ξ_6 and ξ_7 are $1.0746 \frac{\mu}{\rho V l}$, which can be simplified to $1/\text{Re}$. The ξ_{17} is identified as $-0.9729 \frac{p_0}{\rho V^2}$ and can be simplified as the negative of a well-known dimensionless number, Euler number, Eu. By substituting all non-zero regression coefficients into Eqns. 36, we obtain a consistent governing equation:

$$\frac{\partial u^*}{\partial t} = -u \frac{\partial u^*}{\partial x} - v \frac{\partial u^*}{\partial y} - \text{Eu} \frac{\partial p^*}{\partial x} + \frac{1}{\text{Re}} \left(\frac{\partial^2 u^*}{\partial x^2} + \frac{\partial^2 u^*}{\partial y^2} \right), \quad (37)$$

which is identical to the Navier-Stokes momentum equation written for the combination of x and y directions.

We also test the noisy data effect on this problem by imposing 0.5% Gaussian noise to the measurements. The identified non-zero coefficients are shown in Table 11.

Using dimensionless learning, we can identify the expression for ξ_6 and ξ_7 as $1.0473 \frac{\mu}{\rho V l}$. Similarly, the expression for ξ_{17} is identified as -0.9687Eu . Compared to the clean data results in Table 10, ξ_8

and ξ_{13} have higher errors. By assuming the ξ_8 and ξ_{13} as $-Eu$, we obtain the same equation as Eqns. 37.

Table 11: Identified regression coefficients accompanied by several ground truth dimensionless numbers for the Navier-Stokes problem (noisy data).

Case ID	$\xi_8 = \xi_{13}$	$\xi_6 = \xi_7$	ξ_{17}	Re	1/Re	Eu
Case 1	-0.9668	0.0104	-14139.49	100	0.0100	14545.45
Case 2	-0.9378	0.0063	-3699.04	170	0.0059	3897.63
Case 3	-0.9217	0.0053	-2785.92	200	0.0050	3019.97

Table 12: Identified regression coefficients accompanied by several ground truth dimensionless numbers for the Euler equation example (clean data).

Case ID	$\xi_8 = \xi_{13}$	ξ_{17}	Re	Eu
Case 1	-1.0436	-14127.23	100	14545.45
Case 2	-1.0016	-3896.14	170	3897.63
Case 3	-1.0177	-3004.48	200	3019.97

Table 13: Identified regression coefficients accompanied by several ground truth dimensionless numbers for the Euler equation example (noisy data).

Case ID	$\xi_8 = \xi_{13}$	ξ_{17}	Re	Eu
Case 1	-1.0361	-14015.25	100	14545.45
Case 2	-0.9946	-3873.86	170	3897.63
Case 3	-1.0088	-2953.95	200	3019.97

It is worth mentioning that we choose a different sampling region for the same data in this problem, but all other procedures remain the same as in the previous example. The identified non-zero regression coefficients for clean and noisy data (1% Gaussian noise) are shown in Table 12 and Table 13.

From Table 12 and Table 13, we can see that ξ_8 and ξ_{13} are very close to -1 , and ξ_{17} is the only varying regression coefficient. To identify the expression of ξ_{17} for clean and noisy data, we use dimensionless learning and identify the expressions are $-0.9671Eu$ and $-0.9661Eu$, respectively. By simplifying ξ_{17} to $-Eu$ and substituting all non-zero regression coefficients into Eqns. 36, we obtain the governing equation as below:

$$\frac{\partial u^*}{\partial t} = -u \frac{\partial u}{\partial x} - v \frac{\partial u}{\partial y} - Eu \frac{\partial p^*}{\partial x} \quad (38)$$

which is identical to the Euler equation. The reason for simplifying the Navier-Stokes equation to the Euler equation is that the viscous effects of the fluid far from the cylinders and walls are negligible in this problem.

[Section 6.5 of the SI] 6.5 Regression library suggestions

Building a regression library with all the required candidates relies on the researchers' experience and understanding of the problem. If some terms are known to be important to the problems, more specific terms can be included, limiting the regression library to a smaller one. For example, convective and diffusion terms are frequently used in the governing equations of fluid mechanics. It is reasonable to include these terms in the library.

In addition, if there is some prior knowledge about the relationship between different terms, the number of learnable coefficients can be reduced. For example, for a 2-dimensional fluid mechanics problem, we can assume that the two convective and diffusion terms have the same coefficients. It is worth noting that the proposed symmetric invariance does not explicitly include this physical constraint but achieves it implicitly.

In a more general and unknown problem, we suggest including linear terms first, such as the original measurements and their first and second order derivatives. If the fitting performance is ideal, we can stop involving more terms and get the best differential equations. Otherwise, we will keep adding quadratic and other high-order terms and testing the fitting performance. The library will be updated until the best performance model is discovered.

[More demonstrated problems can be found in Sections 6.3 and 6.4 can be found in the SI, including spring-mass-damper problems and dynamic response of impulsively loaded plastic beam problems.]

[The sixth and seventh paragraphs of the discussion section of the main manuscript]

In order to determine the sensitivity and sensibility of the proposed method, we studied three major factors affecting the discovery results. The first factor is the noisy data effect. we demonstrated the proposed algorithm by solving three challenging problems with noisy experimental measurements, which are described in detail in Sections 2.1, 2.2, and 2.3. It is found that in these three problems, even with the noisy data effect, the method achieves high fitting performance in both training and test sets (all R^2 scores are greater than 0.95). The second factor is the scarce data effect. Most machine learning algorithms rely on a large amount of data to achieve good generalization and minimum out-of-bag error. However, because of the complexity and cost of experiments, it is not always feasible to obtain a big dataset for engineering problems. To deal with scarce data and obtain a universal model, the proposed method embeds dimensional invariance with input variables and successfully reduces the solution space to a manageable size. The dimensional invariance can be regarded as a physical regularization and changes the model structure, which enables the proposed method to train a universal model with limited data points. For example, even though we only used 182, 90, and 92 experimental measurements, respectively, in three complex engineering examples from Section 2.1 to 2.3, the identified scaling laws fit very well in all these cases. We also compare the proposed method with popular machine learning algorithms, which have poor generalization in the test set, as described in Section 5.3 of the SI. The third factor is the involved variables. We demonstrate that missing necessary variables or involving redundant variables has no effect on the discovered scaling laws in Sections 5.1 and 5.2 of the SI.

For the discovery of governing equations, as the second part of the method, the accuracy of the discovered equation can be influenced by data noise and the setting of the sparse regression library. The noisy data analyses are performed on five differential equations, including Navier-Stokes (0.5% noise), Euler (1%), vorticity (1%), the governing equation for spring-mass-damper systems (4%), and dynamic loading beam systems (2%). Even with the noisy data effect, the method successfully discovers the correct governing equations, as demonstrated in Section 2.4 of the main manuscript and Section 6 of the SI. The tolerable noise level can be further increased by combining the proposed method with some newly developed approaches which apply physics-informed neural networks and/or deep learning approaches to reduce noise and obtain robust derivatives [45, 47, 48]. To study the sparse regression library effect, we build a general sparse regression library to achieve more

generalizable results. Specifically, we use 29 terms in the vorticity equation case, as described in Section 6 of the SI. In general, adding candidate terms to the library relies heavily on the researchers' experience and understanding of the problem. Yet, we provide a guideline for choosing the regression library given in Section 6.5 of the SI.

[Added references to the main manuscript]

[45] Chen, Zhao and Liu, Yang and Sun, Hao. Physics-informed learning of governing equations from scarce data. *Nature communications*, 12(1):1–13, 2021.

[47] Zhang, Zhiming and Liu, Yongming. A robust framework for identification of PDEs from noisy data. *Journal of Computational Physics*, 446:110657, 2021.

[48] Rao, Chengping and Ren, Pu and Liu, Yang and Sun, Hao. Discovering Nonlinear PDEs from Scarce Data with Physics-encoded Learning. *arXiv preprint arXiv:2201.12354*, 2022.

How is the optimal model selected?

We select the optimal model based on the fitting performance (R² score) and the L1 norm of dimensionless numbers. These two criteria ensure that the scaling law is parsimonious and fits well with the current dataset. We added this explanation to Section 1.5 of the SI.

[Section 1.4 of the SI] Following the flowchart in Fig. 1, a few candidate dimensionless numbers and scaling laws can be obtained. Then, we select the optimal model based on the fitting performance (R² score) of the scaling law and the L1 norm of dimensionless numbers. We encourage a higher R² score and a lower L1 norm. In a few special situations, when different scaling laws have the same R² score and L1 norm, we choose the model with fewer variables. These criteria ensure that the scaling law is parsimonious and fits well on the dataset.

The examples proposed are certainly illuminating but examples known except the application to 3D printing. How does the methodology work with physically more complex and less common examples such as those of ref 11 and 12?

Inspired by the question, we demonstrated the proposed method in both fluid and solid mechanics for two additional challenging problems (and less common cases in mechanics).

In the case of fluid mechanics, we conducted various CFD simulations to generate datasets to demonstrate the application of the proposed method. For a flow problem involving three cylinders at different flow regimes, the Navier-Stokes and Euler equations with explicit dimensionless numbers were successfully determined. A range of Reynolds numbers (50 to 200) has been considered in this work. Using the proposed approach in this work, we are able to discover dimensionless numbers that are physically meaningful in contrast to coefficient values.

Another improvement provided by this proposed dimensionless learning is that the regression coefficients are consistent across different values of input parameters set in simulations. Many data-driven approaches, including SINDy-incorporated methods [16, 45], fall short of achieving a consistent governing equation for the same system with different parameters. For instance, vanilla SINDy-related approaches can only provide a scalar coefficient for each candidate term in their regression library. Yet, these regression coefficients can vary depending on how the simulation or experiment parameters are set. Other advanced SINDy approaches deal with this issue by multiplying the candidate terms with a set of pre-determined parameters or parameterized expressions. Although this can address this inconsistent governing equation problem, it couples the optimization of identifying candidate terms and parameterized coefficients, making the optimization more difficult [16, 46]. This problem can be more difficult to handle if there are many combinations of parametric derivative or non-derivative terms. To find universal governing equations efficiently, the proposed method separates the identification of a consistent governing equation into two steps. The first step is to find a temporary governing equation in which the regression coefficients can be constants or vary

depending on how the simulation or experiment parameters are set. In the next step, dimensionless learning is designed to recover the expression of the varying coefficients by leveraging the dimension of the varying coefficients. By combining the results of these two steps, we can obtain a consistent dimensionally homogeneous governing equation. The detail of the fluid mechanics case along with the invariance technique is explained in Section 2.4 and the discussion section of the revised manuscript and Section 6 of the SI.

In the case of solid mechanics, we generated datasets for the governing equation discovery by analytically solving equilibrium equations for a supported beam loaded with an initial uniform velocity impulse. Structural elements with impulse loadings are among the most challenging problems in solid mechanics, which encouraged us to put the proposed method to the test. The equilibrium equation with a dimensionless response number (the response number) was successfully determined using the proposed workflow. It is worth mentioning that the regression coefficients obtained from dimensionless learning were consistent across different parameter values set in the analytical solution code. The detail of this problem is explained in section 6.4 of the revised SI.

[Section 2.4 of the main manuscript and Section 6 of the SI are shown on the previous response]

Reviewer #2 (Remarks to the Author)

The manuscript "DATA-DRIVEN DISCOVERY OF DIMENSIONLESS NUMBERS AND SCALING LAWS FROM EXPERIMENTAL MEASUREMENTS" presents a novel method for the discovery of scientific laws expressed by means of non-dimensional numbers. In recent times, there has been a growing interest in the development of machine learning techniques able to distill scientific laws from data, and this manuscript can be seen as a product of this interest.

The authors develop first a method for the determination of possible non-dimensional candidates (Eqs. (3)-(5)). This procedure follows closely ref. [21] of the manuscript, so there is no novelty on it.

Once non-dimensional candidate numbers are determined, the authors develop the method able to unveil their significance in expressing the data. To this end, they develop first a method based on the assumption that one of the relevant non-dimensional numbers is a polynomial combination of the other, Eq. (9). In this case, they choose, rather arbitrarily, a fifth-order polynomial. The rest of the paper is devoted to provide the reader with details on how to determine the coefficients of this polynomial expansion and to give some examples on its performance.

Up to this moment, the relevance of the paper is rather low: they employ a known method for the determination of non-dimensional numbers relevant for the physical phenomenon at hand, and then assume a polynomial dependence of the result on them.

However, at the conclusions section, they choose to introduce one ingredient that makes the method much more interesting to me: they employ the SINDy technique by Kutz and Brunton, based on the just found non-dimensional numbers. The resulting technique is now much more interesting and intellectually appealing, but the price paid for this is the (almost) lack of novelty of the proposal. The resulting method is nothing more than a particular case of the SINDy technique introduced in ref. [15].

In sum, the paper is correct and well written, but in my humble opinion, it lacks of significance. The only novelty is the application of the SINDy technique on top of non-dimensional numbers found through the employ of a known algorithm.

In the revised manuscript, we significantly clarify the contribution of the paper namely: “dimensionless number at the feature level, scaling law at the algebraic equation level, and governing equation at the differential equation level.” We also added several new examples in fluid mechanics and solid mechanics to demonstrate the generality of the proposed method.

To demonstrate the novelty and significance of our work, we revise the description of the method in more detail here demonstrating the novelty of the method (as a result of respected reviewers' comments and suggestions) to complement the reviewer's description.

The method workflow consists of two main workflows to identify scientific knowledge from data at multiple levels: dimensionless number at the feature level, scaling law at the algebraic equation level, and governing equation at the differential equation level. These two workflows are described as follows:

1) Discovering dimensionless numbers and scaling laws in the first workflow:

- a) In this section of the workflow, we developed a novel technique to find dimensionless numbers and scaling laws by incorporating dimensional invariance into machine learning algorithms. A two-level machine learning scheme is implemented in the algorithm to automatically discover dominant and unique dimensionless numbers and scaling laws from limited data. Since the first-level scheme guarantees dimensional invariance (or dimensional homogeneity), many representation learning methods can be used to capture scale-free relationships in the second-level scheme. We demonstrated polynomial and tree-based method XGBoost from Sections 2.1 to 2.3 of the main manuscript using several challenging scientific and engineering problems, including turbulent Rayleigh-Bénard convection, vapor depression dynamics in laser melting of metals, and porosity formation in 3D printing. However, the capability of dimensionless learning can be improved by leveraging more methods, including deep neural networks, symbolic regression, and Bayesian machine learning. It is also worth mentioning that the workflow successfully recovers dimensionless numbers and scaling laws from noisy experimental measurements (not synthetic data) for the aforementioned problems.*
- b) The main difference between the proposed dimensionless learning and classical dimensional analysis is described as follows: the traditional dimensional analysis can only identify unique dimensionless groups for simple problems. For complex multivariable problems (especially newly emerged problems without known governing equations), the traditional dimensional analysis identifies a lot of (mathematically infinite) candidate dimensionless groups by examining the units of a physical system and then tries to figure out the correct dimensionless by fitting experimental data. It is a very time-consuming process that heavily relies on the experience of domain experts. The proposed dimensionless learning, on the other hand, is the first trial to reverse the whole process, i.e., we directly start from experimental data and leverage machine learning approaches to identify dimensionless numbers and scaling laws underlying the experimental data. It has the potential to become an automatic end-to-end identification procedure to discover new dimensionless numbers and scaling behaviors for understanding newly emerged scientific and engineering phenomena in the future.*

2) Identifying consistent governing equations as well as dimensionless numbers from data in the second workflow. Specifically, the second workflow includes two sequentially linked operations.

- a) The first operation is symmetric invariant SINDy, which for the first time incorporates the symmetric invariance to determine the regression coefficients accurately. It applies a physical constraint to the regression coefficients to ensure that all symmetry terms have the same coefficients. The performance of the training has been demonstrated in Section 2.4 of the main manuscript and Section 6 of the SI. By reducing the number of learnable coefficients, this method substantially improves both computational efficiency and accuracy. During this operation, it augments the training data for sparse regression by using the measurements*

swapping technique (explained in Section 6.2 of the SI) and subsequently adding this new set of data to the training data. The important improvement of this approach is that it reduces the required sampling point by at least two orders of magnitude.

- b) The governing equation obtained in the previous step can have varying or constant coefficients. To identify dimensionally consistent governing equations, we implement dimensionless learning to identify the explicit expression for the varying coefficients (i.e., dimensionless groups) for the second operation in this part of the workflow. Specifically, we simplify the second level fitting function of the dimensionless learning as a power law with constant coefficients rather than a polynomial in Sections 2.1 and 2.2 of the main manuscript or XGBoost in Section 2.3 of the main manuscript.
- c) By combining identified terms in the first operation with coefficient expressions in the second operation, the proposed method can identify consistent governing equations (e.g., the Navier-Stokes equation) as well as dimensionless numbers (e.g., the Reynolds number). The significance of this operation is that the governing equation coefficients are dimensionally consistent and thus will not change for cases with different parameters.

The integration of dimensionless learning and SINDy provides a simple yet effective and efficient method for identifying dimensionless governing equations and dimensionless numbers, which has not been attempted in the literature. By integrating dimensionless learning with SINDy, the proposed method can directly discover dimensionally homogeneous differential equations and dimensionless numbers from data, which provides even more insights and interpretations of the physical system. Many data-driven approaches, including SINDy-incorporated methods fall short of achieving a consistent governing equation for the same system with different parameters. For instance, vanilla SINDy-related approaches can only provide a scalar coefficient for each candidate term in their regression library. Yet, these regression coefficients can vary depending on how the simulation or experiment parameters are set. Other advanced SINDy approaches deal with this issue by multiplying the candidate terms with a set of pre-determined parameters or parameterized expressions. Although this can address this inconsistent governing equation problem, it couples the optimization of identifying candidate terms and parameterized coefficients, making the optimization more difficult. This problem can be more difficult to handle if there are many combinations of parametric derivative and non-derivative terms. To find universal governing equations efficiently, the proposed method separates the identification of a consistent governing equation into two steps. The first step is to find a temporary governing equation in which the regression coefficients can be constants or vary depending on how the simulation or experiment parameters are set. In the next step, dimensionless learning is designed to recover the expression of the varying coefficients by leveraging the dimension of the varying coefficients. By combining the results of these two steps, we can obtain a consistent dimensionally homogeneous governing equation.

We also demonstrated that the workflow could find dimensionless numbers and governing laws in noisy data. The analyses are performed on five differential equations, including Navier-Stokes (0.5% noise), Euler (1%), vorticity (1%), the governing equation for spring-mass-damper systems (4%), and dynamic loading beam systems (2%). Even with the noisy data effect, the method successfully discovers the correct governing equations, as demonstrated in Section 2.4 of the main manuscript and Section 6 of the SI. To study the sparse regression library effect, we build a general sparse regression library to achieve more generalizable results. Specifically, we use 29 terms in the vorticity equation case, as described in Section 6.1 of the SI. In general, adding candidate terms to the library relies heavily on the researchers' experience and understanding of the problem. Yet, we provide a guideline for choosing the regression library given in Section 6.5 of the SI.

In conclusion, using the above-mentioned workflows, the proposed method provides a unified way to predict the dimensionless numbers, scaling laws, and governing equations with dimensionless parameters, which has never been done before. In the primary part of the workflow, a two-level machine learning scheme is used to automatically discover dominant and unique dimensionless numbers and scaling laws from limited data. It is worth noting that most existing methods attempt

only to predict coefficients, which are constant numbers with no physical meaning [16, 45, 47, 48]. Despite these approaches in the literature, this method determines the physically meaningful coefficients that are used for dynamical system interpretations (such as the Euler number and the reciprocal of the Reynolds number).

All the aforementioned novelty and advantages are shown in the last paragraph of the introduction, the newly added Section 2.4 and revised discussion section of the main manuscript, and Sections 5.3 and 6 of the SI.

[The last paragraph of the Introduction]

In this study, we propose a mechanistic data-driven approach, called dimensionless learning. This method consists of two main workflows to discover scientific knowledge from data. The first workflow embeds the principle of dimensional invariance (i.e., physical laws are independent on an arbitrary choice of basic units of measurements [1]) into a two-level machine learning scheme to automatically discover dominant dimensionless numbers and scaling laws from noisy experimental measurements of complex physical systems. This invariance incentivizes the learning of scale-invariant and physically interpretable low-dimensional patterns of complex high-dimensional systems. We demonstrate the first workflow by solving three challenging problems in science and engineering with noisy experimental measurements collected from the literature. The problems include turbulent Rayleigh-Benard convection, vapor depression dynamics, and porosity formation during 3D printing. In the second workflow, the dimensionless learning is integrated with sparsity-promoting techniques (such as SINDy [15] and proposed symmetric invariant SINDy) to identify dimensionally homogeneous differential equations and dimensionless numbers from data. The analyses are performed on five differential equations with and without noisy data effect, including Navier-Stokes, Euler, vorticity equations, the governing equations for spring-mass-damper systems and dynamic loading beam systems.

[Section 2.4 of the main manuscript]

Section 2.4 Vorticity form of dimensionless Navier-Stokes equation

In this section, we demonstrate dimensionless learning in discovering the governing equation using the Kármán vortex street problem. The goal is to explicitly identify the well-known dimensionless vorticity equation as well as the Reynolds number (Re) from simulation snapshots. The Re is the ratio of inertial forces to viscous forces, which is frequently used to distinguish between the laminar and turbulent flow.

Fig. 4 shows a schematic of integrating dimensionless learning with a proposed physically enhanced SINDy, namely, symmetric invariant SINDy, to identify the dimensionless governing equations as well as dimensionless number(s). Here, we demonstrate the proposed method for a fluid flow problem involving three cylinders with diameter l , fluid density ρ , dynamic viscosity μ , inlet velocity V , and the pressure difference between the upstream and downstream p_0 . Three CFD simulations for different flow regimes are carried out to generate a dataset for the partial differential equation (PDE) discovery. The velocity and vorticity measurements are sampled sparsely (Fig. 4a).

Figure 4. Integration of dimensionless learning with symmetric invariant SINDy for identifying the Navier-Stokes equation with Reynolds number. (a) Original data are generated from parametric simulations. Because of symmetric invariance, another set of transformed data is obtained by flipping the original data along $y = x$. (b) The original and transformed data are concatenated together for symmetric invariant SINDy, which incorporates symmetric invariance into SINDy to ensure that symmetric invariant terms have the same coefficients. (c) The identified temporary governing equations for each simulation case. Some of the coefficients are close to constant, while others vary depending on the simulation case. All the other candidate terms have zero coefficients. More importantly, symmetry terms (e.g., $u \frac{\partial \omega}{\partial x}$ and $v \frac{\partial \omega}{\partial y}$) have identical coefficients. (d) Dimensionless learning is applied to recover the explicit expression for the varying coefficients. The parametric space to be explored includes five parameters. By incorporating dimensional invariance, we need to optimize basis coefficients $\boldsymbol{\gamma}$ and fitting coefficient β . (e) Substituting the discovered regression coefficients ($1/\text{Re}$) into the temporary governing equation. In this step, a consistent dimensionally homogeneous governing equation, which is identical to the Navier-Stokes equation in a vorticity form, is obtained.

With an assumption that the governing equation is invariant to the symmetric transformation along $y = x$, we flip the original data along $y = x$ for each simulation case to obtain the transformed data. The original and transformed data are then sampled with 4,000 measurements each on the same relative position. Based on these measurements, two regression libraries are built to identify the governing equation using linear and quadratic terms for $u, v, \omega, \frac{\partial \omega}{\partial x}, \frac{\partial^2 \omega}{\partial x^2}, \frac{\partial^2 \omega}{\partial y^2}$. In total, there are 29 terms in the regression library. Unlike SINDy, which uses the library from the original data directly, the proposed symmetric invariant SINDy trains the measurements from the original and transformed data together to implicitly ensure the symmetry terms have the same coefficients. That is, the coefficients for symmetry terms are physically constrained. The sum of symmetry terms, such as $\frac{\partial \omega}{\partial x}$ and $v \frac{\partial \omega}{\partial y}$, are invariant to symmetry transformation along $y = x$. The identified temporary governing equation with only four non-zero regression coefficients is shown in Fig. 4c. ξ_{12} and ξ_{19} are identical and close to a constant, while ξ_6 and ξ_7 are also the same but vary with parameters such as ρ, μ , and so on. The proposed symmetric invariant SINDy is described in detail in Section 6.1.2 of the SI.

Examining Fig. 4c, we notice that the identified temporary governing equations vary from case to case. To identify a consistent governing equation, we apply dimensionless learning to find the expression for varying regression coefficients. The parametric space to be explored for $\xi_6 = \xi_7$ can be expressed as follows:

$$\xi_6 = \xi_7 = f(\mu, \rho, V, l, p_0) \quad (17)$$

In contrast to standard dimensionless learning, we simplify the representation function $f(\cdot)$ as a power law with a constant coefficient rather than a fifth-order polynomial in Sections 2.1 and 2.2 or an XGBoost in Section 2.3. Using pattern search-based optimization, the identified expression for ξ_6 and ξ_7 is $1.083 \frac{\mu}{\rho V l} \approx \frac{\mu}{\rho V l}$, which is the reciprocal of the well-known Reynolds number $\text{Re} = \frac{\mu}{\rho V l}$. Section 6.1.3 of the SI contains detailed information on dimensionless learning. By substituting the constant regression coefficients and this discovered expression into the temporary governing equations, we obtain a consistent dimensionless governing equation as follows:

$$\frac{\partial \omega}{\partial t} = -u \frac{\partial \omega}{\partial x} - v \frac{\partial \omega}{\partial y} + \frac{1}{\text{Re}} \left(\frac{\partial^2 \omega}{\partial x^2} + \frac{\partial^2 \omega}{\partial y^2} \right) \quad (18)$$

which is identical to the well-known vorticity form of the Navier-Stokes equation. We further apply the proposed method to 1% Gaussian noisy data. The proposed method successfully identifies the correct governing equation as Eqns. 18. The detailed results for noisy data are shown in Section 6.1.3 of the SI. More applications in fluid and solid mechanics and dynamics systems with and without noise are demonstrated in Section 6 of the SI.

[Discussion section of the main manuscript]

The proposed dimensionless learning is a powerful technique to identify scientific knowledge from data at multiple levels: dimensionless number at the feature level, scaling law at the algebraic equation level, and governing equation at the differential equation level. Unlike purely data-driven approaches that easily suffer from overfitting on small or noisy datasets, this method incorporates fundamental physical knowledge of dimensional invariance and symmetric invariance as physical constraints or regularizations into data-driven models to perform well on limited and/or noisy data. The embedded physical invariance reduces the learning space and eliminates the strong dependence between variables. This method is a physics-based dimension reduction approach that represents features as dimensionless numbers and transforms data points into a low-dimensional pattern that is unaffected by units and scales. Thus, in addition to being applicable to limited and/or noisy data, the presented approach significantly improves the interpretability of representation learning because dimensionless numbers are physically interpretable. Lower dimension and better interpretability also allow for qualitative and quantitative analysis of the systems of interest. This has been demonstrated in three complex engineering problems in earlier sections.

Another advantage of the embedded dimensional invariance in dimensionless learning is improved generalization capability. To show this, in the vapor depression dynamics case, we compared the performance of dimensionless learning and popular machine learning algorithms on unseen material data points. The proposed method achieves the best generalization in the test set, while all other algorithms only achieve a poor generalization. This improvement is due to ensuring geometric, kinematic, and dynamic similarities based on similitude theory within different systems. A detailed description of the generalization comparison can be found in Section 5.3 of the SI. Aside from dimensional invariance, we also used symmetric invariance in this study. The benefits of symmetric invariance are that it intrinsically ensures symmetry terms have the same coefficients and effectively reduces the number of learnable regression coefficients in SINDy.

Dimensionless learning is also very flexible in terms of choosing the representation learning function because of the proposed two-level optimization scheme. Since the first-level scheme guarantees dimensional invariance (or dimensional homogeneity), many representation learning methods can be used to capture scale-free relationships in the second-level scheme. We demonstrated polynomial and tree-based method XGBoost [22] in the previous sections. However, the capability of dimensionless learning can be improved by leveraging more methods, including deep neural networks [41], symbolic regression [42], and Bayesian machine learning [43].

The optimization of dimensionless learning is different from general regression optimization approaches because only dimensionless numbers with small rational powers are preferred, such as -1, 0.5, 1, or 2, etc. Therefore, instead of searching for the best basis coefficients with a lot of decimals like other neural network-based methods, such as DimensionNet [44], zero-order optimization methods are used in this work. It includes grid search or pattern search-based two-level optimization and can be more efficient in finding the best basis coefficients. No gradient information and learning rate are required and the choice of grid interval is more flexible. Even though these zero-order optimization approaches can get stuck in local minima, increasing the number of initial points can easily eliminate this issue. More detailed pros and cons of different optimization methods are described in Section 3.5 of the SI.

The integration of dimensionless learning and SINDy provides a simple but effective and efficient method for identifying dimensionally homogeneous differential equations and dimensionless numbers from data. These equations and numbers provide even more insights and interpretations of the physical system. Many data-driven approaches, including methods that use SINDy [15, 45] fall short of achieving a consistent governing equation for the same system with different parameters. For instance, the most standard SINDy-related approaches can only provide a scalar coefficient for each candidate term in their regression library. Yet, these regression coefficients can vary depending on how the simulation or experiment parameters are set. Other advanced SINDy approaches deal with this issue by multiplying the candidate terms with a set of pre-determined parameters or parameterized expressions. Although this can address this inconsistent governing equation problem, it couples the optimization of identifying candidate terms and parameterized coefficients, making the optimization more difficult [15, 46]. This problem can be more challenging if there are many combinations of parametric derivative or non-derivative terms. To find universal governing equations efficiently, the proposed method separates the identification of a consistent governing equation into two steps. The first step is to find a temporary governing equation in which the regression coefficients can be constants or vary depending on how the simulation or experiment parameters are set. In the next step, dimensionless learning is designed to recover the expression of the varying coefficients by leveraging the dimension of the varying coefficients. By combining the results of these two steps, we can obtain a consistent dimensionally homogeneous governing equation.

In order to determine the sensitivity and sensibility of the proposed method, we studied three major factors affecting the discovery results. The first factor is the noisy data effect. we demonstrated the proposed algorithm by solving three challenging problems with noisy experimental measurements, which are described in detail in Sections 2.1, 2.2, and 2.3. It is found that in these three problems, even with the noisy data effect, the method achieves high fitting performance in both training and test sets (all R2 scores are greater than 0.95). The second factor is the scarce data effect. Most machine learning algorithms rely on a large amount of data to achieve good generalization and minimum out-of-bag error. However, because of the complexity and cost of experiments, it is not always feasible to obtain a big dataset for engineering problems. To deal with scarce data and obtain a universal model, the proposed method embeds dimensional invariance with input variables and successfully reduces the solution space to a manageable size. The dimensional invariance can be regarded as a physical regularization and changes the model structure, which enables the proposed method to train a universal model with limited data points. For example, even though we only used 182, 90, and 92 experimental measurements, respectively, in three complex engineering examples from Section 2.1 to 2.3, the identified scaling laws fit very well in all these cases. We also compare the proposed method with popular machine learning algorithms, which have poor generalization in the test set, as described in Section 5.3 of the SI. The third factor is the involved variables. We demonstrate that missing necessary variables or involving redundant variables has no effect on the discovered scaling laws in Sections 5.1 and 5.2 of the SI.

For the discovery of governing equations, as the second part of the method, the accuracy of the discovered equation can be influenced by data noise and the setting of the sparse regression library. The noisy data analyses are performed on five differential equations, including Navier-Stokes (0.5%

noise), Euler (1%), vorticity (1%), the governing equations for spring-mass-damper systems (4%) and dynamic loading beam systems (2%). Even with the noisy data effect, the method successfully discovers the correct governing equations, as demonstrated in Section 2.4 of the main manuscript and Section 6 of the SI. The tolerable noise level can be further increased by combining the proposed method with some newly developed approaches which apply physics-informed neural networks and/or deep learning approaches to reduce noise and obtain robust derivatives [45, 47, 48]. To study the sparse regression library effect, we build a general sparse regression library to achieve more generalizable results. Specifically, we use 29 terms in the vorticity equation case, as described in Section 6 of the SI. In general, adding candidate terms to the library relies heavily on the researchers' experience and understanding of the problem. Yet, we provide a guideline for choosing the regression library given in Section 6.5 of the SI.

In summary, the proposed dimensionless learning enables systematic and automatic learning of scale-free low-dimensional laws from a high-dimension parameter space, including many experimental conditions with different parameter settings. It can be applied to a wide range of physical, chemical, and biological systems to discover new dimensionless numbers or modify existing ones. Furthermore, it can be combined with other data-driven methods, such as SINDy, to discover dimensionless differential equations from high-resolution measurements. In material science, the identified compact mathematical expressions provide simple transition rules that translate optimal process parameters from one material (or existing materials) to another (or new materials). Dimensionless learning can reduce complex, highly multivariate problem spaces into descriptions involving only a few dimensionless parameters with clear physical meanings. This approach is particularly useful for engineering problems involving many adjustable parameters with various dimensions or units, such as advanced materials processing and manufacturing [49], microfluidic flow control for precise drug delivery, and solar energy systems design [50].

[Section 5.3 of the SI]

5.3 Comparison of different machine learning algorithms with dimensionless learning

The out-of-bag error is a critical metric for machine learning algorithms, especially when dealing with problems with limited data. In this section, we compare the performance of five popular machine learning algorithms with our proposed method over two small datasets. The proposed method shows the best generalization in datasets with different materials and scales.

5.3.2 Model generalization for different materials

The first dataset is the same as the one used in Section 2.2 of the manuscript. This dataset contains three different materials: titanium alloy (Ti6Al4V), aluminum alloy (Al6061), and stainless steel (SS316). Here, we do not shuffle all the data and split it into training and test sets. We only use Ti6Al4V and Al6061 data in the training set, while other data (SS316) is used in the test set. 5-fold cross-validation is used to select the best parameters of each model.

Fig. 9 shows the R^2 score for each method. Even though the Feed Forward Neural Network (FFNN) achieves very good performance on the training and validation sets, the R^2 score for the test set is below -0.1. Random Forest (RF) has relatively high R^2 scores for training and validation, but it still performs poorly on the test set. On the other hand, the proposed method has the highest R^2 score on the test set, indicating that it has the best generalization in different materials. This improvement is the result of ensuring geometric, kinematic, and dynamic similarities based on similitude theory within different systems.

Figure 10: Comparisons of six candidate models for the generalization of different materials. The training and validation sets come from two materials (titanium alloy (Ti6Al4V) and aluminum alloy (Al6061)), while the test set comes from the other material (stainless steel (SS316)). The highest the R^2 score, the better the model. The y-axis only shows R^2 scores above -0.1. The regression methods include Feed-forward Neural Network (FFNN), Linear regression (LR), Extreme Gradient Boosting (XGBoost), K-Nearest Neighbors (KNN), Random Forest (RF), and the proposed method.

5.3.3 Model generalization for different scales

In this section, we use a rough pipe flow example to demonstrate that the proposed method has the best generalization ability in data with different scales. Pipe flow is a classic example to analyze the fluid flow from laminar to turbulent. Reynolds number (Re) is widely used in fluid mechanics to distinguish between laminar and turbulent flow. In general, several measurements can be collected in the steady state of a pipe flow system: fluid velocity v , dynamic viscosity μ , pipe diameter d , fluid density ρ , and pressure p .

Two types of synthetic data are created to build the dataset: small pipe flow data for the training and validation sets, and large pipe flow data for the test set. All the data are generated randomly. The detailed range of each parameter is shown in Table 5.

Table 5: **Different parameter ranges for small and large pipe flows.**

Source	Parameter	Range	Source	Parameter	Range
Small pipe	d	0.1 - 1 cm	Large pipe	d	10 - 100 cm
	μ	$1 \times 10^{-3} - 1 \times 10^{-2}$ g/cm/s		μ	$1 \times 10^{-5} - 1 \times 10^{-3}$ g/cm/s
	v	100 - 1×10^3 cm/s		v	1 - 10 cm/s
	ρ	$1 \times 10^{-3} - 1 \times 10^{-2}$ g/cm ³		ρ	$1 \times 10^{-4} - 1 \times 10^{-3}$ g/cm ³
	p	$1 \times 10^4 - 1 \times 10^5$ Pa		p	$1 \times 10^4 - 1 \times 10^5$ Pa

Fig. 11 compares the performance of six regression models on the training, validation, and test sets. Because the test set consists entirely of large pipe flow rather than small pipe flow, all machine learning algorithms perform very poorly on the test set (below 0). On the other hand, the proposed method can perform very well on the test set. Even though this dataset is synthetic, it shows that the proposed method can be used for data with different scales.

Figure 11: Comparisons of six candidate models to identify a generalized model for data from different scales. The training and validation sets are generated using small pipe cases, while the test set is associated with large pipe cases. The higher the R^2 score, the more accurate the model. The y-axis only shows R^2 scores greater than -0.1.

[Sections 6.1 and 6.2 of the SI are shown below, Sections 6.3 and 6.4 can be found in the SI.]

6. Governing equation discovery from limited data

In this section, we integrate dimensionless learning with symmetric invariant SINDy or SINDy to identify governing equations as well as dimensionless numbers in fluid and solid mechanics. The identified Partial Differential Equations (PDEs) include the vorticity equation, Navier-Stokes equation, Euler equation, and the governing equation for spring-mass-damper systems.

6.1 Vorticity equation

6.1.1 Data generation and preprocessing

The vorticity equation is a vorticity form of the Navier-Stokes equation that describes fluid flow by using the evolution of vorticity. To generate the original data, we conducted three sets of simulations for the Kármán vortex street problem with three circular cylinders using Flow3D [10]. The geometry parameters include the diameter of one cylinder l , the length in x direction l_x , and the length in y direction l_y . The operational and material parameters considered for this problem include the inlet flow velocity V , the dynamics viscosity μ , the fluid density ρ , and the pressure difference between upstream and downstream p_0 . Other simulation parameters include the number of mesh cells in x and y directions (N_x and N_y). The detailed information on these parameters is shown in Table 6.

Table 6: Simulation parameters of fluid flow through three circular cylinders to identify vorticity equation.

Parameter Unit	l cm	V cm/s	μ kg/cm/s	ρ g/cm ³	p_0 Pa	Re 1	l_x cm	l_y cm	N_x 1	N_y 1
Case 1	0.0850	68.00	0.1156	0.0010	1.30×10^4	50	0.9	0.4	500	222
Case 2	0.1000	90.00	0.1100	0.0011	8.42×10^4	90	0.9	0.4	500	222
Case 3	0.1000	100.00	0.1100	0.0011	8.53×10^4	100	0.9	0.4	500	222

Figure 12: Vorticity contour plot from a simulation with $Re=100$. The red color rectangle represents the sampling region.

Table 7: Coordinate transformation using flipping operation.

Original data	t	x	y	u	v	ω
Transformed data	$t^0 = t$	$x^0 = y$	$y^0 = x$	$u^0 = v$	$v^0 = u$	$\omega^0 = -\omega$

To identify the governing equation from data, 100 time-step snapshots are prepared and sparsely sampled. Fig. 12 shows a typical snapshot and the sampling region. In the sampling region, 4,000 data points are randomly sampled. For each data point, there are six variables: a specific time step (t), two spatial Cartesian coordinates (x and y), and three time-varying measurements (velocity in x direction u , velocity in y direction v , and vorticity ω). These six variables are non-dimensionalized by the following characteristic variables: characteristic length l (the cylinder diameter), characteristic velocity V (inlet flow velocity), characteristic time $t_{ref} = \frac{l}{V}$, and characteristic vorticity $\omega_{ref} = \frac{l}{V}$.

For each simulation case, we flip the original data along $y = x$ to obtain symmetrically transformed data. The original and transformed variables are shown in Table 7.

6.1.2 Symmetric invariant SINDy

After augmenting the dataset, we conduct symmetric invariant SINDy for each case in Table 6 to identify the corresponding temporary governing equations. Fig. 13 shows a schematic of concatenating the sparse regression terms from the original and transformed data.

There are several implicit benefits for sparse regression because of this concatenation. First, terms with the same absolute value in the original and transformed data must have the same coefficients, indicating that the regression coefficients are physically constrained. For example, the coefficients for $\frac{\partial \omega}{\partial x}$ (the fourth column) and $\frac{\partial \omega}{\partial y}$ (the fifth column) must be the same ($\xi_5 = \xi_6$), and the coefficients for u (the first column) and v (the second column) must be zero ($\xi_1 = \xi_2 = 0$). Second, the learnable coefficients can be reduced because of the physical constraint, which reduces the solution space and the required number of training data accordingly. Third, data augmentation is one of the important regularization techniques to avoid overfitting [11]. By leveraging symmetric invariance, the training data is doubled, which significantly improves the learning accuracy for the regression coefficients.

In this case, we include 29 general terms in the library as the candidate terms:

$$\frac{\partial \omega}{\partial t} = [u, v, \omega, \frac{\partial \omega}{\partial x}, \frac{\partial \omega}{\partial y}, \frac{\partial^2 \omega}{\partial x^2}, \frac{\partial^2 \omega}{\partial y^2}, \frac{\partial^2 \omega}{\partial x \partial y}, uu, uv, u\omega, u \frac{\partial \omega}{\partial x}, u \frac{\partial \omega}{\partial y}, u \frac{\partial^2 \omega}{\partial x^2}, u \frac{\partial^2 \omega}{\partial y^2}, u \frac{\partial^2 \omega}{\partial x \partial y}, vv, v\omega, v \frac{\partial \omega}{\partial x}, v \frac{\partial \omega}{\partial y}, v \frac{\partial^2 \omega}{\partial x^2}, v \frac{\partial^2 \omega}{\partial y^2}, v \frac{\partial^2 \omega}{\partial x \partial y}, \omega\omega, \omega \frac{\partial \omega}{\partial x}, \omega \frac{\partial \omega}{\partial y}, \omega \frac{\partial^2 \omega}{\partial x^2}, \omega \frac{\partial^2 \omega}{\partial y^2}, \omega \frac{\partial^2 \omega}{\partial x \partial y}] \xi, \quad (27)$$

where the dimension of ξ is 29×1 and the derivative terms are calculated by a local polynomial interpolation approach [12, 13]. A second-order polynomial is fitted to each data point using eight neighboring points, and the derivatives can be calculated by the fitted polynomial.

Figure 13: Concatenating the regression library for symmetric invariant SINDy from the original and transformed data. Columns with the same color have the same absolute value.

Table 8: Dataset including computed non-zero regression coefficients and parameters for the vorticity simulation.

Parameter Unit	l cm	V cm/s	μ kg/cm/s	ρ g/cm ³	p_0 Pa	Re 1	$\xi_{12} = \xi_{19}$ 1	$\xi_6 = \xi_7$ 1
Case 1	0.0850	68.00	0.1156	0.0010	1.30×10^3	50	-0.9925	0.0212
Case 2	0.1000	90.00	0.1100	0.0011	8.42×10^4	90	-0.9909	0.0126
Case 3	0.1000	100.00	0.1100	0.0011	8.53×10^4	100	-0.9941	0.01101

To find sparse regression coefficients ξ , we use a sequential threshold least-squares algorithm [12]. After optimization, each case will have only four non-zero regression coefficients. Thus, we are able to construct a non-zero coefficient dataset in combination with the parameters (and their values) listed in Table 6 for the three different cases, as shown in Table 8.

From Table 8, we notice that $\xi_{12} = \xi_{19}$ is very close to a constant value (-1), while $\xi_6 = \xi_7$ varies with different cases. Therefore, the temporary governing equation can be simplified as:

$$\frac{\partial \omega}{\partial t} = -u \frac{\partial \omega}{\partial x} - v \frac{\partial \omega}{\partial y} + \xi_6 \left(\frac{\partial^2 \omega}{\partial x^2} + \frac{\partial^2 \omega}{\partial y^2} \right) \quad (28)$$

6.1.3 Dimensionless learning

To obtain a consistent governing equation, we apply dimensionless learning to identify the expression of ξ_6 and ξ_7 . The causal relationship can be expressed as:

$$\xi_6 = \xi_7 = f(\mu, l, V, \rho, p_0). \quad (29)$$

The dimension matrix \mathbf{D} for Eqns. 29 is:

$$\begin{bmatrix} -1 & 1 & 1 & -3 & -1 \\ -1 & 0 & -1 & 0 & -2 \\ 1 & 0 & 0 & 1 & 1 \end{bmatrix} \quad (30)$$

The columns of the matrix, from right to left, represent the dimensions of $\mu, l, V, \rho,$ and p_0 respectively. The rows, from top to bottom, represent the dimensions of length, time, and mass respectively.

Unlike Sections 2.1 and 2.2 of the main manuscript, where a polynomial is used to fit the relationship between dimensionless numbers, we can simplify the fitting function $f(\cdot)$ in Eqns. 29 as a power law with a constant coefficient β . Then, the Eqns. 29 can be simplified as:

$$\xi_6 = \xi_7 = \beta \mu^{w_1} l^{w_2} V^{w_3} \rho^{w_4} p_0^{w_5}, \quad (31)$$

where $w = [w_1, w_2, w_3, w_4, w_5]^T$ denotes the powers that generate the dimensionless number and are to be determined.

The powers w need to satisfy the following linear system of equations:

$$\mathbf{D}w = 0. \quad (32)$$

Furthermore, the solutions of the linear system (Eqns. 32) can be written as a linear combination of two basis vectors w_{b1} and w_{b2} :

$$w = \gamma_1 w_{b1} + \gamma_2 w_{b2}, \quad (33)$$

where $w_{b1} = [-1, 1, 1, 1, 0]^T$ and $w_{b2} = [-1, 1, -1, 0, 1]^T$.

To summarize, there are three unknown parameters in the identification of ξ_6 and ξ_7 : two basis coefficients (γ_1 and γ_2) and a constant coefficient β . The training data come from the non-zero regression coefficient dataset in Table 8. Applying a pattern search-based optimization method with a grid search range of $[-2, 2]$ and a grid interval of 0.5, the optimized expressions for ξ_6 and ξ_7 can be written as:

$$\xi_6 = \xi_7 = 1.083 \frac{\mu}{\rho v l} \approx \frac{\mu}{\rho v l}. \quad (34)$$

This form is similar to the reciprocal of the classical Reynolds number, indicating that the integration of dimensionless learning can directly identify the expression of sparse regression coefficients. By substituting the expression for ξ_6 and ξ_7 (Eqns. 34) into Eqns. 28, we obtain a consistent dimensionless governing equation for all cases:

$$\frac{\partial \omega}{\partial t} = -u \frac{\partial \omega}{\partial x} - v \frac{\partial \omega}{\partial y} + \frac{1}{\text{Re}} \left(\frac{\partial^2 \omega}{\partial x^2} + \frac{\partial^2 \omega}{\partial y^2} \right), \quad (35)$$

which is identical to the well-known Navier-Stokes equation in a vorticity form.

Furthermore, we test the noisy data effect on this problem. After adding 1% Gaussian noise to the measurements, the identified regression coefficients are: (1) case 1, $\xi_{12} = \xi_{19} = -0.9994$, $\xi_6 = \xi_7 = 0.0206$; (2) case 2, $\xi_{12} = \xi_{19} = -0.9909$, $\xi_6 = \xi_7 = 0.0126$; and (3) case 3, $\xi_{12} = \xi_{19} = -0.9941$, $\xi_6 = \xi_7 = 0.0111$. Applying dimensionless learning, both ξ_6 and ξ_7 are identified as

$1.0581 \frac{1}{\text{Re}}$, which is close to the reciprocal of the Reynolds number. Therefore, even with the noisy data effect, the proposed method can identify the governing equation.

6.2 Navier-Stokes equation

In this section, we demonstrate that the dimensionless learning in conjunction with symmetric invariant SINDy can be used to identify the Navier-Stokes equation in a Cartesian coordinate system. We performed three series of simulations for the Kármán vortex street problem with three circular cylinders to prepare the original data using Flow3D. The simulation parameters with their values for different cases are shown in Table 10. The time-varying measurements include a specific time step (t), two spatial Cartesian coordinates (x and y), and three time-varying measurements (fluid velocity in x direction u , velocity in y direction v , and pressure p).

Following the same procedures described in Section 6.1, the regression model is defined as:

$$\frac{\partial u^*}{\partial t} = [u^*, uu^*, vu^*, \frac{\partial u^*}{\partial x}, \frac{\partial u^*}{\partial y}, \frac{\partial^2 u^*}{\partial x^2}, \frac{\partial^2 u^*}{\partial y^2}, u \frac{\partial u^*}{\partial x}, u \frac{\partial u^*}{\partial y}, u \frac{\partial^2 u^*}{\partial x^2}, u \frac{\partial^2 u^*}{\partial y^2}, v \frac{\partial u^*}{\partial x}, v \frac{\partial u^*}{\partial y}, v \frac{\partial^2 u^*}{\partial x^2}, v \frac{\partial^2 u^*}{\partial y^2}, p^*, \frac{\partial p^*}{\partial x}] \xi, \quad (36)$$

Table 9: Simulation parameters for the fluid flow through three circular cylinders to identify the Navier-Stokes equation.

Parameter Unit	l cm	V cm/s	μ kg/cm/s	ρ g/cm ³	p_0 Pa	Re 1	l_x cm	l_y cm	N_x 1	N_y 1
Case 1	0.1000	100.00	0.1100	0.0011	8.53×10^5	100	0.9	0.4	500	222
Case 2	0.0950	165.43	0.1040	0.0011	8.53×10^5	170	0.9	0.4	500	222
Case 3	0.1050	170.43	0.1020	0.0011	9.13×10^5	200	0.9	0.4	500	222

Table 10: Identified regression coefficients accompanied by several ground truth dimensionless numbers for the Navier-Stokes problem (clean data).

Case ID	$\xi_8 = \xi_{13}$	$\xi_6 = \xi_7$	ξ_{17}	Re	1/Re	Eu
Case 1	-0.9759	0.0105	-14127.23	100	0.0100	14545.45
Case 2	-0.9814	0.0063	-3850.28	170	0.0059	3897.63
Case 3	-0.9820	0.0059	-2977.75	200	0.0050	3019.97

where ξ is a vector with 17 terms, u^* is the combination of u and v , v^* is the combination of v and u , and p^* is the combination of p and p . It is worth mentioning that u^* , v^* , and p^* are designed for the flipping operation so that we can concatenate the regression libraries for the original and transformed data and train them together. This operation helps to incorporate symmetric invariance into the learning scheme. After the transformation, $u^{*'}$, and $v^{*'}$ become v^* and u^* , respectively, while $p^{*'}$ remains the same as p^* .

Following the same procedure as Section 6.1, we obtain the optimized sparse regression coefficients. There are only five terms with non-zero regression coefficients, as shown in Table 10.

Because $\xi_8 = \xi_{13}$ are close to -1, it is reasonable to assign -1 to both ξ_8 and ξ_{13} . Next, we apply dimensionless learning to identify the expression for the other three non-zero coefficients. Using the same pattern search optimization described in Section 6.1, the identified ξ_6 and ξ_7 are $1.0746 \frac{\mu}{\rho \nu l}$, which can be simplified to $1/\text{Re}$. The ξ_{17} is identified as $-0.9729 \frac{\nu_0}{\rho \nu^2}$ and can be simplified as the negative of a well-known dimensionless number, Euler number, Eu . By substituting all non-zero regression coefficients into Eqns. 36, we obtain a consistent governing equation:

$$\frac{\partial u^*}{\partial t} = -u \frac{\partial u^*}{\partial x} - \nu \frac{\partial u^*}{\partial y} - \text{Eu} \frac{\partial p^*}{\partial x} + \frac{1}{\text{Re}} \left(\frac{\partial^2 u^*}{\partial x^2} + \frac{\partial^2 u^*}{\partial y^2} \right), \quad (37)$$

which is identical to the Navier-Stokes momentum equation written for the combination of x and y directions.

We also test the noisy data effect on this problem by imposing 0.5% Gaussian noise to the measurements. The identified non-zero coefficients are shown in Table 11.

Using dimensionless learning, we can identify the expression for ξ_6 and ξ_7 as $1.0473 \frac{\mu}{\rho \nu l}$. Similarly, the expression for ξ_{17} is identified as -0.9687Eu . Compared to the clean data results in Table 10, ξ_8 and ξ_{13} have higher errors. By assuming the ξ_8 and ξ_{13} as $-\text{Eu}$, we obtain the same equation as Eqns. 37.

Table 11: Identified regression coefficients accompanied by several ground truth dimensionless numbers for the Navier-Stokes problem (noisy data).

Case ID	$\xi_8 = \xi_{13}$	$\xi_6 = \xi_7$	ξ_{17}	Re	1/Re	Eu
Case 1	-0.9668	0.0104	-14139.49	100	0.0100	14545.45
Case 2	-0.9378	0.0063	-3699.04	170	0.0059	3897.63
Case 3	-0.9217	0.0053	-2785.92	200	0.0050	3019.97

Table 12: Identified regression coefficients accompanied by several ground truth dimensionless numbers for the Euler equation example (clean data).

Case ID	$\xi_8 = \xi_{13}$	ξ_{17}	Re	Eu
Case 1	-1.0436	-14127.23	100	14545.45
Case 2	-1.0016	-3896.14	170	3897.63
Case 3	-1.0177	-3004.48	200	3019.97

Table 13: Identified regression coefficients accompanied by several ground truth dimensionless numbers for the Euler equation example (noisy data).

Case ID	$\xi_8 = \xi_{13}$	ξ_{17}	Re	Eu
Case 1	-1.0361	-14015.25	100	14545.45
Case 2	-0.9946	-3873.86	170	3897.63
Case 3	-1.0088	-2953.95	200	3019.97

It is worth mentioning that we choose a different sampling region for the same data in this problem, but all other procedures remain the same as in the previous example. The identified non-zero regression coefficients for clean and noisy data (1% Gaussian noise) are shown in Table 12 and Table 13.

From Table 12 and Table 13, we can see that ξ_8 and ξ_{13} are very close to -1, and ξ_{17} is the only varying regression coefficient. To identify the expression of ξ_{17} for clean and noisy data, we use dimensionless learning and identify the expressions are $-0.9671Eu$ and $-0.9661Eu$, respectively. By simplifying ξ_{17} to $-Eu$ and substituting all non-zero regression coefficients into Eqns. 36, we obtain the governing equation as below:

$$\frac{\partial u^*}{\partial t} = -u \frac{\partial u}{\partial x} - v \frac{\partial u}{\partial y} - Eu \frac{\partial p^*}{\partial x} \quad (38)$$

which is identical to the Euler equation. The reason for simplifying the Navier-Stokes equation to the Euler equation is that the viscous effects of the fluid far from the cylinders and walls are negligible in this problem.

[More demonstrated problems can be found in Sections 6.3 and 6.4 can be found in the SI, including spring-mass-damper problems and dynamic response of impulsively loaded plastic beam problems.]

[Added references of the main manuscript]

[16] Brunton, Steven L and Proctor, Joshua L and Kutz, J Nathan. Discovering governing equations from data by sparse identification of nonlinear dynamical systems. Proceedings of the national academy of sciences, 113(15):3932–3937, 2016

[45] Chen, Zhao and Liu, Yang and Sun, Hao. Physics-informed learning of governing equations from scarce data. Nature communications, 12(1):1–13, 2021.

[47] Zhang, Zhiming and Liu, Yongming. A robust framework for identification of PDEs from noisy data. Journal of Computational Physics, 446:110657, 2021.

[48] Rao, Chengping and Ren, Pu and Liu, Yang and Sun, Hao. Discovering Nonlinear PDEs from Scarce Data with Physics-encoded Learning. arXiv preprint arXiv:2201.12354, 2022.

Reviewer #3 (Remarks to the Author)

The paper presents three strategies to learn dimensionless numbers and scaling laws for complex systems, respectively relying on grid search, gradient descent and pattern search. The proposed approaches are shown to result in physically interpretable dimensionless parameters and dynamic models, providing a compact yet informative mean to understand the behavior of complex systems. The effectiveness of the approaches in learning dimensionless parameters is shown over two case studies, respectively devoted to characterize turbulent Rayleigh-Bérnard convection and the porosity formation in 3D printed materials. On the other hand, the approach is applied to a linear spring-damper-mass system to highlight its benefits in retrieving dimensionless governing equations for dynamical systems.

Overall, the contribution seems novel, the proposed approaches sound and most of the presented case study suitably support the claims of the authors. Although I believe that the contribution can have a positive impact, especially for what concerns the estimation of dimensionless parameters, I have some concerns on how the proposed strategies compare to alternative approaches and I have some perplexities on some features of the presented techniques. As such, I think the paper should be revised to be accepted.

We appreciate the respected reviewer's time and instructive comments. The reviewer asked interesting questions and made great suggestions about our work, which improved its overall quality. Inspired by the reviewer's suggestions and comments, we revised the manuscript and SI extensively to resolve confusion, explained the detailed information of the proposed method, and compared it to other approaches.

First of all, I was wondering how the methods compare with suitably constructed Neural Networks (e.g., the one proposed by some of the authors in "Hierarchical Deep Learning Neural Network (HiDeNN): An artificial intelligence (AI) framework for computational science and engineering"). I acknowledge that Neural Networks might be less interpretable than the proposed models, but I feel that this consideration is likely to hold only when simple parameterisations of the relationship between the input and output dimensionless parameters are considered. I thus think a comparison would be important to highlight the benefit of the introduced approach.

Thank you for pointing out the HiDeNN approach and suggesting a comparison.

When compared to the DimensionNet proposed in the HiDeNN paper, the dimensionless learning proposed in this paper is general framework to discover dimensionless numbers and scaling laws with a two-level optimization scheme. It covers the DimensionNet as a specific example. Furthermore, this framework is further improved in this manuscript in terms of the efficiency and flexibility of training and the simplicity of model selection.

- First, the two-level dimensionless learning is very efficient in training. The proposed method, in particular, divides training into two schemes: first-level training for the basis coefficients and second-level training for the best representation or fitting function. Instead of obtaining basis coefficients with many decimals like DimensionNet, basis coefficients can be searched in grids with certain intervals for the first-level training. In addition, we can choose polynomial as a fitting function in the second-level optimization, which means the optimization is very fast and does not require so many hyperparameters. However, DimensionNet relies on Deep Neural Network training in the second level of training, which requires many more iterations and careful Neural Network design such as the number of hidden layers and neurons.*
- Second, the proposed method is more flexible and superior than DimensionNet. Depending on their needs and preferences, users can select different training schemes for each level of optimization. In most cases, we can assume that the scaling law is a simple curve or straight line, making a polynomial an excellent choice for the second-level optimization. However, different fitting functions can be used for more complex cases. The latter was a case in Section 2.3 of the manuscript in which we used an advanced regression method called XGBoost to fit the two input dimensionless numbers with one output dimensionless number and achieved a very good fitting performance. However, the model structure of DimensionNet can only be Neural Networks.*
- Third, the loss function of the proposed method is only mean squared error, while DimensionNet uses regularization terms to constrain the solution space. Because of the regularization term, we need to assign weights for each term and have more hyperparameters. Therefore, the proposed method requires fewer hyperparameters to fine-tune.*

- *Fourth, the proposed two-level scheme makes it easier to select the best scaling law and dimensionless numbers. After determining the basis coefficients in the first level of training, we can directly train a fitting function for the next step. However, DimensionNet requires analyzing the distribution of the learned weights for the Scaling Network and identifying patterns behind the learned weights, which can be challenging in complex cases.*

In addition, we agree with the reviewer that a more advanced fitting function in the second-level scheme is necessary for complex problems. We demonstrate this in Section 2.3 of the manuscript, where we fit the data with the XGBoost regression model. It is also worth mentioning that the XGBoost model can be replaced by a Neural Network or other advanced models.

We added a brief description in the fourth paragraph of the discussion section of the main manuscript.

[The fourth paragraph of the discussion section of the main manuscript] The optimization of dimensionless learning is different from general regression optimization approaches because only dimensionless numbers with small rational powers are preferred, such as -1, 0.5, 1, or 2, etc. Therefore, instead of searching for the best basis coefficients with a lot of decimals like other neural network-based methods, such as DimensionNet [44] (HiDeNN paper), zero-order optimization methods are used in this work. It includes grid search or pattern search-based two-level optimization and can be more efficient in finding the best basis coefficients. No gradient information and learning rate are required and the choice of grid interval is more flexible. Even though these zero-order optimization approaches can get stuck in local minima, increasing the number of initial points can easily eliminate this issue. More detailed pros and cons of different optimization methods are described in Section 3.5 of the SI.

Secondly, I believe that the case study considered to show the capability of the method to reconstruct the governing equations is excessively simple. Indeed, for the considered system one can use standard machine learning (and system identification) techniques to learn its model from data with little effort. Therefore, why would one prefer your approaches to fairly standard (and easier) ones? The choice of the proposed methods over existing ones should be motivated and a comparison highlighting its benefits over state-of-the-art techniques should be presented to support the conclusions drawn by the authors. Alternatively, I suggest them to consider a different (and possibly more complex) system to assess the performance of the approach presented in the work.

Following this suggestion, we demonstrated the proposed method in more complex systems in the revised paper. The proposed method is an integration of dimensionless learning and SINDy. It provides a simple yet effective and efficient method for identifying dimensionless governing equations and dimensionless numbers. By integrating dimensionless learning with SINDy, the proposed method can directly discover dimensionally homogeneous differential equations and dimensionless numbers from data, which provides even more insights and interpretations of the physical system. Many data-driven approaches, including SINDy-incorporated methods fall short of achieving a consistent governing equation for the same system with different parameters. For instance, the most standard SINDy-related approaches can only provide a scalar coefficient for each candidate term in their regression library. Yet, these regression coefficients can vary depending on how the simulation or experiment parameters are set. Other advanced SINDy approaches deal with this issue by multiplying the candidate terms with a set of pre-determined parameters or parameterized expressions. Although this can address this inconsistent governing equation problem, it couples the optimization of identifying candidate terms and parameterized coefficients, making the optimization more difficult. This problem can be more difficult to handle if there are many combinations of parametric derivative and non-derivative terms. To find universal governing equations efficiently, the proposed method separates the identification of a consistent governing equation into two steps. The first step is to find a temporary governing equation in which the regression coefficients can be constants or vary depending on how the simulation or experiment parameters are set. In the next step, dimensionless

learning is designed to recover the expression of the varying coefficients by leveraging the dimension of the varying coefficients. By combining the results of these two steps, we can obtain a consistent dimensionally homogeneous governing equation.

The proposed method has been demonstrated to identify Navier-Stokes, Euler, vorticity equations, and the governing equations for spring-mass-damper systems and dynamic loading beam systems with and without noise. Even with noisy data effects, the proposed method can discover the correct governing equations. A summary of this response can be found in the fifth and seventh paragraphs of the discussion section in the main manuscript. The detailed description can be found in Section 2.4 of the main text and Section 6 of the SI.

[The fifth paragraph of the discussion section in the main manuscript]

The integration of dimensionless learning and SINDy provides a simple but effective and efficient method for identifying dimensionally homogeneous differential equations and dimensionless numbers from data. These equations and numbers provide even more insights and interpretations of the physical system. Many data-driven approaches, including methods that use SINDy [15, 45] fall short of achieving a consistent governing equation for the same system with different parameters. For instance, the most standard SINDy-related approaches can only provide a scalar coefficient for each candidate term in their regression library. Yet, these regression coefficients can vary depending on how the simulation or experiment parameters are set. Other advanced SINDy approaches deal with this issue by multiplying the candidate terms with a set of pre-determined parameters or parameterized expressions. Although this can address this inconsistent governing equation problem, it couples the optimization of identifying candidate terms and parameterized coefficients, making the optimization more difficult [15, 46]. This problem can be more challenging if there are many combinations of parametric derivative or non-derivative terms. To find universal governing equations efficiently, the proposed method separates the identification of a consistent governing equation into two steps. The first step is to find a temporary governing equation in which the regression coefficients can be constants or vary depending on how the simulation or experiment parameters are set. In the next step, dimensionless learning is designed to recover the expression of the varying coefficients by leveraging the dimension of the varying coefficients. By combining the results of these two steps, we can obtain a consistent dimensionally homogeneous governing equation.

[The seventh paragraph of the discussion section in the main manuscript]

For the discovery of governing equations, as the second part of the method, the accuracy of the discovered equation can be influenced by data noise and the setting of the sparse regression library. The noisy data analyses are performed on five differential equations, including Navier-Stokes (0.5% noise), Euler (1%), vorticity (1%), the governing equation for spring-mass-damper systems (4%), and dynamic loading beam systems (2%). Even with the noisy data effect, the method successfully discovers the correct governing equations, as demonstrated in Section 2.4 of the main manuscript and Section 6 of the SI. The tolerable noise level can be further increased by combining the proposed method with some newly developed approaches which apply physics-informed neural networks and/or deep learning approaches to reduce noise and obtain robust derivatives [45, 47, 48]. To study the sparse regression library effect, we build a general sparse regression library to achieve more generalizable results. Specifically, we use 29 terms in the vorticity equation case, as described in Section 6 of the SI. In general, adding candidate terms to the library relies heavily on the researchers' experience and understanding of the problem. Yet, we provide a guideline for choosing the regression library given in Section 6.5 of the SI.

[Section 2.4 of the main manuscript]

Section 2.4 Vorticity form of dimensionless Navier-Stokes equation

In this section, we demonstrate dimensionless learning in discovering the governing equation using the Kármán vortex street problem. The goal is to explicitly identify the well-known dimensionless vorticity equation as well as the Reynolds number (Re) from simulation snapshots. The Re is the ratio

of inertial forces to viscous forces, which is frequently used to distinguish between the laminar and turbulent flow.

Fig. 4 shows a schematic of integrating dimensionless learning with a proposed physically enhanced SINDy, namely, symmetric invariant SINDy, to identify the dimensionless governing equations as well as dimensionless number(s). Here, we demonstrate the proposed method for a fluid flow problem involving three cylinders with diameter l , fluid density ρ , dynamic viscosity μ , inlet velocity V , and the pressure difference between the upstream and downstream p_0 . Three CFD simulations for different flow regimes are carried out to generate a dataset for the partial differential equation (PDE) discovery. The velocity and vorticity measurements are sampled sparsely (Fig. 4a).

Figure 4. Integration of dimensionless learning with symmetric invariant SINDy for identifying the Navier-Stokes equation with Reynolds number. (a) Original data are generated from parametric simulations. Because of symmetric invariance, another set of transformed data is obtained by flipping the original data along $y = x$. (b) The original and transformed data are concatenated together for symmetric invariant SINDy, which incorporates symmetric invariance into SINDy to ensure that symmetric invariant terms have the same coefficients. (c) The identified temporary governing equations for each simulation case. Some of the coefficients are close to constant, while others vary depending on the simulation case. All the other candidate terms have zero coefficients. More importantly, symmetry terms (e.g., $u \frac{\partial \omega}{\partial x}$ and $v \frac{\partial \omega}{\partial y}$) have identical coefficients. (d) Dimensionless learning is applied to recover the explicit expression for the varying coefficients. The parametric space to be explored includes five parameters. By incorporating dimensional invariance, we need to optimize basis coefficients $\boldsymbol{\gamma}$ and fitting coefficient β . (e) Substituting the discovered regression coefficients ($1/\text{Re}$) into the temporary governing equation. In this step, a consistent dimensionally homogeneous governing equation, which is identical to the Navier-Stokes equation in a vorticity form, is obtained.

With an assumption that the governing equation is invariant to the symmetric transformation along $y = x$, we flip the original data along $y = x$ for each simulation case to obtain the transformed data. The original and transformed data are then sampled with 4,000 measurements each on the same relative position. Based on these measurements, two regression libraries are built to identify the governing equation using linear and quadratic terms for $u, v, \omega, \frac{\partial \omega}{\partial x}, \frac{\partial^2 \omega}{\partial x^2}, \frac{\partial^2 \omega}{\partial y^2}$. In total, there are 29 terms in the regression library. Unlike SINDy, which uses the library from the original data directly, the proposed symmetric invariant SINDy trains the measurements from the original and transformed data together to implicitly ensure the symmetry terms have the same coefficients. That is, the

coefficients for symmetry terms are physically constrained. The sum of symmetry terms, such as $\frac{\partial \omega}{\partial x}$ and $v \frac{\partial \omega}{\partial y}$, are invariant to symmetry transformation along $y = x$. The identified temporary governing equation with only four non-zero regression coefficients is shown in Fig. 4c. ξ_{12} and ξ_{19} are identical and close to a constant, while ξ_6 and ξ_7 are also the same but vary with parameters such as ρ, μ , and so on. The proposed symmetric invariant SINDy is described in detail in Section 6.1.2 of the SI.

Examining Fig. 4c, we notice that the identified temporary governing equations vary from case to case. To identify a consistent governing equation, we apply dimensionless learning to find the expression for varying regression coefficients. The parametric space to be explored for $\xi_6 = \xi_7$ can be expressed as follows:

$$\xi_6 = \xi_7 = f(\mu, \rho, V, l, p_0) \quad (17)$$

In contrast to standard dimensionless learning, we simplify the representation function $f(\cdot)$ as a power law with a constant coefficient rather than a fifth-order polynomial in Sections 2.1 and 2.2 or an XGBoost in Section 2.3. Using pattern search-based optimization, the identified expression for ξ_6 and ξ_7 is $1.083 \frac{\mu}{\rho V l} \approx \frac{\mu}{\rho V l}$, which is the reciprocal of the well-known Reynolds number $Re = \frac{\mu}{\rho V l}$. Section 6.1.3 of the SI contains detailed information on dimensionless learning. By substituting the constant regression coefficients and this discovered expression into the temporary governing equations, we obtain a consistent dimensionless governing equation as follows:

$$\frac{\partial \omega}{\partial t} = -u \frac{\partial \omega}{\partial x} - v \frac{\partial \omega}{\partial y} + \frac{1}{Re} \left(\frac{\partial^2 \omega}{\partial x^2} + \frac{\partial^2 \omega}{\partial y^2} \right) \quad (18)$$

which is identical to the well-known vorticity form of the Navier-Stokes equation. We further apply the proposed method to 1% Gaussian noisy data. The proposed method successfully identifies the correct governing equation as Eqns. 18. The detailed results for noisy data are shown in Section 6.1.3 of the SI. More applications in fluid and solid mechanics and dynamics systems with and without noise are demonstrated in Section 6 of the SI.

[Sections 6.1 and 6.2 of the SI are shown below, Sections 6.3 and 6.4 can be found in the SI.]

6. Governing equation discovery from limited data

In this section, we integrate dimensionless learning with symmetric invariant SINDy or SINDy to identify governing equations as well as dimensionless numbers in fluid and solid mechanics. The identified Partial Differential Equations (PDEs) include the vorticity equation, Navier-Stokes equation, Euler equation, and the governing equation for spring-mass-damper systems.

6.1 Vorticity equation

6.1.1 Data generation and preprocessing

The vorticity equation is a vorticity form of the Navier-Stokes equation that describes fluid flow by using the evolution of vorticity. To generate the original data, we conducted three sets of simulations for the Kármán vortex street problem with three circular cylinders using Flow3D [10]. The geometry parameters include the diameter of one cylinder l , the length in x direction l_x , and the length in y direction l_y . The operational and material parameters considered for this problem include the inlet flow velocity V , the dynamics viscosity μ , the fluid density ρ , and the pressure difference between upstream and downstream p_0 . Other simulation parameters include the number of mesh cells in x and y directions (N_x and N_y). The detailed information on these parameters is shown in Table 6.

Table 6: Simulation parameters of fluid flow through three circular cylinders to identify vorticity equation.

Parameter Unit	l cm	V cm/s	μ kg/cm/s	ρ g/cm ³	p_0 Pa	Re	l_x cm	l_y cm	N_x 1	N_y 1
Case 1	0.0850	68.00	0.1156	0.0010	1.30×10^4	50	0.9	0.4	500	222
Case 2	0.1000	90.00	0.1100	0.0011	8.42×10^4	90	0.9	0.4	500	222
Case 3	0.1000	100.00	0.1100	0.0011	8.53×10^4	100	0.9	0.4	500	222

Figure 12: Vorticity contour plot from a simulation with Re=100. The red color rectangle represents the sampling region.

Table 7: Coordinate transformation using flipping operation.

Original data	t	x	y	u	v	ω
Transformed data	$t^0 = t$	$x^0 = y$	$y^0 = x$	$u^0 = v$	$v^0 = u$	$\omega^0 = -\omega$

To identify the governing equation from data, 100 time-step snapshots are prepared and sparsely sampled. Fig. 12 shows a typical snapshot and the sampling region. In the sampling region, 4,000 data points are randomly sampled. For each data point, there are six variables: a specific time step (t), two spatial Cartesian coordinates (x and y), and three time-varying measurements (velocity in x direction u , velocity in y direction v , and vorticity ω). These six variables are non-dimensionalized by the following characteristic variables: characteristic length l (the cylinder diameter), characteristic velocity V (inlet flow velocity), characteristic time $t_{ref} = \frac{V}{l}$, and characteristic vorticity $\omega_{ref} = \frac{l}{V}$.

For each simulation case, we flip the original data along $y = x$ to obtain symmetrically transformed data. The original and transformed variables are shown in Table 7.

6.1.2 Symmetric invariant SINDy

After augmenting the dataset, we conduct symmetric invariant SINDy for each case in Table 6 to identify the corresponding temporary governing equations. Fig. 13 shows a schematic of concatenating the sparse regression terms from the original and transformed data.

There are several implicit benefits for sparse regression because of this concatenation. First, terms with the same absolute value in the original and transformed data must have the same coefficients, indicating that the regression coefficients are physically constrained. For example, the coefficients for $\frac{\partial \omega}{\partial x}$ (the fourth column) and $\frac{\partial \omega}{\partial y}$ (the fifth column) must be the same ($\xi_5 = \xi_6$), and the coefficients for u (the first column) and v (the second column) must be zero ($\xi_1 = \xi_2 = 0$). Second, the learnable

coefficients can be reduced because of the physical constraint, which reduces the solution space and the required number of training data accordingly. Third, data augmentation is one of the important regularization techniques to avoid overfitting [11]. By leveraging symmetric invariance, the training data is doubled, which significantly improves the learning accuracy for the regression coefficients.

In this case, we include 29 general terms in the library as the candidate terms:

$$\frac{\partial \omega}{\partial t} = [u, v, \omega, \frac{\partial \omega}{\partial x}, \frac{\partial \omega}{\partial y}, \frac{\partial^2 \omega}{\partial x^2}, \frac{\partial^2 \omega}{\partial y^2}, \frac{\partial^2 \omega}{\partial x \partial y}, uu, uv, u\omega, u \frac{\partial \omega}{\partial x}, u \frac{\partial \omega}{\partial y}, u \frac{\partial^2 \omega}{\partial x^2}, u \frac{\partial^2 \omega}{\partial y^2}, u \frac{\partial^2 \omega}{\partial x \partial y}, vv, v\omega, v \frac{\partial \omega}{\partial x}, v \frac{\partial \omega}{\partial y}, v \frac{\partial^2 \omega}{\partial x^2}, v \frac{\partial^2 \omega}{\partial y^2}, v \frac{\partial^2 \omega}{\partial x \partial y}, \omega\omega, \omega \frac{\partial \omega}{\partial x}, \omega \frac{\partial \omega}{\partial y}, \omega \frac{\partial^2 \omega}{\partial x^2}, \omega \frac{\partial^2 \omega}{\partial y^2}, \omega \frac{\partial^2 \omega}{\partial x \partial y}] \xi, \quad (27)$$

where the dimension of ξ is 29×1 and the derivative terms are calculated by a local polynomial interpolation approach [12, 13]. A second-order polynomial is fitted to each data point using eight neighboring points, and the derivatives can be calculated by the fitted polynomial.

Figure 13: Concatenating the regression library for symmetric invariant SINDy from the original and transformed data. Columns with the same color have the same absolute value.

Table 8: Dataset including computed non-zero regression coefficients and parameters for the vorticity simulation.

Parameter Unit	l cm	V cm/s	μ kg/cm/s	ρ g/cm ³	p_0 Pa	Re 1	$\xi_{12} = \xi_{19}$ 1	$\xi_6 = \xi_7$ 1
Case 1	0.0850	68.00	0.1156	0.0010	1.30×10^3	50	-0.9925	0.0212
Case 2	0.1000	90.00	0.1100	0.0011	8.42×10^4	90	-0.9909	0.0126
Case 3	0.1000	100.00	0.1100	0.0011	8.53×10^4	100	-0.9941	0.01101

To find sparse regression coefficients ξ , we use a sequential threshold least-squares algorithm [12]. After optimization, each case will have only four non-zero regression coefficients. Thus, we are able to construct a non-zero coefficient dataset in combination with the parameters (and their values) listed in Table 6 for the three different cases, as shown in Table 8.

From Table 8, we notice that $\xi_{12} = \xi_{19}$ is very close to a constant value (-1), while $\xi_6 = \xi_7$ varies with different cases. Therefore, the temporary governing equation can be simplified as:

$$\frac{\partial \omega}{\partial t} = -u \frac{\partial \omega}{\partial x} - v \frac{\partial \omega}{\partial y} + \xi_6 \left(\frac{\partial^2 \omega}{\partial x^2} + \frac{\partial^2 \omega}{\partial y^2} \right) \quad (28)$$

6.1.3 Dimensionless learning

To obtain a consistent governing equation, we apply dimensionless learning to identify the expression of ξ_6 and ξ_7 . The causal relationship can be expressed as:

$$\xi_6 = \xi_7 = f(\mu, l, V, \rho, p_0). \quad (29)$$

The dimension matrix \mathbf{D} for Eqns. 29 is:

$$\begin{bmatrix} -1 & 1 & 1 & -3 & -1 \\ -1 & 0 & -1 & 0 & -2 \\ 1 & 0 & 0 & 1 & 1 \end{bmatrix} \quad (30)$$

The columns of the matrix, from right to left, represent the dimensions of μ, l, V, ρ , and p_0 respectively. The rows, from top to bottom, represent the dimensions of length, time, and mass respectively.

Unlike Sections 2.1 and 2.2 of the main manuscript, where a polynomial is used to fit the relationship between dimensionless numbers, we can simplify the fitting function $f(\cdot)$ in Eqns. 29 as a power law with a constant coefficient β . Then, the Eqns. 29 can be simplified as:

$$\xi_6 = \beta \mu^{w_1} l^{w_2} V^{w_3} \rho^{w_4} p_0^{w_5}, \quad (31)$$

where $w = [w_1, w_2, w_3, w_4, w_5]^T$ denotes the powers that generate the dimensionless number and are to be determined.

The powers w need to satisfy the following linear system of equations:

$$\mathbf{D}w = 0. \quad (32)$$

Furthermore, the solutions of the linear system (Eqns. 32) can be written as a linear combination of two basis vectors w_{b1} and w_{b2} :

$$w = \gamma_1 w_{b1} + \gamma_2 w_{b2}, \quad (33)$$

where $w_{b1} = [-1, 1, 1, 1, 0]^T$ and $w_{b2} = [-1, 1, -1, 0, 1]^T$.

To summarize, there are three unknown parameters in the identification of ξ_6 and ξ_7 : two basis coefficients (γ_1 and γ_2) and a constant coefficient β . The training data come from the non-zero regression coefficient dataset in Table 8. Applying a pattern search-based optimization method with a grid search range of $[-2, 2]$ and a grid interval of 0.5, the optimized expressions for ξ_6 and ξ_7 can be written as:

$$\xi_6 = \xi_7 = 1.083 \frac{\mu}{\rho V l} \approx \frac{\mu}{\rho V l} \quad (34)$$

This form is similar to the reciprocal of the classical Reynolds number, indicating that the integration of dimensionless learning can directly identify the expression of sparse regression coefficients. By substituting the expression for ξ_6 and ξ_7 (Eqns. 34) into Eqns. 28, we obtain a consistent dimensionless governing equation for all cases:

$$\frac{\partial \omega}{\partial t} = -u \frac{\partial \omega}{\partial x} - v \frac{\partial \omega}{\partial y} + \frac{1}{\text{Re}} \left(\frac{\partial^2 \omega}{\partial x^2} + \frac{\partial^2 \omega}{\partial y^2} \right), \quad (35)$$

which is identical to the well-known Navier-Stokes equation in a vorticity form.

Furthermore, we test the noisy data effect on this problem. After adding 1% Gaussian noise to the measurements, the identified regression coefficients are: (1) case 1, $\xi_{12} = \xi_{19} = -0.9994$, $\xi_6 = \xi_7 = 0.0206$; (2) case 2, $\xi_{12} = \xi_{19} = -0.9909$, $\xi_6 = \xi_7 = 0.0126$; and (3) case 3, $\xi_{12} = \xi_{19} = -0.9941$, $\xi_6 = \xi_7 = 0.0111$. Applying dimensionless learning, both ξ_6 and ξ_7 are identified as $1.0581 \frac{1}{\text{Re}}$, which is close to the reciprocal of the Reynolds number. Therefore, even with the noisy data effect, the proposed method can identify the governing equation.

6.2 Navier-Stokes equation

In this section, we demonstrate that the dimensionless learning in conjunction with symmetric invariant SINDy can be used to identify the Navier-Stokes equation in a Cartesian coordinate system. We performed three series of simulations for the Kármán vortex street problem with three circular cylinders to prepare the original data using Flow3D. The simulation parameters with their values for different cases are shown in Table 10. The time-varying measurements include a specific time step (t), two spatial Cartesian coordinates (x and y), and three time-varying measurements (fluid velocity in x direction u , velocity in y direction v , and pressure p).

Following the same procedures described in Section 6.1, the regression model is defined as:

$$\frac{\partial u^*}{\partial t} = [u^*, uu^*, vu^*, \frac{\partial u^*}{\partial x}, \frac{\partial u^*}{\partial y}, \frac{\partial^2 u^*}{\partial x^2}, \frac{\partial^2 u^*}{\partial y^2}, u \frac{\partial u^*}{\partial x}, u \frac{\partial u^*}{\partial y}, u \frac{\partial^2 u^*}{\partial x^2}, u \frac{\partial^2 u^*}{\partial y^2}, v \frac{\partial u^*}{\partial x}, v \frac{\partial u^*}{\partial y}, v \frac{\partial^2 u^*}{\partial x^2}, v \frac{\partial^2 u^*}{\partial y^2}, p^*, \frac{\partial p^*}{\partial x}] \xi, \quad (36)$$

Table 9: Simulation parameters for the fluid flow through three circular cylinders to identify the Navier-Stokes equation.

Parameter Unit	l cm	V cm/s	μ kg/cm/s	ρ g/cm ³	p_0 Pa	Re 1	l_x cm	l_y cm	N_x 1	N_y 1
Case 1	0.1000	100.00	0.1100	0.0011	8.53×10^5	100	0.9	0.4	500	222
Case 2	0.0950	165.43	0.1040	0.0011	8.53×10^5	170	0.9	0.4	500	222
Case 3	0.1050	170.43	0.1020	0.0011	9.13×10^5	200	0.9	0.4	500	222

Table 10: Identified regression coefficients accompanied by several ground truth dimensionless numbers for the Navier-Stokes problem (clean data).

Case ID	$\xi_8 = \xi_{13}$	$\xi_6 = \xi_7$	ξ_{17}	Re	1/Re	Eu
Case 1	-0.9759	0.0105	-14127.23	100	0.0100	14545.45
Case 2	-0.9814	0.0063	-3850.28	170	0.0059	3897.63
Case 3	-0.9820	0.0059	-2977.75	200	0.0050	3019.97

where ξ is a vector with 17 terms, u^* is the combination of u and v , v^* is the combination of v and u , and p^* is the combination of p and p . It is worth mentioning that u^* , v^* , and p^* are designed for the flipping operation so that we can concatenate the regression libraries for the original and transformed data and train them together. This operation helps to incorporate symmetric invariance into the learning scheme. After the transformation, $u^{*'}$, and $v^{*'}$ become v^* and u^* , respectively, while $p^{*'}$ remains the same as p^* .

Following the same procedure as Section 6.1, we obtain the optimized sparse regression coefficients. There are only five terms with non-zero regression coefficients, as shown in Table 10.

Because $\xi_8 = \xi_{13}$ are close to -1, it is reasonable to assign -1 to both ξ_8 and ξ_{13} . Next, we apply dimensionless learning to identify the expression for the other three non-zero coefficients. Using the same pattern search optimization described in Section 6.1, the identified ξ_6 and ξ_7 are $1.0746 \frac{\mu}{\rho V l}$, which can be simplified to $1/\text{Re}$. The ξ_{17} is identified as $-0.9729 \frac{p_0}{\rho V^2}$ and can be simplified as the negative of a well-known dimensionless number, Euler number, Eu . By substituting all non-zero regression coefficients into Eqns. 36, we obtain a consistent governing equation:

$$\frac{\partial u^*}{\partial t} = -u \frac{\partial u^*}{\partial x} - v \frac{\partial u^*}{\partial y} - \text{Eu} \frac{\partial p^*}{\partial x} + \frac{1}{\text{Re}} \left(\frac{\partial^2 u^*}{\partial x^2} + \frac{\partial^2 u^*}{\partial y^2} \right), \quad (37)$$

which is identical to the Navier-Stokes momentum equation written for the combination of x and y directions.

We also test the noisy data effect on this problem by imposing 0.5% Gaussian noise to the measurements. The identified non-zero coefficients are shown in Table 11.

Using dimensionless learning, we can identify the expression for ξ_6 and ξ_7 as $1.0473 \frac{\mu}{\rho V l}$. Similarly, the expression for ξ_{17} is identified as -0.9687Eu . Compared to the clean data results in Table 10, ξ_8 and ξ_{13} have higher errors. By assuming the ξ_8 and ξ_{13} as $-\text{Eu}$, we obtain the same equation as Eqns. 37.

Table 11: Identified regression coefficients accompanied by several ground truth dimensionless numbers for the Navier-Stokes problem (noisy data).

Case ID	$\xi_8 = \xi_{13}$	$\xi_6 = \xi_7$	ξ_{17}	Re	1/Re	Eu
Case 1	-0.9668	0.0104	-14139.49	100	0.0100	14545.45
Case 2	-0.9378	0.0063	-3699.04	170	0.0059	3897.63
Case 3	-0.9217	0.0053	-2785.92	200	0.0050	3019.97

Table 12: Identified regression coefficients accompanied by several ground truth dimensionless numbers for the Euler equation example (clean data).

Case ID	$\xi_8 = \xi_{13}$	ξ_{17}	Re	Eu
Case 1	-1.0436	-14127.23	100	14545.45
Case 2	-1.0016	-3896.14	170	3897.63
Case 3	-1.0177	-3004.48	200	3019.97

Table 13: Identified regression coefficients accompanied by several ground truth dimensionless numbers for the Euler equation example (noisy data).

Case ID	$\xi_8 = \xi_{13}$	ξ_{17}	Re	Eu
Case 1	-1.0361	-14015.25	100	14545.45
Case 2	-0.9946	-3873.86	170	3897.63
Case 3	-1.0088	-2953.95	200	3019.97

It is worth mentioning that we choose a different sampling region for the same data in this problem, but all other procedures remain the same as in the previous example. The identified non-zero regression coefficients for clean and noisy data (1% Gaussian noise) are shown in Table 12 and Table 13.

From Table 12 and Table 13, we can see that ξ_8 and ξ_{13} are very close to -1, and ξ_{17} is the only varying regression coefficient. To identify the expression of ξ_{17} for clean and noisy data, we use dimensionless learning and identify the expressions are $-0.9671Eu$ and $-0.9661Eu$, respectively. By simplifying ξ_{17} to $-Eu$ and substituting all non-zero regression coefficients into Eqns. 36, we obtain the governing equation as below:

$$\frac{\partial u^*}{\partial t} = -u \frac{\partial u}{\partial x} - v \frac{\partial u}{\partial y} - Eu \frac{\partial p^*}{\partial x} \quad (38)$$

which is identical to the Euler equation. The reason for simplifying the Navier-Stokes equation to the Euler equation is that the viscous effects of the fluid far from the cylinders and walls are negligible in this problem.

[More demonstrated problems can be found in Sections 6.3 and 6.4 can be found in the SI, including spring-mass-damper problems and dynamic response of impulsively loaded plastic beam problems.]

As it concerns the proposed approaches, the grid search one used to attain the results proposed in the main manuscript is never formalized. This makes it difficult to grasp the details of the latter and, thus, understand its benefits and flows. In addition, if the proposed method is a naive grid search, the approach seems to me a rather simplistic one to detect the coefficients of the basis combinations, possibly leading to poor results when the mesh of the grid over which the search is performed is not dense enough. I was thus wondering if the authors can comment on this aspect, possibly providing a detailed explanation of their design choices in the supplementary material.

We applied naïve grid search to select the best dimensionless numbers and scaling laws in Sections 2.1 and 2.2 of the main text because there are only three basis coefficients and one input dimensionless number. During the optimization, we further kept one basis coefficient as a constant to avoid obtaining the same dimensionless number with different power exponents. Therefore, only two

basis coefficients and the fitting function are to be determined and the grid interval can be relatively small, such as 0.01. However, when there are multiple input dimensionless numbers or the number of basis coefficients is large, a large grid interval such as 0.5 is necessary to speed up the optimization. Also, the other two proposed two-level optimization methods (pattern search and gradient descent approach) in this paper are encouraged to be used to avoid the exponential growth of basis combinations. We added a subsection to Section 3.1 of the SI to demonstrate the details of grid search.

[Section 3.1 of the SI] For dimensionless learning, the key point is to identify the basis coefficients $\boldsymbol{\gamma}$ and polynomial coefficients $\boldsymbol{\beta}$. If a problem only involves a few basis coefficients (less than three) and a few input dimensionless numbers (less than two), grid search can work well to find the best coefficients, which is described in Algorithm 1. In this scenario, the grid interval can be relatively small such as 0.01. For example, there are three basis coefficients and one input dimensionless number for the turbulent Rayleigh-Bénard convection (Section 2.1 in the main manuscript) and the vapor depression dynamics in laser-metal interaction (Section 2.2 in the main manuscript). A fine grid interval of 0.01 is used in both cases. However, when there are multiple input dimensionless numbers or the number of basis coefficients is large, a large grid interval such as 0.5 is necessary to speed up the optimization. Also, the other two proposed two-level optimization methods in this paper are encouraged to be used to avoid the exponential growth of basis combinations.

Algorithm 1 Grid search-based two-level optimization for dimensionless learning

Require: Input variables \boldsymbol{p} , Dimension matrix \boldsymbol{D} , all possible sparse output dimensionless numbers $\{\Pi\}^H$, basis vectors $\{\boldsymbol{w}_{bi}\}^K$, the number of input dimensionless number M , the polynomial function degree Q , the grid interval d .

for $h \leftarrow 1$ to H **do** ▷ Loop for all possible output dimensionless numbers.

for $i \leftarrow 1$ to M **do** ▷ Loop for all input dimensionless numbers.

$\gamma_{i1} \leftarrow 1$.

 Divide $[-2, 2]$ uniformly with a grid interval d for $\gamma_{i2}, \gamma_{i3}, \dots, \gamma_{iK}$.

end for

 Loop for all the basis combination for M dimensionless numbers.

 For each combination, construct the input dimensionless numbers $\Pi_1, \Pi_2, \dots, \Pi_M$.

 Fit $\Pi_h = f(\Pi_1, \Pi_2, \dots, \Pi_M)$ with a polynomial with degree Q .

 Measure the fitting performance with R^2 score.

end for

Choose the best scaling law $\Pi_h = f(\Pi_1, \Pi_2, \dots, \Pi_M)$ with a high R^2 score and a low L1 norm of dimensionless numbers.

Moreover, several thresholds and parameters (e.g., the gradient descent step) have to be selected at different stages of the procedure. Nonetheless, their choice is never explained, insights on how to select these parameters are chosen are never provided nor a sensitivity analysis is performed to explain how the latter should be selected. I believe this severely undermines the possibility to exploit the approach in other applications. I thus suggest the authors to at least provide some guidelines on their choices. I would also like to point out that, to the best I could detect, several threshold and parameters needed within the different routines are never mentioned, thus making it impossible to replicate the results presented in the paper. Similar consideration holds for the exploration spaces and the density of the grid reported in Algorithms 2 and 3 (see the Supplementary material), whose generality is somewhat obscure.

We revised the paper to explain this in detail in a newly added section 3.4 of the SI and added some descriptions to the captions of Figures 4 to 6 in SI. This section describes the hyperparameters for different optimization methods. We also made the code and data available on GitHub (<https://github.com/xiaoyuxie-vico/PyDimension>) so that readers could replicate this work.

[Section 3.4 of the SI] Section 3.4 Hyperparameter setting

The hyperparameter settings have a significant impact on the model performance. The following is a summary of the hyperparameter selection:

1. Grid search-based two-level optimization:

- a. The exploration range is set to $[-2, 2]$ because most dimensionless numbers have a small power law index;
 - b. The grid interval is set to 0.01 if there are only a few input dimensionless numbers and basis coefficients. Otherwise, the interval is set as 0.5.
 - c. For each input dimensionless number, the first basis coefficient is set as a constant, such as 0.5 or 1, and so on.
2. Gradient-based two-level optimization:
- a. For each input dimensionless number, the first basis coefficient is set as a constant as the grid search, while the other basis coefficients are random floats in the range $[-2, 2]$;
 - b. Because the BFGS optimization algorithm is used, there is no need to manually select a learning rate or gradient descent step. It is a second-order optimization algorithm that uses line search algorithms to determine the best learning rate for each iteration.
3. Pattern search-based two-level optimization:
- a. For each input dimensionless number, the first basis coefficient is set as a constant as the grid search;
 - b. The exploration range is set to $[-2, 2]$ because most dimensionless numbers have a small power exponent;
 - c. The grid interval can be set to 0.01, 0.1, 0.25, or 0.5, depending on the number of input dimensionless numbers and basis coefficients. Smaller exploration spaces are more flexible in selecting a grid interval, while we suggest using grid intervals of 0.5 for larger exploration spaces.
4. Other important hyperparameters:
- a. The L1 norm threshold for output dimensionless numbers is set to 5 (no more than 5 is acceptable).
 - b. The L1 norm threshold for input dimensionless numbers varies depending on the problems. The thresholds are 8, 9, and 13, respectively, for the problems in Sections 2.1 to 2.3 in the main manuscript.
 - c. To avoid being trapped by local minima in gradient-based and pattern search-based approaches, the number of initial points is set to 1,000. The best model is selected by the fitting performance (R^2 score) and the L1 norm of dimensionless numbers.

[The captions of Figures 4 in SI] The grid range is $[-2, 2]$, but only the range in $[0, 2]$ is shown in the figure because this area has a higher R^2 score.

[The captions of Figures 5 in SI] The grid range is $[-2, 2]$ and the interval is 0.05.

[The captions of Figures 6 in SI] The grid range is $[-2, 2]$ and the interval is 0.05, but only the range in $[0, 2]$ is shown in the figure because this area has a higher R^2 score.

Still concerning the grid search-based approach, I was wondering if the authors can comment on how it scales with respect to the number of coefficients to be learned and compared with the other two strategies they propose. Indeed, I expect that the grid search approach becomes rather computationally demanding when the dimensions increase. Since three possible techniques are presented, I think it would be important to better highlight their differences and advantages/disadvantages over the others.

We agree with the reviewer's assessment of the grid search-based optimization approach. The computational cost can rise as the number of basis coefficients and dimensionless numbers increases. However, it is reasonable to assume that most complex systems are dominated by a few dimensionless numbers. In the complex problems demonstrated in Sections 2.1 and 2.2 of the main manuscript, there is only one input dimensionless number. Even with a grid interval of 0.01, we can efficiently identify the best scaling law and dimensionless numbers. We described the advantages and disadvantages of different optimization methods in Section 3.5 of the SI.

[Section 3.5 of the SI] In this section, we summarize and highlight the differences between the three proposed optimization methods to clearly demonstrate their advantages and disadvantages. The detailed comparison is shown in Table 1. It is worth noting that because we only want simple expressions of dimensionless numbers, zero-order optimization, such as the grid search-based approach or pattern search-based approach, is more suitable for finding the optimal basis coefficients, i.e., dimensionless numbers.

Table 1: **Comparison of three proposed optimization methods.**

Optimization method	Advantages	Disadvantages
Grid search	 a. Zero-order optimization. No need to compute gradients; b. Easy to implement; c. Applicable in low dimension problem; d. Flexible in the choice of grid interval; 	 a. Computational cost grows exponentially with increasing dimensionality; b. Need model selection after grid search;
Gradient descent	 a. More capable of overcoming local minima; b. No need to manually select learning rate; 	 a. The optimized basis coefficients may not be integers or multiples of 0.5; b. Need more iterations to find the optimal solution;
Pattern search	 a. Zero-order optimization. No need to compute gradients; b. Less iterations compared to gradient descent-based method; c. Flexible in the choice of step size (grid interval); d. Applicable for both low and high dimension problems; 	 a. Easy to get stuck in local minima; b. Need more randomly initial points; c. Computationally expensive for high dimensional problems;

Additionally, it is known that alternated approaches like the gradient based and pattern search based ones presented in the Supplementary material can get trapped in local minima and that the quality of the resulting solution is heavily dependent on the chosen initial condition. Are the authors testing several initial conditions and then picking the best result or are they using any other strategy to cope with this phenomenon? If this is not the case, can the authors comment on this?

As the reviewer pointed out, the initial points for basis coefficients can influence the gradient-based and pattern search-based methods. Thus, we randomly sampled 1,000 initial points and chose the best results as the final optimal solution for the three problems from Sections 2.1 to 2.3 of the main manuscript. The model selection is based on its fitting performance (R^2 score) and the L1 norm of dimensionless numbers. We include descriptions in Section 3.4 of the SI.

[The last point of Section 3.4 of the SI] To avoid being trapped by local minima in gradient-based and pattern search-based approach, the number of initial points is set as 1,000. The best model is selected by the fitting performance (R^2 score) and the L1 norm of dimensionless numbers.

In the following please find some additional comments on both the manuscript and the supplementary material. All the references and the equation numbering corresponds to those in the paper.

On the manuscript

- I believe that the limitations of the approach proposed in [14] should be better highlighted and its

performance should be eventually compared with the one of the proposed approaches at least for one of the considered case study, to support the validity of the approach proposed by the authors.

Constantine, Rosario, and Iaccarino proposed a rigorous mathematical framework to estimate unique and relevant dimensionless groups. However, their method has at least two limitations. First, to use their method, the independent input variables should yield gaussian distributions and dependent input variables should yield a known probability density function. This constraint limits the wide applications of their method because many experimental datasets in the literature (including the ones used in our study) do not have gaussian input variables. Thus, their method cannot be used to analyze the datasets used in the manuscript. Second, noises or errors in the input and output are negligible in their method.

In contrast, we have demonstrated that our method could find dimensionless numbers and governing laws (scaling laws and differential equations) in noisy data. The analyses are firstly performed on three challenging engineering problems with noisy experimental measurements to discover dimensionless numbers and scaling laws. The problems include turbulent Rayleigh-Benard convection, vapor depression dynamics, and porosity formation during 3D printing. Then, we performed the analyses on five differential equations, including Navier-Stokes (0.5% noise), Euler (1%), vorticity (1%), the governing equation for spring-mass-damper systems (4%), and dynamic loading beam systems (2%). Even with the noisy data effect, the method successfully discovers the correct governing equations, as demonstrated in Section 2.4 of the main manuscript and Section 6 of the SI.

- I was wondering what is the reason why you fix γ_1 to 0.5 (see line 199). Can you explain this choice and eventually link it to the computational complexity of the grid search method, if connected to you decision of fixing one parameter beforehand?

There are two reasons to set the first basis coefficient γ_1 a constant value such as 0.5 or 1. First, a constant γ_1 helps to avoid the identification of equivalent dimensionless numbers. For example, assume that the best basis coefficients are $\{\gamma_i\}_{i=1}^K$ and that the corresponding dimensionless number is Π . Basis coefficients $\{C\gamma_i\}_{i=1}^K$ can also be used to formalize a dimensionless number Π^C , where C is a constant. By changing the fitting function, it could also achieve a high R^2 score. Therefore, we need to specify a constant for the first basis coefficient γ_1 . Specifically, we can set γ_1 to 0.5 or 1 to achieve a small $L1$ norm of a dimensionless number. Other numbers to consider for γ_1 include -0.5 and -1. Second, by keeping one basis coefficient constant, all optimization methods can be made less computationally complex. We expanded on this in the second paragraph of Section 3.4 of the SI. We also briefly mentioned this in the third-to-last paragraph of Section 2.1 of the main manuscript.

[The second paragraph of Section 3.4 of the SI] There are two reasons to set the first basis coefficient γ_1 a constant value such as 0.5 or 1. First, a constant γ_1 helps to avoid the identification of equivalent dimensionless numbers. For example, assume that the best basis coefficients are $\{\gamma_i\}_{i=1}^K$ and that the corresponding dimensionless number is Π . Basis coefficients $\{C\gamma_i\}_{i=1}^K$ can also be used to formalize a dimensionless number Π^C , where C is a constant. By changing the fitting function, it could also achieve a high R^2 score. Therefore, we need to specify a constant for the first basis coefficient γ_1 . Specifically, we can set γ_1 to 0.5 or 1 to achieve a small $L1$ norm of a dimensionless number. Other numbers to consider for γ_1 include -0.5 and -1. Second, by keeping one basis coefficient constant, all optimization methods can be made less computationally complex.

[The third-to-last paragraph of Section 2.1 of the main manuscript] We set γ_1 to one to avoid the identification of equivalent dimensionless numbers with different powers and reduce the computational cost.

- Can you please clarify how you shift from the values of γ_2 and γ_3 presented at line 202 and the ones

reported in line 206? Are you considering a different point of the grid, increasing its density or changing some parameter of the grid search algorithm?

In Section 2.2 of the main manuscript, the first basis coefficient γ_1 is set to 0.5, and the grid search exploration range is $[-2, 2]$ with a 0.01 grid interval. Figure 2 of the main manuscript shows the R^2 scores for different grid points. The dimensionless number shown in the previous line 202 is a local optimum in Figure 2.c, whereas the dimensionless number shown in the previous line 206 is a global optimum in Figure 2.b. We do not change the density or any other parameters to obtain them. The x-axis label in Figure 2.c, we believe, is causing the confusion. We rearranged the expression for that x-axis label to be consistent with the dimensionless number mentioned in the third paragraph of Section 2.2 of the main manuscript.

[Figure 2 of the main manuscript] The x-axis label of Figure 2.c is modified.

- I would like the authors to clarify their comment at line 285. Indeed, how can one be sure to have considered all possible candidates for the dominant dynamics?

Building a regression library with all the required candidates relies on the researchers' experience and understanding of the problem. If some terms are known to be important to the problems, more specific terms can be included, limiting the regression library to a smaller one. For example, convective and diffusion terms are frequently used in the governing equations of fluid mechanics. It is reasonable to include these terms in the library.

In addition, if there is some prior knowledge about the relationship between different terms, the number of learnable coefficients can be reduced. For example, for a 2-dimensional fluid mechanics problem, we can assume that the two convective and diffusion terms have the same coefficients. It is

worth noting that the proposed symmetric invariance does not explicitly include this physical constraint but achieves it implicitly.

In a more general and unknown problem, we suggest including linear terms first, such as the original measurements and their first and second order derivatives. If the fitting performance is ideal, we can stop involving more terms and get the best differential equations. Otherwise, we will keep adding quadratic and other high-order terms and testing the fitting performance. The library will be updated until the best performance model is discovered. We added these descriptions in Section 6.5 of the SI.

[Section 6.5 of the SI]

Section 6.5 Regression library suggestions

Building a regression library with all the required candidates relies on the researchers' experience and understanding of the problem. If some terms are known to be important to the problems, more specific terms can be included, limiting the regression library to a smaller one. For example, convective and diffusion terms are frequently used in the governing equations of fluid mechanics. It is reasonable to include these terms in the library.

In addition, if there is some prior knowledge about the relationship between different terms, the number of learnable coefficients can be reduced. For example, for a 2-dimensional fluid mechanics problem, we can assume that the two convective and diffusion terms have the same coefficients. It is worth noting that the proposed symmetric invariance does not explicitly include this physical constraint but achieves it implicitly.

In a more general and unknown problem, we suggest including linear terms first, such as the original measurements and their first and second order derivatives. If the fitting performance is ideal, we can stop involving more terms and get the best differential equations. Otherwise, we will keep adding quadratic and other high-order terms and testing the fitting performance. The library will be updated until the best performance model is discovered. We added these descriptions in Section 6.5 of the SI.

- In the discussion, the authors mention several Neural Networks where some sort of invariance is enforced in the network structure. I do understand that these are introduced to highlight the importance of embedding invariance within learning but, at the same time, I do not truly see the relevance of this in the discussion. I thus suggest the authors to at least rethink the location of these comments.

To make the discussion more concise, we removed the previous descriptions of invariance in several Neural Networks in the Discussion Section of the main manuscript.

- Can you please further comment on the applications you have in mind that can benefit from your approach for what concerns low control of precise drug delivery and financial market analysis? Alternatively, can you cite some sources where they highlight some requirements for these applications that are aligned with the features of the proposed method?

We added a fluid dynamics example to demonstrate that the proposed method can be used to discover dimensionally consistent Navier-Stokes equations, which are governing equations in microfluidic flow of drug delivery. Thus, our method can be potentially applied to this area to identify effective models for flow control. We added a reference ([50]) to support this claim.

It might be too aggressive to claim that our method can be used for financial market analysis, thus we remove that in the revised manuscript.

[Add reference in the main manuscript]

[50] Patel, Mukund R and Beik, Omid. Wind and solar power systems: design, analysis, and operation. CRC press, 2021

On the supplementary material

- I advice the authors to introduce the NEP index (used in the Manuscript) when discussing the results on porosity formation in the Supplementary Material.

Thanks for the advice. We modified Table 2 of the SI using the NEP index.

[Table 2 of the SI]

Table 2: **Identifying more dimensionless numbers for the porosity formation problem.** The first two columns are input dimensionless numbers, respectively. The third and fourth columns show the corresponding model performance in the training and test sets, respectively.

Π_1	Π_2	R^2 score (Training set)	R^2 score (Test set)
NED	NEP	0.9728	0.7581
NED	$NEP \frac{H}{d}$	0.9675	0.7025
$NED \frac{d^3}{L^3}$	$NEP \frac{L}{d}$	0.9561	0.9512
$NED \frac{d^2}{HL}$	$NEP \frac{HL\sqrt{C_p(T_1-T_0)}}{V_s d^2}$	0.9596	0.9380

- In Algorithm 1, I believe there is an exceeding squared bracket in $abs(\min([W$

It was a typo in the previous Algorithm 1 for calculating basis vectors. We replaced the pseudocode with a description and simple python code to clearly demonstrate the method. The revision is shown in the third paragraph of Section 1.2 of the SI.

[The third paragraph of Section 1.2 of the SI] To identify the basis vectors $\{\mathbf{w}_{bi}\}^K$ for $\mathbf{D}\mathbf{w} = \mathbf{D} \sum_{i=1}^K \gamma_i \mathbf{w}_{bi} = 0$, we first need to rearrange the dimension matrix \mathbf{D} in reduced row echelon form using Gaussian elimination. Then, we set each free variable to 1 and all others to 0 and substitute them to solve the system and obtain a basis vector. The number of free variables is equal to K . We can use the Python package "sympy" to implement this method:

```
[python code to calculate basis vectors]
from sympy import Matrix
D = Matrix(D)
Basis_vectors = D.nullspace()
```

- Can you please provide additional details on the degree of exploration of the parameter space you need to obtain satisfactory results with your approaches? Do you need specific excitation condition/data informativity conditions to be fulfilled for the approach to properly work? How does the latter compare with the one required by other approaches?

The proposed method needs an excitation condition for the first basis coefficient to ensure that the identified dimensionless numbers are unique. Generally, we can set the first basis coefficient γ_1 a constant, such as 0.5 or 1, etc. There are several reasons for this restriction. First, a constant γ_1 aids

in avoiding the identification of equivalent dimensionless numbers. For example, suppose that the best basis coefficients are $\{\gamma_i\}_{i=1}^K$ and that the corresponding dimensionless number is Π . The basis coefficients $\{C\gamma_i\}_{i=1}^K$ can also be used to express a dimensionless number Π^C , where C is a constant. By changing the fitting function, it could also achieve a high R^2 score. Therefore, we need to specify a constant for the first basis coefficient γ_1 . Second, by keeping one basis coefficient constant, all optimization methods can be made less computationally complex. We expanded on this in the second paragraph of Section 3.4 of the SI.

[The second paragraph of Section 3.4 of the SI]

- I was wondering whether the procedure reported in the flow chart in Figure 1 is performed in cross-validation or by using the training and tests sets only. If the latter is used in this second configuration, what are the chances to overfit the available training and validation data? Can the authors discussed this?

We agree that cross-validation can help determine some hyperparameters and avoid overfitting. For example, it can be used to determine the polynomial degree of the second-level optimization. We added a description of this in Section 1.3 of the SI and used a 5-fold cross-validation in the vapor depression dynamics problem in Sections 3.4 and 5.3 of the SI.

[The second paragraph of Section 1.3 of the SI] It is worth mentioning that, while cross-validation is not shown in Fig. 1, we recommend using it to tune the hyperparameters for the fitting function. For example, cross-validation can be used to determine the polynomial degree. There are also examples applying 5-fold cross-validation in Sections 3.4 and 5.3 of the SI.

[Sections 3.4 of the SI] Generally, it is hard to determine some hyperparameters before the training, such as the order of polynomial degree. We encourage readers to first choose a second-level fitting function with a high capacity, such as a five-order polynomial. After identifying some scaling laws with high performance, we can reduce the model capacity and select the best parsimonious model based on the L1 norm of the dimensionless numbers and the R^2 score of the scaling law. However, if there is some prior knowledge of the systems, we can choose a proper fitting function in advance.}

It is also worth mentioning that cross-validation is an effective technique to determine hyperparameters. We demonstrate 5-fold cross-validation in determining polynomial degree in the vapor depression dynamics problem in laser-material interaction, which is shown in Fig. 7. The R^2 score of a polynomial function with a degree of three is 0.9702, which is only slightly lower than that of a polynomial function with a degree of four (0.9708). To obtain a simple scaling law without degrading performance too much, we set the polynomial degree to three.

Figure 7. Polynomial degree identification for the keyhole dynamics problem using 5-fold cross-validation.

[Section 5.3 of the SI]

5.3 Comparison of different machine learning algorithms with dimensionless learning

The out-of-bag error is a critical metric for machine learning algorithms, especially when dealing with problems with limited data. In this section, we compare the performance of five popular machine learning algorithms with our proposed method over two small datasets. The proposed method shows the best generalization in datasets with different materials and scales.

5.3.3 Model generalization for different materials

The first dataset is the same as the one used in Section 2.2 of the manuscript. This dataset contains three different materials: titanium alloy (Ti6Al4V), aluminum alloy (Al6061), and stainless steel (SS316). Here, we do not shuffle all the data and split it into training and test sets. We only use Ti6Al4V and Al6061 data in the training set, while other data (SS316) is used in the test set. 5-fold cross-validation is used to select the best parameters of each model.

Fig. 9 shows the R^2 score for each method. Even though the Feed Forward Neural Network (FFNN) achieves very good performance on the training and validation sets, the R^2 score for the test set is below -0.1 . Random Forest (RF) has relatively high R^2 scores for training and validation, but it still performs poorly on the test set. On the other hand, the proposed method has the highest R^2 score on the test set, indicating that it has the best generalization in different materials. This improvement is the result of ensuring geometric, kinematic, and dynamic similarities based on similitude theory within different systems.

Figure 10: Comparisons of six candidate models for the generalization of different materials. The training and validation sets come from two materials (titanium alloy (Ti6Al4V) and aluminum alloy (Al6061)), while the test set comes from the other material (stainless steel (SS316)). The highest the R^2 score, the better the model. The y-axis only shows R^2 scores above -0.1 . The regression methods include Feed-forward Neural Network (FFNN), Linear regression (LR), Extreme Gradient Boosting (XGBoost), K-Nearest Neighbors (KNN), Random Forest (RF), and the proposed method.

5.3.4 Model generalization for different scales

In this section, we use a rough pipe flow example to demonstrate that the proposed method has the best generalization ability in data with different scales. Pipe flow is a classic example to analyze the fluid flow from laminar to turbulent. Reynolds number (Re) is widely used in fluid mechanics to distinguish between laminar and turbulent flow. In general, several measurements can be collected in the steady state of a pipe flow system: fluid velocity v , dynamic viscosity μ , pipe diameter d , fluid density ρ , and pressure p .

Two types of synthetic data are created to build the dataset: small pipe flow data for the training and validation sets, and large pipe flow data for the test set. All the data are generated randomly. The detailed range of each parameter is shown in Table 5

Table 5: **Different parameter ranges for small and large pipe flows.**

Source	Parameter	Range	Source	Parameter	Range
Small pipe	d	0.1 - 1 cm	Large pipe	d	10 - 100 cm
	μ	$1 \times 10^{-3} - 1 \times 10^{-2}$ g/cm/s		μ	$1 \times 10^{-5} - 1 \times 10^{-3}$ g/cm/s
	v	100 - 1×10^3 cm/s		v	1 - 10 cm/s
	ρ	$1 \times 10^{-3} - 1 \times 10^{-2}$ g/cm ³		ρ	$1 \times 10^{-4} - 1 \times 10^{-3}$ g/cm ³
	p	$1 \times 10^4 - 1 \times 10^5$ Pa		p	$1 \times 10^4 - 1 \times 10^5$ Pa

Fig. 11 compares the performance of six regression models on the training, validation, and test sets. Because the test set consists entirely of large pipe flow rather than small pipe flow, all machine learning algorithms perform very poorly on the test set (below 0). On the other hand, the proposed method can perform very well on the test set. Even though this dataset is synthetic, it shows that the proposed method can be used for data with different scales.

Figure 11: Comparisons of six candidate models to identify a generalized model for data from different scales. The training and validation sets are generated using small pipe cases, while the test set is associated with large pipe cases. The higher the R^2 score, the more accurate the model. The y-axis only shows R^2 scores greater than -0.1.

- I think the introduction of a mathematical formulation for the problem solved in Section 1.3 would clarify it and highlight the parameters one needs to select at this stage, considering that a lasso regularization seems to be exploited.

We revised this section and clarified an $L1$ norm threshold used in the paper. The threshold for the case in Sections 2.1 and 2.2 of the main manuscript is 5 (no more than 5 is acceptable). The reason for this is that we assume the output dimensionless numbers should have a simple expression, despite the fact that the input dimensionless numbers can be quite complex. One typical example is the vapor depression dynamics example in laser-metal interaction. The output dimensionless number is a dimensionless length, keyhole aspect ratio, which is the keyhole depth to laser radius ratio. The $L1$ norm of this output dimensionless number is only 2, while the input dimensionless number is relatively

more complex with an L1 norm of 6.5. Section 1.3 of the SI includes the revision in the first two paragraphs.

[The first two paragraphs of Section 1.3 of the SI] Prior to training, we calculate all possible sparse output dimensionless numbers based on an L1 norm constraint for \mathbf{w}' , which is the sum of the absolute values for all elements in \mathbf{w}' . Typically, the L1 norm threshold for output dimensionless numbers is set to 5 (no more than 5 is acceptable). The motivation is that we assume the output dimensionless numbers should have a simple expression, despite the fact that the input dimensionless numbers can be relatively complex. The vapor depression dynamics problem in laser-metal interaction is a typical example. The output dimensionless number is a dimensionless length, keyhole aspect ratio, which is the keyhole depth to laser radius ratio. This output dimensionless number has an L1 norm of only 2, whereas the input dimensionless number (relatively more complex) has an L1 norm of 6.5.

Due to a low L1 norm threshold for the output, there are only a very few candidate output dimensionless numbers. For example, in the problem of keyhole dynamics (Section 2.2 of the main manuscript), there are seven candidate output dimensionless numbers, whereas in the case of turbulent Rayleigh-Bernard convection (Section 2.1 of the main manuscript), there are only two candidate output dimensionless numbers.

- In general, how do you decide the shape of the function f characterising the relationship between the input and output dimensionless parameters? Is it selected based on some priors on the physics of the considered system, by cross-validation or with other methods? By clarifying this, the approach would be easier to apply over problems other than the ones presented in the paper.

In general, some hyperparameters, such as the order of a polynomial, are difficult to predict before training. We recommend starting with a second-level fitting function with a high capacity, such as a five-order polynomial. After identifying some scaling laws with high performance, we can reduce the model capacity and select the best parsimonious model based on the L1 norm of the dimensionless numbers and the R^2 score of the scaling law. However, if we have some prior knowledge of the systems, we can select a suitable function in advance.

It is also worth mentioning that cross-validation is an effective technique to determine hyperparameters. We demonstrate 5-fold cross-validation in determining polynomial degree in the vapor depression dynamics problem in laser-material interaction, which is shown in Fig. 7. The R^2 score of a polynomial function with a degree of three is 0.9702, which is only slightly lower than that of a polynomial function with a degree of four (0.9708). To obtain a simple scaling law without degrading performance too much, we set the polynomial degree to three. We added descriptions of this response in Section 3.4 of the SI.

[Section 3.4 of the SI is shown on page 49 of the rebuttal letter]

- Please check the Supplementary Material as in several occasions the coefficients γ are wrongly indicated as λ (see e.g., point 1. of the list at the beginning of Section 1.4).

Thanks for pointing out this error. All symbols and equations in the manuscript and SI have also been checked.

[Point 1 of the list of Section 1.4 in the SI] Basis coefficients $\{\gamma^j\}_{j=1}^{m+1}$ for all dimensionless numbers.

- There is a typo in Section 3.1: “show gradient the” should be “show the gradient descent”

Thanks for the comment. We revised this sentence.

- I was wondering whether the Algorithm 2 and 3 can be generalised for a generic parameterizations of f , rather than focusing on the polynomial one.

Yes, Algorithms 2 and 3 can be generalized for a generic parameterization of f . The proposed two-level optimization scheme decoupled the optimization of basis coefficients and fitting function parameters. Polynomial, tree-based methods (decision tree, random forest, or XGBoost), neural networks, and other fitting functions can be used to build the scaling law. In Section 2.3 of the main manuscript, we used an XGBoost instead of a polynomial as the fitting function and achieved very good fitting performance (the R^2 score in the test set is more than 0.95). We also emphasized this flexibility in the first paragraph of the Discussion section of the main manuscript.

[The first paragraph of the Discussion section of the main manuscript] Dimensionless learning with different fitting functions. The two-level optimization scheme makes dimensionless learning very flexible. The first-level scheme guarantees dimensional invariance (or dimensional homogeneity), and thus many representation learning methods can be used for the second-level scheme to capture scale-free relationships. We demonstrate polynomial and tree-based method XGBoost methods in the previous sections. However, the capability of the dimensionless learning can be improved by leveraging more methods, including deep neural networks symbolic regression, and Bayesian machine learning.

- In Algorithm 3 you consider 3^K points nearby R_{center}^2 where to compute the R^2 . Can you please provide some insights that guided your design choice (eventually formal ones)?

The reason for setting $3^K - 1$ points near the center point is to ensure each dimension can move in two directions along the coordinate axes. Another option is to use $2K$ neighboring points to make sure that the descent direction is only along with one of the coordinate axes. We conducted experiments for both, but the first option outperforms the second in terms of avoiding local optimal. We corrected the number of neighboring points in Algorithm 3 in the SI.

[The algorithm 3 in the SI] Calculate R^2 score for $3^K - 1$ nearby points in a K dimensional Cartesian coordinate system.

- Given the relation identified from data, in Section 4.1 you then simplify it removing the bias, claiming that the latter is negligible. Can you please show what is the actual impact of such reduction when validating your results?

Figure 7 of the SI shows seven sets of dimensionless numbers discovered in keyhole dynamics by the proposed method. We can see that multiplying a dimensionless number by another dimensionless variable can result in a new dimensionless number with better fitting performance. For example, in the first row of Figure 7, the scaling law is $e^* = f(Ke)$ with a 0.9826 R^2 score in the test set, while the scaling law in the second row is $\frac{r_0 V_s}{\alpha} e^* = f(\frac{r_0 V_s}{\alpha} Ke)$ and the R^2 score in the test set is slightly higher (0.9852).

To demonstrate the similarity of different dimensionless numbers in Figure 7 of the SI, when the bias of $e^* = f(Ke) = 0.12Ke - 0.3$ is removed, the R^2 score falls only slightly from 0.9826 to 0.9768.

- Can you provide a physical interpretation to the increase in the R^2 you highlight with respect to the results reported in Table 1?

Based on our dimensionless learning, we identified another identified low-dimensional pattern $\Phi = f(NEP \frac{L}{d}, NED \frac{d^3}{L^3})$ achieves an even higher R^2 (0.95), as shown in Fig. 3d of the main manuscript. The NED and NEP have been interpreted in terms of the physical meaning in the manuscript. Aside

from them, two geometrical ratios ($\frac{L}{d}$ and $\frac{d^3}{L^3}$) are involved to maximize the fitting performance. These two ratios have clear physical meanings: the term $\frac{L}{d}$ means the linear ratio of powder bed layer thickness and laser beam diameter, while the term $\frac{d^3}{L^3}$ means the volumetric ratio of laser beam diameter and powder bed layer thickness. These two ratios account for the effect of multiple-track and multiple-layer scanning. These interpretations have been added to the revised manuscript. We also modified the caption for Figures 3.c and 3.d to clearly show the identified dimensionless numbers.

[The last paragraph of Section 2.3 of the main manuscript] Another identified low-dimensional pattern $\Phi = f(\text{NEP} \frac{L}{d}, \text{NED} \frac{d^3}{L^3})$ achieves an even higher R^2 (0.95), as shown in Fig. 3d. Two geometrical ratios ($\frac{L}{d}$ and $\frac{d^3}{L^3}$) are involved to maximize the fitting performance. These two ratios have clear physical meanings: the term $\frac{L}{d}$ means the linear ratio of powder bed layer thickness and laser beam diameter, while the term $\frac{d^3}{L^3}$ means the volumetric ratio of laser beam diameter and powder bed layer thickness. These two ratios account for the effect of multiple-track and multiple-layer scanning.

[The modified Figure 3 in the main manuscript]

- The results obtained by including the additional temperature difference shown in Section 5.2 highlight that both the first and the second options attain similar performance in testing, with the one accounting for the input you claim to be negligible resulting in a slightly better fit in training. Given this result, why would one pick the second option rather than the first? Has this choice to be guided by the relatively simpler dimensionless parameters obtained in the second case? Although I think that this is the reason driving the conclusions of the authors, I think it would be clearer if they highlight this.

Thank the reviewer for pointing out this question. The reason that we chose the second option is that it involves one less variable ($T_v - T_0$) in the input dimensionless number, although the L1 norm is the same. We mentioned the detailed criteria for selecting the dimensionless numbers and scaling laws in the last paragraph of Section 1.4 of the SI.

[The second paragraph of Section 5.2 of the SI] When comparing the first two rows, the second expression (keyhole number Ke) does not include variable ($T_v - T_0$) but still has the same R^2 score in the test set. On the other hand, the expressions of the dimensionless numbers in the last three rows are more complex and perform poorly (lower R^2 score).

[The last paragraph of Section 1.4 of the SI]

Following the flowchart in Fig. 1, a few candidate dimensionless numbers and scaling laws can be obtained. Then, we select the optimal model based on the fitting performance (R^2 score) of the scaling law and the L1 norm of dimensionless numbers. We encourage a higher R^2 score and a lower L1 norm. In a few special situations, when different scaling laws have the same R^2 score and L1 norm, we choose the model with fewer variables. These criteria ensure that the scaling law is parsimonious and fits well on the dataset.

- The axis labels in figure 10 are misplaced.

Thank you for bringing this to our attention. Based on the reviewer's next comment, we corrected Figure 10 and replaced the spring-mass-damper example with another dataset. The revised figure is shown in Figure 14 of the SI.

[The replotted figure 14 of the SI]

Figure 14. Plot showing displacements for four sets of generated data for the spring-mass-damping systems.

- By looking at the dataset considered used in the example with the spring-mass-damper system the data seem noiseless. Can the author explain this, possibly introducing noise in case the data they used are noiseless? Moreover, in one of the datasets they use, the displacement is dangerously close to zero for most of the time. I believe that this dataset is not that informative (thus not be helpful in retrieving an accurate result). Can the authors comment on this?

Thanks for the questions and comments regarding the datasets. In the revised manuscript, we applied the proposed method to more cases and tested the noisy data effect on each one. Section 6 of the revised SI includes a more detailed description. We also replaced the original dataset with a more reasonable dataset by avoiding displacement near zero most of the time. Figure 14 of the revised SI shows the revised figure to show the dataset.

[The new dataset for the spring-mass-damper system problem is shown on the previous response.]

- Are the libraries and parameterisations in (34)-(35) driven by insights on the features of the true dynamics of the system? Please highlight your parameterization choices.

The previous Eqns. 34 and 35 were replaced by a new regression library in Section 6.3.2 of the SI because we updated the PDE discovery approach as a new one, which is shown in Section 2.4 of the main manuscript and Section 6 of the SI. In the new regression library of spring-mass-damper systems, we include linear and quadratic terms: x , $\frac{d^2x}{dt^2}$, x^2 , $x \frac{dx}{dt}$, $x \frac{d^2x}{dt^2}$. With these candidate terms, the R^2 score for the fitted temporary differential equation is already more than 0.99 using noisy data. Therefore, the current library includes all necessary candidate terms and we do not include any other assumptions in the choice of candidates.

In a more general and unknown problem, we suggest including linear terms at first, such as the original measurements and their first and second order derivatives. If the fitting performance is perfect, we can stop involving more terms and get the best differential equations. Otherwise, we will keep adding quadratic terms and testing the fitting performance. The library will be updated until the best performance model is discovered.

More specifically, if some terms are known to be important to the problems, one can include more specific terms and restrict the regression library to a smaller one. For example, convective terms and diffusion terms are usually important to fluid mechanics problems. It is reasonable to include these terms in the library.

Apart from some known terms, the number of learnable coefficients can be reduced if prior knowledge about the relationship between different terms exists. For example, for a 2-dimensional fluid mechanics problem, we can assume that the -convective and diffusion terms -have the same coefficients.

Detailed suggestions for the choice of candidate terms are described in Section 6.5 of the SI

[Section 6.5 of the SI] 6.5 Regression library suggestions

Building a regression library with all the required candidates relies on the researchers' experience and understanding of the problem. If some terms are known to be important to the problems, more specific terms can be included, limiting the regression library to a smaller one. For example, convective and diffusion terms are frequently used in the governing equations of fluid mechanics. It is reasonable to include these terms in the library.

In addition, if there is some prior knowledge about the relationship between different terms, the number of learnable coefficients can be reduced. For example, for a 2-dimensional fluid mechanics problem, we can assume that the two convective and diffusion terms have the same coefficients. It is worth noting that the proposed symmetric invariance does not explicitly include this physical constraint but achieves it implicitly.

In a more general and unknown problem, we suggest including linear terms first, such as the original measurements and their first and second order derivatives. If the fitting performance is ideal, we can stop involving more terms and get the best differential equations. Otherwise, we will keep adding quadratic and other high-order terms and testing the fitting performance. The library will be updated until the best performance model is discovered.

REVIEWER COMMENTS

Reviewer #1 (Remarks to the Author):

Dear authors,

Dear Editor

the answer given by the authors explains completely all my doubts and covers all my curiosity about the proposed method. The improvement done on the paper for my side is enough for its publication in NC.

Reviewer #2 (Remarks to the Author):

My main concerns about the manuscript continue to be the same, its lack of true novelty. The response of the authors only include new examples, which are nevertheless valuable, but does not clarify my initial doubts.

Reviewer #3 (Remarks to the Author):

I thank the authors for addressing many of my concerns.

At the same time, I feel that (although interesting) the additional extensive examples presented in the revised manuscript and supplementary material are not enough for the approach to be practically used (and fully understood) by others.

Although the authors have made the codes fully available, the hyper-parameters chosen by the authors still seem cherry-picked.

I strongly suggest the authors to perform and show a sensitivity analysis on the main parameters of their approaches, so that the interested reader can see what changing one parameter involves, rather than "solely trusting" the additional comments provided on the matter by the authors.

I still feel that the example regarding the mass-damper-spring system still lack a comparison with simpler (more straightforward) approaches, like least squares. Why should I pick the method proposed when I have such a simple case study? I appreciate the response of the authors on the matter, but I am still not convinced that, in such a simple case, the approach proposed by the authors have any advantage over simpler ones.

Lastly, I think the code (or parts of it) are not needed in the text, since the codes are available. I thus suggest the authors to remove them.

In several occasions, the coefficient coming from the data-driven algorithms are approximated by the authors (the round the numbers). What is the impact of such approximations? Why are you performing them rather than keep the numbers coming from the algorithms?

I did not fully understand the need for Table 7. I do not think it adds any additional insight and, thus, I think it can be safely removed.

Rebuttal letter

Reviewer #1 (Remarks to the Author)

Dear authors,

Dear Editor

the answer given by the authors explains completely all my doubts and covers all my curiosity about the proposed method. The improvement done on the paper for my side is enough for its publication in NC.

We appreciate the reviewer's instructive comments, which improved the overall quality of this work.

Reviewer #2 (Remarks to the Author)

My main concerns about the manuscript continue to be the same, its lack of true novelty. The response of the authors only include new examples, which are nevertheless valuable, but does not clarify my initial doubts.

The novelty of this paper lies in the combination of physical invariance and machine learning to achieve scientific knowledge discovery from scarce data at multiple levels: 1) dimensionless numbers at the feature level, 2) scaling laws at the algebraic equation level, and 3) governing equations at the differential equation level. To the best of our knowledge, this is the first comprehensive data-driven approach to discovering these three types of physics in a unified framework. We demonstrated these in eight challenging scientific and engineering problems. Specifically, the significance and novelty of this paper can be understood from three perspectives:

- 1. The proposed method not only identifies dominant dimensionless numbers and scaling laws underlying the data, but it is also an automatic end-to-end identification procedure for discovering new dimensionless numbers and scaling behaviors for understanding newly emerging scientific and engineering phenomena. In contrast, Buckingham's Π -theorem only reveals the sufficient (NOT necessary) number of dimensionless numbers for describing a physical system, while classical dimensional analysis can provide a large (mathematically infinite) set of candidate dimensionless groups by examining the overall units of the studied system. The entire process is time-consuming and heavily dependent on the experience of domain experts, severely limiting the application of dimensional analysis.*
- 2. The proposed method successfully addressed several fundamental issues with machine learning approaches (poor applicability on small data, limited generalization, and interpretability) by integrating dimensional invariance into machine learning. It is not purely data-driven, but rather a hybrid of cutting-edge data-driven approaches and fundamental physics. The first advantage of this configuration is that the embedded physical invariance reduces the solution space by eliminating strong dependencies between variables, allowing high-performing models to be trained with small datasets. Second, embedding dimensional invariance improves generalization. It allows us to discover universal scaling laws that scale well in data collected from experiments with various materials, process conditions, and scales. Third, in terms of interpretability, the proposed approach significantly improves the interpretability of machine learning because it modifies the model structure to explicitly discover physically interpretable dimensionless numbers, and the model output is an explicit equation.*
- 3. The proposed method provides a novel procedure, which divides the identification process of differential equations into two steps to identify consistent parameterized governing equations efficiently. The first step is to identify a temporary governing equation in which the regression coefficients can be fixed or variable depending on how the simulation or experiment parameters are set. In the next step, dimensionless learning aims to recover the expression of the varying coefficients by leveraging the dimension of these coefficients. By combining these two steps, the proposed method can efficiently obtain a consistent dimensionally homogeneous governing equation with a small amount of data. In contrast, the standard SINDy falls short of achieving a*

consistent parameterized differential equation for the same system with different parameters, while other advanced SINDy approaches couple the optimization of identifying candidate terms and parameterized coefficients, making optimization more difficult.

Reviewer #3 (Remarks to the Author)

I thank the authors for addressing many of my concerns.

At the same time, I feel that (although interesting) the additional extensive examples presented in the revised manuscript and supplementary material are not enough for the approach to be practically used (and fully understood) by others.

We appreciate the reviewer's instructive comments on our work. Inspired by the reviewer's suggestions and questions in the first round, we put a lot of effort into adding challenging problems in fluid and solid mechanics to the paper. We showcased eight challenging problems, including three case studies for scaling law discovery and five problems on identifying governing equations. There are still many scientific problems that dimensionless learning can be applied to discover their scaling laws and governing equations in future works.

Following the reviewer's comment, we have improved the description of these additional examples for clarity in the revised manuscript and Supplementary Information (SI). The revisions are shown in Section 2.4 of the main manuscript and Section 6 of the SI.

[Section 2.4 of the main manuscript]

In this section, we describe the second workflow of dimensionless learning: identifying dimensionally homogeneous differential equations and dimensionless numbers from time-varying data. This approach combines dimensionless learning with sparsity-promoting methods. Like the method discussed in Sections 2.1 to 2.3 for incorporating dimensional invariance into machine learning, we enhance the sparsity-promoting method SINDy with another fundamental physical invariance, symmetric invariance. We refer to this physically enhanced SINDy as symmetric invariant SINDy.

Fig. 4 shows a schematic of this workflow for identifying the underlying governing equation and dimensionless number(s) from simulation snapshots of the Kármán vortex street problem. This fluid mechanics problem involves three cylinders with diameters of l (see Fig. 4a). Different fluid flow patterns can be obtained through simulations by changing fluid density ρ , dynamic viscosity μ , inlet velocity V , and the pressure difference between upstream and downstream p_0 .

Figure 4. Integration of dimensionless learning with symmetric invariant SINDy for identifying the Navier-Stokes equation with Reynolds number. (a) Original data are generated from parametric simulations. To achieve symmetric invariance, another set of transformed data is obtained by flipping the original data along $y = x$. (b) The original and transformed data are concatenated for symmetric invariant SINDy, which implicitly incorporates symmetric invariance into SINDy to ensure that symmetric invariant terms have the same coefficients. (c) The identified temporary governing equations for each simulation case obtained through optimizing the symmetric invariant SINDy. Some of the coefficients are close to constant, while others vary depending on the simulation case. All the other candidate terms have zero coefficients. (d) Dimensionless learning is applied to identify an explicit expression for the varying coefficients. The parametric space to be explored includes five parameters. By incorporating dimensional invariance, we need to optimize basis coefficients $\boldsymbol{\gamma}$ and fitting coefficient β . (e) Substituting the discovered regression coefficients ($1/\text{Re}$) into the temporary governing equation. In this step, a consistent dimensionally homogeneous governing equation, which is identical to the Navier-Stokes equation in the vorticity form, is obtained.

In the first step (Fig. 4a), three CFD simulations are carried out to generate datasets for the discovery of the governing equation. The dataset for each simulation contains not only the above-mentioned geometry and fluid properties, but also time-dependent variables (i.e., velocities u and v and vorticity ω) in the spatiotemporal domain. Then, 4,000 velocity and vorticity measurements from different locations and time steps are randomly sparsely sampled. Detailed description of data generation and preprocessing can be found in Section 6.1.1 of the SI.

Next, we apply symmetric invariant SINDy on the dataset for each simulation case to discover temporal governing equations (Fig. 4b). To incorporate symmetric invariance, we flip the original data along $y = x$ for each simulation case to obtain the transformed data. This is because we assume the governing equation should be invariant to the symmetric transformation along $y = x$. This assumption helps double the dataset for temporal governing equation discovery while incurring no additional computational cost to run more simulations. More information about this operation can be found in Section 6.1.2 of the SI.

Based on these measurements, a regression library is built to identify the governing equation using linear and quadratic terms for $u, v, \omega, \frac{\partial \omega}{\partial x}, \frac{\partial^2 \omega}{\partial x^2}, \frac{\partial^2 \omega}{\partial y^2}$. The regression library contains 29 terms in total. Detailed information on candidate terms is shown in Section 6.5 of the SI.

After preparing the regression library, the proposed symmetric invariant SINDy trains all the measurements from the original and transformed data together. This operation implicitly ensures that the symmetry terms have the same coefficients. That is, the coefficients for symmetry terms are physically constrained. For example, $u \frac{\partial \omega}{\partial x}$ and $v \frac{\partial \omega}{\partial y}$ can be regarded as symmetry terms and have the same coefficient as the symmetric invariant SINDy. See Section 6.1.2 of the SI for more detailed descriptions of this operation.

By optimizing the symmetric invariant SINDy for all cases, we obtain three temporary governing equations with only four non-zero regression coefficients (Fig. 4c). ξ_{12} and ξ_{19} are identical and close to a constant, while ξ_6 and ξ_7 are also identical but vary with parameters such as ρ, μ , and so on. The following steps are to build a consistent parameterized governing equation that is valid in all simulations, as explained below.

Since the varying coefficients (ξ_6 and ξ_7) are due to changes of geometry and fluid properties, we apply dimensionless learning to find the expression for these two coefficients (Fig. 4d). The parametric space, which includes variables affecting the behavior of the dynamical system, to be explored for $\xi_6 = \xi_7$ can be expressed as follows:

$$\xi_6 = \xi_7 = f(\mu, \rho, V, l, p_0) \quad (17)$$

In contrast to the standard dimensionless learning, we simplify the representation function $f(\cdot)$ as a power law with a constant coefficient rather than a high-order polynomial, as applied in Sections 2.1 and 2.2, or an XGBoost, as used in Section 2.3. This is because parametric differential equations usually consist of derivatives and/or derivatives multiplied by variable coefficients, which are power-law functions.

In this case, we choose the pattern search-based optimization to solve Eqn. 17. A detailed description of this proposed optimization algorithm can be found in Section 3.3 of the SI. The identified expression for ξ_6 and ξ_7 is $1.083 \frac{\mu}{\rho \nu l} \approx \frac{\mu}{\rho \nu l}$, which is the reciprocal of the well-known Reynolds number $Re = \frac{\mu}{\rho \nu l}$. By substituting the constant regression coefficients for ξ_{12} and ξ_{19} and the discovered expression for ξ_6 and ξ_7 into the temporary governing equations, we obtain a consistent dimensionless governing equation in all cases as follows:

$$\frac{\partial \omega}{\partial t} = -u \frac{\partial \omega}{\partial x} - v \frac{\partial \omega}{\partial y} + \frac{1}{Re} \left(\frac{\partial^2 \omega}{\partial x^2} + \frac{\partial^2 \omega}{\partial y^2} \right) \quad (18)$$

which is identical to the well-known vorticity form of the Navier-Stokes equation. This demonstrates the effectiveness of the proposed method in discovering governing equations and dimensionless number(s).

We further apply the proposed method to data with 1% Gaussian noise. Following the same procedure, the proposed method successfully identifies the correct governing equation as Eqn. 18. The detailed results for noisy data are shown in Section 6.1.3 of the SI. More applications of the proposed method in fluid and solid mechanics and dynamics systems with and without noise are demonstrated in Section 6 of the SI.

[The entire Section 6 of the SI were refined in terms of clarity, which can be seen from the SI.]

Although the authors have made the codes fully available, the hyper-parameters chosen by the authors still seem cherry-picked.

The rationale behind selecting hyperparameters in this study can be summarized as follows: physical insights and experience, the tradeoff between accuracy and efficiency, and general trial and error selection or cross-validation.

First, based on physical insights and experience, dimensionless numbers should have simple forms. Thus, the exploration range of basis coefficients is set to $[-2, 2]$, and the grid interval can be set to 0.01, 0.1, 0.25, or 0.5. These settings help limit the power exponents of parameters in dimensionless numbers and avoid the identification of complex dimensionless numbers with many decimals.

Second, to achieve a balance between equation accuracy and algorithm efficiency, we recommend using a small grid interval (0.01 or 0.1) for problems with fewer basis coefficients (less than or equal to 4) and a large grid interval (0.25 or 0.5) for problems with more basis coefficients (more than 4). This is because small grid intervals can lead to inefficiency given a high dimensional solution space of basis coefficients. It is also worth mentioning that large grid intervals are usually sufficient to find the optimal dimensionless numbers because of the same assumption as in the first point: dimensionless numbers should be a power law with a simple form. In addition, to avoid being trapped by local minima in gradient-based and pattern search-based approaches, the number of initial points is set to 1,000 in this paper. However, because the computational cost of the proposed method is not high, more random initial points can be chosen.

Third, several hyperparameters should be determined through trial and error or cross-validation. We summarize them as below:

- 1. The L1 norm threshold for input dimensionless numbers can be different in problems. Higher L1 norm thresholds allow for the discovery of more complicated dimensionless numbers, and vice versa. Specifically, the thresholds in this study are 8, 9, and 13, respectively, for the problems in Sections 2.1 to 2.3 of the main manuscript.*
- 2. In contrast to the input dimensionless numbers, which have a higher L1 norm threshold (greater than or equal to 8), the output dimensionless numbers in this paper are suggested to have a lower L1 norm, such as 5. This is because the output dimensionless numbers are usually related to the quantity of interest / dependent variable. We prefer a simple expression for the output dimensionless numbers so that the quantity of interest is easier to understand.*
- 3. Cross-validation is an effective technique to determine hyperparameters. We demonstrate 5-fold cross-validation in determining polynomial degree in the vapor depression dynamics problem in laser-material interaction, which is shown in Fig. 7. The R^2 score of a polynomial function with a degree of three is 0.9702, which is only slightly lower than that of a polynomial function with a degree of four (0.9708). To obtain a simple scaling law without degrading performance too much, we can set the polynomial degree to three.*
- 4. It is worth mentioning that because the BFGS optimization algorithm is used in the gradient-based optimization scheme, there is no need to manually select a learning rate or gradient descent step. The BFGS algorithm is a second-order optimization algorithm that uses line search algorithms to determine the best learning rate for each iteration. If readers prefer to use other gradient descent methods, they need to select the appropriate learning rate and gradient step through trial and error.*

Figure 7. Polynomial degree identification for the keyhole dynamics problem using 5-fold cross-validation

We modified the first four paragraphs of Section 3.4 of the SI to clearly illustrate how to choose hyperparameters for the problems solved in this work.

[The first four paragraphs of Section 3.4 of the SI]

The rationale behind selecting hyperparameters in this study can be summarized as follows: physical insights and experience, the tradeoff between accuracy and efficiency, and general trial and error selection or cross-validation.

First, based on physical insights and experience, dimensionless numbers should have simple forms. Thus, the exploration range of basis coefficients is set to $[-2, 2]$, and the grid interval can be set to 0.01, 0.1, 0.25, or 0.5. These settings help limit the power exponents of parameters in dimensionless numbers and avoid the identification of complex dimensionless numbers with many decimals.

Second, to achieve a balance between equation accuracy and algorithm efficiency, we recommend using a small grid interval (0.01 or 0.1) for problems with fewer basis coefficients (less than or equal to 4) and a large grid interval (0.25 or 0.5) for problems with more basis coefficients (more than 4). This is because small grid intervals can lead to inefficiency given a high dimensional solution space of basis coefficients. It is also worth mentioning that large grid intervals are usually sufficient to find the optimal dimensionless numbers because of the same assumption as in the first point: dimensionless numbers should be a power law with a simple form. In addition, to avoid being trapped by local minima in gradient-based and pattern search-based approaches, the number of initial points is set to 1,000 in this paper. However, because the computational cost of the proposed method is not high, more random initial points can be chosen.

Third, several hyperparameters should be determined through trial and error or cross-validation. We summarize them as below:

1. The L1 norm threshold for input dimensionless numbers can be different in problems. Higher L1 norm thresholds allow for the discovery of more complicated dimensionless numbers, and vice versa. Specifically, the thresholds in this study are 8, 9, and 13, respectively, for the problems in Sections 2.1 to 2.3 of the main manuscript.
2. In contrast to the input dimensionless numbers, which have a higher L1 norm threshold (greater than or equal to 8), the output dimensionless numbers in this paper are suggested to have a lower L1 norm, such as 5. This is because the output dimensionless numbers are usually related to the quantity of interest / dependent variable. We prefer a simple expression for the output dimensionless numbers so that the quantity of interest is easier to understand.

3. Cross-validation is an effective technique to determine hyperparameters. We demonstrate 5-fold cross-validation in determining polynomial degree in the vapor depression dynamics problem in laser-material interaction, which is shown in Fig. 7. The R^2 score of a polynomial function with a degree of three is 0.9702, which is only slightly lower than that of a polynomial function with a degree of four (0.9708). To obtain a simple scaling law without degrading performance too much, we can set the polynomial degree to three.
4. It is worth mentioning that because the BFGS optimization algorithm is used in the gradient-based optimization scheme, there is no need to manually select a learning rate or gradient descent step. The BFGS algorithm is a second-order optimization algorithm that uses line search algorithms to determine the best learning rate for each iteration. If readers prefer to use other gradient descent methods, they need to select the appropriate learning rate and gradient step through trial and error.

Figure 7. Polynomial degree identification for the keyhole dynamics problem using 5-fold cross-validation

I strongly suggest the authors to perform and show a sensitivity analysis on the main parameters of their approaches, so that the interested reader can see what changing one parameter involves, rather than "solely trusting" the additional comments provided on the matter by the authors.

Following the reviewer's suggestion, we performed sensitivity analysis in Section 5.2 of the SI to quantify the sensitivity of each independent parameter.

Table 3 of the SI shows the identified dimensionless numbers after including an extra variable, the temperature difference between boiling and room temperature ($T_v - T_0$), in the input variable list. When the first two rows are compared, the second expression (keyhole number Ke) does not include the variable ($T_v - T_0$). The R^2 score of keyhole number Ke is slightly lower in the training set, but it maintains the same R^2 score in the test set and has a simpler form. The dimensionless number expressions in the last three rows, on the other hand, become more complex after involving ($T_v - T_0$) and perform poorly (lower R^2 score).

To quantitatively understand the effect of ($T_v - T_0$) on the prediction of keyhole aspect ratio e^* , we performed Sobol sensitivity analysis with an open-source python library called SALib [6,7]. We begin by calculating the ranges for each input variable by examining the lower and upper bounds for the experimental data. Then, we generate 2^{10} model inputs by using the Sobol sequence [8], which is a quasi-random low-discrepancy sequence method for generating uniform samples. Next, we build a predictive model using the input dimensionless number $\frac{\eta P}{(T_v - T_0)^{0.25} (T_l - T_0)^{0.75} \rho_p \sqrt{\alpha V_s r_0^3}}$ with the highest

R^2 scores in the training and test sets and evaluate the model predictions using the generated model inputs. Lastly, we use Sobol sensitivity analysis to analyze the model prediction and compute the sensitivity indices.

Fig. 10 shows the sensitivity analysis results for different possible variables. We can see that the sensitivity of $(T_v - T_0)$ is nearly zero, indicating that this variable has negligible influence on model prediction even though it is in the denominator of the input dimensionless number. Therefore, we choose keyhole number Ke as the input dimensionless number because it not only has a simpler form but also does not consider the unimportant variable. This also demonstrates that adding one additional variable $(T_v - T_0)$ has no effect on the dimensionless numbers discovered in this problem.

Figure 10. Sensitivity analysis for the keyhole dynamics problem with an additional variable $(T_v - T_0)$. The Sobol 1st order indices measure how a single variable influences the output variable, while the Sobol total index sums the effect of single and multiple input variables on the output variable. The large difference between the Sobol 1st order indices and the total index indicates that the variable has a strong interaction with other variables.

Similarly, we conducted sensitivity analysis for another experiment in which we add a new variable, the latent heat of melting L_m , to input independent variables. In this case, we consider the input dimensionless number $\frac{\eta P}{L_m^{0.25} \rho (C_p (T_l - T_0))^{0.75} \sqrt{\alpha V_s r_0^3}}$ in the third row of Table 4. The model associated with this input dimensionless number has high R^2 scores in the training and test sets, albeit slightly lower than that of the model associated with keyhole number Ke .

Fig. 11 shows the sensitivity analysis results after including L_m in the dimensionless number expression. We notice that the sensitivity of L_m is very close to zero, indicating that adding this variable has no effect on the model prediction. Therefore, it is reasonable to omit this variable. In this experiment, we also demonstrate that the proposed dimensionless learning is robust even when an additional variable is considered.

Figure 11. Sensitivity analysis for the keyhole dynamics problem with an additional variable L_m .

[Section 5.2 of the SI]

When the first two rows are compared, the second expression (keyhole number Ke) does not include the variable $(T_v - T_0)$. The R^2 score of keyhole number Ke is slightly lower in the training set, but it maintains the same R^2 score in the test set and has a simpler form. The dimensionless number expressions in the last three rows, on the other hand, become more complex after involving $(T_v - T_0)$ and perform poorly (lower R^2 score).

To quantitatively understand the effect of $(T_v - T_0)$ on the prediction of keyhole aspect ratio e^* , we performed Sobol sensitivity analysis with an open-source python library called SALib [6,7]. We begin by calculating the ranges for each input variable by examining the lower and upper bounds for the experimental data. Then, we generate 2^{10} model inputs by using the Sobol sequence [8], which is a quasi-random low-discrepancy sequence method for generating uniform samples. Next, we build a predictive model using the input dimensionless number $\frac{\eta P}{(T_v - T_0)^{0.25} (T_l - T_0)^{0.75} \rho_p \sqrt{\alpha V_s r_0^3}}$ with the highest

R^2 scores in the training and test sets and evaluate the model predictions using the generated model inputs. Lastly, we use Sobol sensitivity analysis to analyze the model prediction and compute the sensitivity indices.

Fig. 10 shows the sensitivity analysis results for different possible variables. We can see that the sensitivity of $(T_v - T_0)$ is nearly zero, indicating that this variable has negligible influence on model prediction even though it is in the denominator of the input dimensionless number. Therefore, we choose keyhole number Ke as the input dimensionless number because it not only has a simpler form but also does not consider the unimportant variable. This also demonstrates that adding one additional variable $(T_v - T_0)$ has no effect on the dimensionless numbers discovered in this problem.

Figure 10. Sensitivity analysis for the keyhole dynamics problem with an additional variable ($T_v - T_0$). The Sobol 1st order indices measure how a single variable influences the output variable, while the Sobol total index sums the effect of single and multiple input variables on the output variable. The large difference between the Sobol 1st order indices and the total index indicates that the variable has a strong interaction with other variables.

[Section 5.2 of the SI]

We also conduct sensitivity analysis for another experiment in which we add a new variable, the latent heat of melting L_m , to input independent variables. In this case, we consider the input dimensionless number $\frac{\eta P}{L_m^{0.25} \rho (C_p (T_1 - T_0))^{0.75} \sqrt{\alpha V_s r_0^3}}$ in the third row of Table 4. The model associated with this input dimensionless number has high R^2 scores in the training and test sets, albeit slightly lower than that of the model associated with keyhole number Ke .

Fig. 11 shows the sensitivity analysis results after including L_m in the dimensionless number expression. We notice that the sensitivity of L_m is very close to zero, indicating that adding this variable has no effect on the model prediction. Therefore, it is reasonable to omit this variable. In this experiment, we also demonstrate that the proposed dimensionless learning is robust even when an additional variable is considered.

Figure 11. Sensitivity analysis for the keyhole dynamics problem with an additional variable L_m .

I still feel that the example regarding the mass-damper-spring system still lack a comparison with simpler (more straightforward) approaches, like least squares. Why should I pick the method

proposed when I have such a simple case study? I appreciate the response of the authors on the matter, but I am still not convinced that, in such a simple case, the approach proposed by the authors have any advantage over simpler ones.

The proposed method provides a novel procedure, which divides the identification process of differential equations into two steps to identify consistent parameterized governing equations efficiently. The first step is to identify a temporary governing equation in which the regression coefficients can be fixed or variable depending on how the simulation or experiment parameters are set. In the next step, dimensionless learning aims to recover the expression of the varying coefficients by leveraging the dimension of these coefficients. By combining these two steps, the proposed method can efficiently obtain a consistent dimensionally homogeneous governing equation with a small amount of data. In contrast, the standard SINDy falls short of achieving a consistent parameterized differential equation for the same system with different parameters. For example, the governing equation for the spring-mass-damper system is $\frac{dx}{dt} = -\frac{k}{c}x - \frac{m}{c}\frac{d^2x}{dt^2}$. If we use different parameters (damping coefficient c , spring constant k , or mass m) in this system, SINDy can only provide scalar coefficients for x and $\frac{d^2x}{dt^2}$ rather than the expressions $-\frac{k}{c}$ and $-\frac{m}{c}$, respectively. Other advanced SINDy approaches deal with this issue by multiplying the candidate terms by a set of predetermined parameters. Although these approaches can address this inconsistent governing equation problem, it couples the optimization of identifying candidate terms and parameterized coefficients, making the optimization more difficult. If there are many combinations of parametric derivative or non-derivative terms, this problem can become more difficult and unmanageable.

In the Discussion Section of the main manuscript, we added descriptions to make the distinctions and advantages of differential equation discovery clearer for readers.

[The fifth paragraph of the Discussion Section of the main manuscript]

The proposed method divides the identification process of differential equations into two steps to identify consistent parameterized governing equations efficiently. The first step is to identify a temporary governing equation in which the regression coefficients can be fixed or variable depending on how the simulation or experiment parameters are set. In the next step, dimensionless learning aims to recover the expression of the varying coefficients by leveraging the dimension of these coefficients. By combining these two steps, the proposed method can efficiently obtain a consistent dimensionally homogeneous governing equation with a small amount of data. In contrast, the standard SINDy falls short of achieving a consistent parameterized differential equation for the same system with different parameters. For example, the governing equation for the spring-mass-damper system is $\frac{dx}{dt} = -\frac{k}{c}x - \frac{m}{c}\frac{d^2x}{dt^2}$. If we use different parameters (damping coefficient c , spring constant k , or mass m) in this system, SINDy can only provide scalar coefficients for x and $\frac{d^2x}{dt^2}$ rather than the expressions $-\frac{k}{c}$ and $-\frac{m}{c}$, respectively. Other advanced SINDy approaches deal with this issue by multiplying the candidate terms by a set of predetermined parameters. Although these approaches can address this inconsistent governing equation problem, it couples the optimization of identifying candidate terms and parameterized coefficients, making the optimization more difficult. If there are many combinations of parametric derivative or non-derivative terms, this problem can become more difficult and unmanageable.

Lastly, I think the code (or parts of it) are not needed in the text, since the codes are available. I thus suggest the authors to remove them.

We agree with the reviewer and have removed the codes from Section 1.2 of the SI.

In several occasions, the coefficient coming from the data-driven algorithms are approximated by the authors (the round the numbers). What is the impact of such approximations? Why are you performing them rather than keep the numbers coming from the algorithms?

There are two reasons to round the learned coefficients. First, rounding the numbers helps to compare the learned and ground truth coefficients. Second, we assumed that the learned equations and coefficients should be simple and parsimonious, so we rounded the coefficients. However, at the same time, we keep the learned coefficients in the main manuscript and SI so that readers can compare the original learned coefficients and the rounded coefficients.

I did not fully understood the need for Table 7. I do not think it adds any additional insight and, thus, I think it can be safely removed.

Thanks for the suggestion. The table shows how the symmetrically transformed data was related to the original data. The original measurements were $t, x, y, u, v,$ and ω , while the transformed data $t, x', y', u', v',$ and ω' were expressed as $t, y, x, v, u,$ and $-\omega$. As suggested by the reviewer, we removed the table and modified Figure 13 of the SI to clearly show the data transformation. The updated Figure 13 is shown below:

Figure 13: Concatenating the regression library for symmetric invariant SINDy from the original and transformed data. Columns of the same color have the same absolute value.

REVIEWER COMMENTS

Reviewer #3 (Remarks to the Author):

I thank the authors for answering my previous questions.

I have only one question left, but I think it is crucial that the authors clarify this point in the manuscript.

Are the data you use generally noise-free, since you generate them through simulations? If they are indeed noiseless, how sensitive are the performance of the proposed approach to noise? It is fundamental that the authors clarify this point, since measured data are always corrupted by noise and, thus, the approach might not be truly data-driven but only simulation-driven if it is not robust to noise.

Rebuttal letter

Reviewer #3 (Remarks to the Author):

I thank the authors for answering my previous questions.

I have only one question left, but I think it is crucial that the authors clarify this point in the manuscript.

Are the data you use generally noise-free, since you generate them through simulations? If they are indeed noiseless, how sensitive are the performance of the proposed approach to noise? It is fundamental that the authors clarify this point, since measured data are always corrupted by noise and, thus, the approach might not be truly data-driven but only simulation driven if it is not robust to noise.

We agree with the reviewer that the noise in data is an important factor to consider when assessing the proposed method in this work. To demonstrate this, we already used the experimental data (including noise) and/or simulation data (including the added Gaussian noise) in different case studies throughout the current paper and SI document.

First, we demonstrated the method for discovering scaling laws and dimensionless numbers in three challenging problems using noisy experimental measurements collected from the literature in Sections 2.1 to 2.3. Even with the noisy experimental data, the method achieves high fitting performance in both training and test sets (all R^2 scores are higher than 0.95).

Second, in order to discover differential equations, we not only tested the method on noise-free simulation data, but also added Gaussian noise to the simulation data to demonstrate the performance of the proposed method. The noisy data analyses are carried out on five differential equations: the Navier-Stokes equation (0.5 % Gaussian noise), the Euler equation (1%), the vorticity equation (1%), the governing equations for spring-mass-damper systems (4%), and dynamic loading beam systems (2%). Even with the noisy data effect, the method successfully identifies the true governing equations.

We summarized the details of the data and approach for all the case studies in a single table to avoid confusion for the readers.

Table 1. Summary of the demonstrated case studies using the proposed method in this paper. It provides information regarding the data, noise level, and approach to different problems demonstrated in different sections in the paper and SI.

Type of knowledge discovery	Case study	Optimization method	Data Source	Noise in data	Identified Π	Identified model/equation	Sec. No.
	Turbulent Rayleigh-Bénard convection	Grid search; Gradient descent; Pattern search;	Experiments (literature)	Yes	Ra, Nu	Polynomial	Sec. 2.1
Scaling law discovery	Vapor depression dynamics	Grid search; Gradient descent; Pattern search;	Experiments (literature)	Yes	Ke, e^*	Polynomial ($e^* = 0.12Ke - 0.30$)	Sec. 2.2
	Porosity formation in 3D printing	Pattern search;	Experiments (literature)	Yes	$\frac{d^3}{L^3}$, $\frac{L}{d}$, $\frac{L}{d}$	XGBoost (Tree-based model)	Sec. 2.3
	Vorticity equation	SINDy + dimensionless learning;	Simulation	1%	Re	$\frac{\partial \omega}{\partial t} = -u \frac{\partial \omega}{\partial x} - v \frac{\partial \omega}{\partial y} + \frac{1}{Re} \left(\frac{\partial^2 \omega}{\partial x^2} + \frac{\partial^2 \omega}{\partial y^2} \right)$	Sec. 2.4 & SI Sec. 7.1
	Navier-Stokes equation	SINDy + dimensionless learning;	Simulation	0.5%	Re, Eu	$\frac{\partial u^*}{\partial t} = -u \frac{\partial u^*}{\partial x} - v \frac{\partial u^*}{\partial y} - Eu \frac{\partial p^*}{\partial x} + \frac{1}{Re} \left(\frac{\partial^2 u^*}{\partial x^2} + \frac{\partial^2 u^*}{\partial y^2} \right)$	SI Sec. 7.2
Differential equation discovery	Euler equation	SINDy + dimensionless learning;	Simulation	1%	Eu	$\frac{\partial u^*}{\partial t} = -u \frac{\partial u^*}{\partial x} - v \frac{\partial u^*}{\partial y} - Eu \frac{\partial p^*}{\partial x}$	SI Sec. 7.2
	Spring-mass-damper system	SINDy + dimensionless learning;	Simulation	4%	N/A	$\frac{dx}{dt} = -\frac{k}{c}x - \frac{m}{c} \frac{d^2x}{dt^2}$	SI Sec. 7.3
	Dynamic loading beam systems	SINDy + dimensionless learning;	Simulation	2%	Rn	$\frac{\partial^2 M^*}{\partial x^{*2}} = Rn \frac{\partial v^*}{\partial t^*}$	SI Sec. 7.4

We added this table in Section 2 of the Supplementary Information (SI) document and referred this table to the discussion section in the main manuscript.

REVIEWERS' COMMENTS

Reviewer #3 (Remarks to the Author):

Thank you for addressing my last concern. I think that the paper can now be accepted for publication.